# A Survey on Behavioral Data Representation Learning

## Abstract

Behavioral data, reflecting dynamic and complex interactions among entities, are pivotal for advancing multidisciplinary research and practical applications. Effective modeling and representation of behavioral data facilitate enhanced understanding, predictive analytics, and informed decision-making across diverse domains. This paper presents a comprehensive taxonomy of behavioral data representation learning methods, categorized by data modalities: tabular data, event sequences, dynamic graphs, and natural language. Within each category, we further dissect methods based on distinct modeling strategies and capabilities, and provide detailed reviews of their developments. Additionally, we extensively discuss significant downstream applications, datasets, and benchmarks, highlighting their roles in guiding methodological development and evaluating performance. To support further exploration in behavioral data representation learning, we release a continuously maintained repository at Anonymous GitHub that curates the methods and papers covered in this survey.

## 1 Introduction

Behavioral data capture intricate and dynamic interactions between individuals and their environments (Gomez-Marin et al., 2014; Wilson & Collins, 2019; von Ziegler et al., 2021), serving as foundational evidence across multiple disciplines such as psychology, neuroscience, social sciences, economics, and artificial intelligence (Gomez-Marin et al., 2014; He et al., 2023; Purificato et al., 2024). Therefore, they are inherently rich, relational, and dynamic, often described in various data modalities such as tabular data, event sequences, dynamic graphs, and textual data.

Accurate modeling of behavioral data is crucial as it enables a deeper understanding of underlying behavioral mechanisms, facilitates predictive analytics, and informs practical interventions in diverse areas such as marketing, policy-making, financial risk control, and social network analysis. Given its complexity, behavioral data often exhibit high dimensionality, temporal dependencies, and relational structures, necessitating sophisticated representation learning methods.

Representation learning of behavioral data involves transforming raw behavioral inputs into structured, compact representations, i.e., embeddings, that capture essential behavioral patterns suitable for computational models. Unlike traditional feature extraction methods that rely heavily on manual intervention, representation learning leverages advanced machine learning and deep learning techniques to automatically distill informative features directly from complex datasets.

While this survey concentrates on learned behavioral representation, predictions including forecasting, classification, recommendation, link prediction, or language modeling are frequently used as training objectives or empirical proxies (Bengio, 2012; Bengio et al., 2013). We therefore include both behavior-specific methods and general representation learning methods that can serve behavioral analysis (Jiao et al., 2025). Our inclusion criterion is not whether a method was designed exclusively for behavioral data, but whether its modeling assumptions address recurring challenges in behavioral data: heterogeneous attributes, sparse observations, irregular temporal dynamics, evolving interactions among entities, semantic intent, and multi-modal context (Shwartz-Ziv & Armon, 2022; Sun et al., 2019b; Rossi et al., 2020; Feuerriegel et al., 2025). This perspective allows the survey to act as a practical guide for researchers who need to choose representations for behavioral analysis rather than a catalogue of methods tied to a single application domain.

To provide a systematic understanding of behavioral data representation learning, we propose a taxonomy structured around the different forms in which behavioral data can be represented: **(1) Tabular Data**: This form represents behaviors as structured tables, characterized by static or aggregated features such as demographic attributes, user preferences, transaction frequencies, or sensor readings. Tabular data often enables efficient processing and straightforward interpretability, making it ideal for applications like intelligent monitoring, risk assessment, and market analytics. Tabular data modeling has undergone significant evolution, progressing from early tree-based methods to a competitive landscape dominated by both deep learning approaches and tree-based models. More recently, the emergence of large language models has introduced a new paradigm, further advancing the field. **(2) Event Sequences**: Behaviors are frequently recorded as sequential events, each associated with a timestamp, providing insights into temporal dependencies, user trajectories, and action progressions. Common examples include clickstreams from websites, transaction logs, medical events, and sensor data sequences. Representation learning methods for event sequences usually utilize sequential modeling approaches, such as Recurrent Neural Networks (RNNs) (Cho et al., 2014) and Transformers (Vaswani et al., 2017), to capture inherent sequential dynamics and temporal patterns. **(3) Dynamic Graphs**: Many behavioral phenomena naturally form dynamic relational structures, representing interactions among entities evolving over time, such as social networks, financial transactions, and collaborative systems. Dynamic graphs capture complex relational dynamics, enabling detailed analysis of interactions and structural changes. Representation learning methods for dynamic graphs employ Graph Neural Networks (GNNs) (Scarselli et al., 2009), integrated with recurrent models (Cho et al., 2014; Graves, 2012), attention mechanisms (Vaswani et al., 2017), and memory architectures (Rossi et al., 2020), to accurately encode evolving graph structures and temporal dependencies. **(4) Textual Data**: Behavioral data often manifest through textual interactions and documents, including online reviews, chat logs, social media content, and textual financial disclosures. Textual data inherently contains rich semantic information reflecting sentiments, intents, and contextual interactions. Representation learning in this domain leverages advanced large-scale pre-trained Language Models (LMs), such as BERT, GPT (OpenAI, 2023), and specialized variants like FinBERT (Huang et al., 2023a), to effectively extract contextual and semantic insights, facilitating sentiment analysis, intention recognition, and behavioral prediction tasks.

Within each form, we further subdivide representation learning methods based on specific architectural choices, temporal modeling strategies, or the integration of external domain knowledge and multi-modal fusion capabilities.

Additionally, we extensively explore significant downstream applications, including personalized recommendation systems, financial risk assessment, anomaly and fraud detection, social network analytics, adaptive educational technologies, healthcare monitoring, and behavioral interventions. Each application domain underscores the practical implications, efficacy, and adaptability of behavioral data representation learning methodologies. To facilitate research and practical applications, we further provide an overview of commonly used datasets and benchmarks, outlining their characteristics, scales, domains of applicability, and evaluation standards. This comprehensive survey approach enables clearer alignment between methodological choices, application-specific goals, and available data resources. We summarize our contributions as follows:

- **Novel Taxonomy**: We introduce a new and comprehensive taxonomy for behavioral data representation learning. First, from the perspective of data modalities, we categorize representation methods into distinct classes based on the form of the input behavioral data, including tabular data, event sequences, dynamic graphs, and textual data, and we further provide within-modality classifications. This modality-based taxonomy helps researchers efficiently select appropriate methods according to data characteristics and application scenarios. Second, we classify methods according to various downstream tasks and applications, enabling clear alignment between methodological choices and specific analytical goals.

- **Comprehensive Methodological Reviews**: We provide a thorough and detailed review of the state-of-the-art methods within each category, highlighting their core principles, underlying assumptions, key innovations, and limitations. Our comprehensive analysis includes critical discussions of model architectures, training paradigms, and evaluation metrics, facilitating a deep understanding of existing approaches and guiding future methodological innovations.

- **Integration of Emerging Technologies**: We critically analyze the impact and potential integration of emerging technologies on behavioral data representation learning. Furthermore, we outline promising future directions and open challenges, emphasizing how interdisciplinary advancements can further enhance behavioral representation learning and expand its applicability and effectiveness across various practical and scientific domains.

## 2 Survey Methodology

We collected the literature covered in this survey through an iterative search and screening process. Our primary sources included peer-reviewed conference and journal papers, complemented by influential preprints when a formal publication was not yet available or when the preprint had already become widely used in the community.

In practice, we searched major scholarly indexes and repositories including Google Scholar, DBLP, Semantic Scholar, and arXiv, and prioritized venues that frequently publish work on behavioral data modeling and representation learning.

Our search process was guided by the taxonomy developed in this survey and combined broad queries with modality- and task-specific keywords. Representative search terms included "behavioral data representation learning", "user behavior modeling", "tabular data representation", "event sequence modeling", "temporal graph", "dynamic graph learning", "temporal interaction graph", "textual behavior modeling", "LLM for user modeling", and related application-oriented terms such as recommendation, fraud detection, healthcare, and social media analysis.

We then expanded the candidate set by following the references of key papers and adjacent surveys. We included papers that proposed, substantially adapted, or systematically evaluated representation learning methods for behavioral data, as well as benchmark and dataset papers that are central to empirical evaluation.

Because the field has accelerated rapidly in the large-model era, our coverage focuses primarily on work published from 2016 to early 2026, while retaining earlier seminal papers when they are necessary to explain the historical foundations of later methods.

## 3 Definitions

### 3.1 Human Behavior

Human behavior refers to the observable actions or reactions of a person in response to its environment or internal state. Behavior is not a static or isolated event; it emerges from a complex interplay of factors and unfolds over time (Gomez-Marin et al., 2014). This highlights that behavior is inherently relational, arising through interactions between an individual (i.e., the behavioral subject) and its surroundings (i.e., target entities, including other people), and is dynamic, progressing as a sequence of events rather than a single event. For example, in social situations multiple individuals may dynamically interact and display species-specific behaviors, with each person's actions influencing the others' in real time. Behavior is also high-dimensional in nature. Even a single behavioral episode can be described by numerous variables (positions of limbs, neural signals, environmental context, etc.) (von Ziegler et al., 2021). This high dimensionality means behavioral data can be extremely rich, but it also poses challenges for analysis and interpretation.

### 3.2 Behavioral Data

Behavioral data are the recorded or measured manifestations of behavior, referring to data collected to represent what an individual does. Such data are grounded in observable actions or outputs of the subject. Behavioral data can take many forms depending on how the behavior is observed and logged. The key aspect is that the behavior has been externalized into a digital or structured form that can be stored and analyzed. In early behavioral science, data often came from manual observation notes or simple counts of events, whereas today they are frequently complex digital datasets (potentially combining multiple channels) (Wilson &

Collins, 2019). Regardless of the format, behavioral data serve as the empirical evidence of behavior, which researchers can analyze to draw inferences about underlying processes or to predict future actions.

Compared with other static observational data, behavioral data have several recurring properties that directly shape representation learning. First, behavior is often *temporally situated*: the same action may imply different intent depending on order, recency, duration, or inter-event interval (Li et al., 2020a; Du et al., 2016; Mei & Eisner, 2017). Second, behavior is frequently *relational*: purchases, communications, transactions, and social actions connect multiple entities, so the representation of one subject may depend on its interaction context (Kumar et al., 2019; Rossi et al., 2020; Xu et al., 2020). Third, behavioral records are usually *heterogeneous and incomplete*: they mix numerical, categorical, textual, and relational signals, while also suffering from missing values, sparse histories, and unequal observation frequency (Shwartz-Ziv & Armon, 2022; Yoon et al., 2020; Bahri et al., 2022). Fourth, behavior is *non-stationary*: preferences, health status, market regimes, and social roles change over time, a difficulty explicitly targeted by recent phase-aware and frequency-aware behavioral encoders (Mohapatra et al., 2025; Li et al., 2025b). These properties explain why the survey also covers methods beyond behavior-specific models: a general method becomes relevant when it provides a representation capacity needed by behavioral data, such as feature interaction modeling, temporal state tracking, relational aggregation, semantic encoding, or cross-modal alignment.

## 3.3 Representation Learning

Representation learning aims to automatically learn effective and generalizable feature representations from raw data through machine learning or deep learning methods. It transforms high-dimensional, complex, and difficult-to-process raw data into more compact, low-dimensional, and semantically rich vector representations. These learned representations can not only significantly improve the performance of downstream tasks, such as classification and relation prediction, but also reduce the reliance on manual feature engineering during analysis. Representation learning has achieved remarkable success across a wide range of domains, including Computer Vision (CV), Speech Processing, and Natural Language Processing (NLP).

In recent years, with the rapid development and maturation of deep learning research, particularly the emergence of Convolutional Neural Networks (CNNs) (Goodfellow et al., 2016), RNNs (Cho et al., 2014), GNNs (Scarselli et al., 2009), and self-attention mechanisms (Vaswani et al., 2017), the expressive power of representation learning for modeling diverse data types has been greatly enhanced. This progress has further provided powerful tools for the analysis of behavioral data.

## 3.4 Behavioral Data Representation Learning

Behavioral data representation learning is the process of encoding raw behavioral data into structured, compact representations (often low-dimensional vectors or embeddings) that are suitable for computational modeling and analysis. It enables models to automatically learn the salient features or abstract factors of the data directly from the data itself, rather than relying on manual feature selection or domain-specific heuristics (Bengio, 2012; Bengio et al., 2013).

In conventional analyses, human experts must design specific behavioral metrics or features a priori (for example, deciding to measure the frequency of certain actions or the duration of particular events). As behavioral datasets grow more complex and high-dimensional, this manual process becomes a bottleneck: many hand-crafted features do not capture the rich, hierarchical structure of behavior, and creating new features for each problem is labor-intensive and not scalable. Along with the development of machine learning and deep learning, researchers have started to use these emerging techniques to discover a latent encoding of behavioral patterns without explicit human supervision, which enables models to automatically capture complex, non-linear temporal dependencies, reduce reliance on handcrafted features, and generalize across diverse behavioral scenarios (Goodfellow et al., 2016; Bengio et al., 2021).

Many behavioral representation learning methods are optimized through downstream objectives, including next-event prediction, classification, ranking, link prediction, reconstruction, or language modeling (Sun et al., 2019b; Rossi et al., 2020; Devlin et al., 2019). In this survey, we treat these objectives as mechanisms for shaping and evaluating representations rather than as the final scope of the survey (Bengio, 2012; Bengio

et al., 2013). A useful behavioral representation should encode the aspects of behavior needed for later analysis: stable attributes when profiling is sufficient, temporal order when trajectories matter, exact timestamps when irregular timing is informative, relational context when interactions among entities drive the target, and semantics when intent or explanation is central. Downstream performance remains important evidence, but it should be interpreted together with what the representation preserves and what it discards.

## 3.5 Challenges and Representation Capacities

The diversity of behavioral data can be summarized as a set of representational requirements. Aggregated attributes require models that capture heterogeneous feature interactions and remain robust to missing or noisy values (Chen & Guestrin, 2016; Ke et al., 2017; Grinsztajn et al., 2022). Temporally ordered actions require encoders that preserve order, recency, duration, and long-range dependencies (Kang & McAuley, 2018; Sun et al., 2019b; Li et al., 2020a). Irregular event streams require continuous-time or time-aware representations that avoid imposing artificial time intervals (Du et al., 2016; Mei & Eisner, 2017; Zuo et al., 2020). Multi-entity interactions require relational representations that aggregate information from evolving neighborhoods or typed relations (Kumar et al., 2019; Rossi et al., 2020; Xu et al., 2020; Yu et al., 2023b). Textual traces require semantic encoders that recover intent, sentiment, explanation, and contextual meaning (Devlin et al., 2019; OpenAI, 2023; Feuerriegel et al., 2025). Multi-modal behavioral systems require alignment mechanisms that combine these signals without collapsing them into a single lossy format (Han et al., 2024; Zhang et al., 2025a; Fu et al., 2025).

This mapping from unique challenge to representation capacity motivates our taxonomy. Tabular methods treat behavior as an aggregated feature profile; event-sequence methods treat behavior as an ordered trajectory; dynamic graph methods treat behavior as evolving interactions among entities; and textual or LLM-based methods treat behavior as semantic evidence or as a bridge across modalities. The same real-world dataset can often be represented in more than one form, so the taxonomy should be read as a guide to modeling assumptions.

## 3.6 Forms of Behavioral Data and Their Representations

Behavioral data can be represented or constructed in various forms or modalities depending on how it is measured and what aspects of behavior are of interest. Each modality may require different representation learning techniques. We outline common data modalities for behavior and how they capture behavioral information.

**Static Feature Vectors**: One straightforward way to represent behavior is as a vector of features summarizing an individual's behavioral tendencies. In this modality, the behavior is reduced to a defined set of attributes or metrics. Such a representation treats behavioral data as a point in a multi-dimensional feature space. Historically, many behavioral studies relied on this kind of summarization – for example, recording the number of online purchases, the average session duration on a platform, or the frequency of logins, which yields numeric descriptors that are relatively easy to analyze statistically. The advantage of static feature vectors is their simplicity and interpretability; however, they may omit the temporal dynamics or context of the behavior.

**Event Sequences**: Human behavior often unfolds as a sequence of discrete, temporally ordered events, which may occur sporadically or in bursts. This could be a sequence of timestamped events or state changes generated over time by one or more sources. Representing behavior as a sequence preserves the temporal dynamics like the order and timing of events, which are crucial for understanding patterns such as routines, progressions of actions, or cause-and-effect in interactions. Event sequence representations enable the use of sequence-modeling techniques, such as Markov Chain Models (Eddy, 1996), RNNs (Cho et al., 2014; Graves, 2012) or Transformer and its variants (Sun et al., 2019b), to learn temporal patterns in the behavior.

**Dynamic Graphs**: Some behavioral data are best described not just as a linear sequence but as a set of entities and their interactions evolving over time. Graph-based representations are useful when behavior involves relational structure, such as social interactions or complex event dependencies. In a dynamic graph, nodes usually represent entities, and edges represent interactions or temporal relations between these

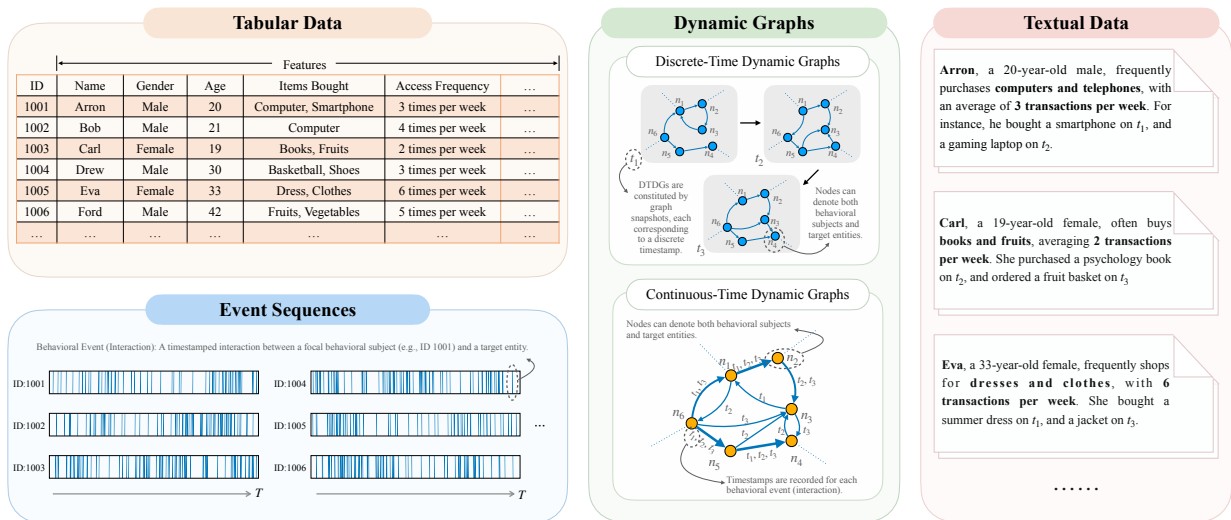

Figure 1: Taxonomy of behavioral data: behavioral data can take various forms, including tabular data, event sequences, dynamic graphs, and textual data.

nodes (Kumar et al., 2019). By applying dynamic graphs, a complex action sequence becomes a structured graph rather than a simple timeline, and analyzing the behavior can involve graph matching or graph neural network algorithms to identify patterns. Dynamic graph representations can naturally capture concurrent events (when multiple things happen at once) and long-range dependencies.

**Multi-modal Representations**: Real-world behavior often spans multiple data modalities at once. Many forms of behavioral data are inherently linguistic, such as dialogue transcripts, social media interactions, textual evaluations, or narrative logs of user interactions. These textual behavioral data inherently embed rich semantic information including emotional states, intentions, motivations, and subjective experiences (Feuerriegel et al., 2025). Moreover, recent advancements in large-scale pre-trained language models, such as GPT models (Radford et al., 2019; Brown et al., 2020; OpenAI, 2023), have profoundly changed the landscape (Xiong et al., 2025). These models enable not only direct analysis of inherently textual behavioral data but also allow the transformation of traditionally non-linguistic behavioral datasets into natural language (Feuerriegel et al., 2025).

## 4 Taxonomy of Behavioral Data Representation Learning

Based on the different modalities of behavioral data introduced in Sec. 3.6, this section presents representation learning methods tailored to each modality and provides a finer-grained taxonomy within each category.

Our modality-based organization should be understood as a taxonomy of representation assumptions. Tabular methods assume that the relevant behavior can be summarized into feature profiles (Shwartz-Ziv & Armon, 2022; Grinsztajn et al., 2022); event-sequence methods assume that order and temporal context carry the main signal (Kang & McAuley, 2018; Sun et al., 2019b; Li et al., 2020c); dynamic graph methods assume that evolving interactions among entities are central (Kumar et al., 2019; Rossi et al., 2020; Xu et al., 2020; Yu et al., 2023b); and textual or LLM-based methods assume that behavioral meaning can be represented through language or enriched by semantic knowledge (Devlin et al., 2019; OpenAI, 2023; Feuerriegel et al., 2025). These assumptions determine what each method family preserves, what it abstracts away, and which behavioral analysis scenarios it naturally supports. Accordingly, in the following subsections we keep the existing within-modality classifications, but discuss each family in terms of the behavioral representation problem it addresses, the trade-offs it introduces, and the kinds of downstream analyses for which its learned representations are most useful. A consolidated cross-modal comparison covering the technical building

Table 1: Tabular data representation learning methods.

| Method | Training Paradigm | Model Architecture | Year |
|---|---|---|---|
| Tree-based Methods | | | |
| XGBoost (Chen & Guestrin, 2016) | Supervised learning | Tree | 2016 |
| LightGBM (Ke et al., 2017) | Supervised learning | Tree | 2017 |
| CatBoost (Prokhorenkova et al., 2018) | Supervised learning | Tree | 2018 |
| Deep Learning-based Methods | | | |
| Wide&Deep (Cheng et al., 2016) | Supervised learning | GLM+MLP | 2016 |
| DeepFM (Guo et al., 2017) | Supervised learning | FM+MLP | 2017 |
| xDeepFM (Lian et al., 2018) | Supervised learning | FM+MLP+CIN | 2018 |
| TabNN (Ke et al., 2018) | Supervised learning | GBDT+MLP | 2018 |
| RLN (Shavitt & Segal, 2018) | Supervised learning | MLP | 2018 |
| NODE (Popov et al., 2020) | Supervised learning | NODE | 2020 |
| SuperTML (Sun et al., 2019a) | Supervised learning | CNN | 2019 |
| TabNet (Arik & Pfister, 2021) | Supervised + Self-supervised learning | TabNet | 2021 |
| DeepGBM (Ke et al., 2019) | Supervised + Online learning | DeepGBM | 2019 |
| NON (Luo et al., 2020) | Supervised learning | NON | 2020 |
| DNF-Net (Abutbul et al., 2020) | Supervised learning | DNF-Net | 2020 |
| VIME (Yoon et al., 2020) | Self-supervised + Semi-supervised learning | VIME | 2020 |
| TabTransformer (Huang et al., 2020a) | Supervised + Semi-supervised learning | Transformer | 2020 |
| ARM-Net (Cai et al., 2021) | Supervised learning | ARM-Net | 2021 |
| NPT (Kossen et al., 2021) | Supervised learning | Transformer | 2021 |
| Regularized DNNs (Kadra et al., 2021) | Supervised learning | MLP | 2021 |
| Boost-GNN (Ivanov & Prokhorenkova, 2021) | Supervised learning | GBDT+GNN | 2021 |
| DNN2LR (Liu et al., 2020) | Supervised learning | DNN2LR | 2020 |
| IGTD (Zhu et al., 2021) | Supervised learning | IGTD+CNN | 2021 |
| FT-Transformer (Gorishniy et al., 2021) | Supervised learning | Transformer | 2021 |
| SAINT (Somepalli et al., 2021) | Supervised + Self-supervised learning | SAINT | 2021 |
| SCARF (Bahri et al., 2022) | Self-supervised learning | MLP | 2022 |
| GANDALF (Joseph & Raj, 2022) | Supervised learning | GFLU | 2022 |
| TabDDPM (Kotelnikov et al., 2023) | Unsupervised learning | Diffusion Model | 2023 |
| PTab (Liu et al., 2022b) | Supervised + Self-supervised learning | BERT | 2022 |
| TabCBM (Zarlenga et al., 2023) | Supervised learning | MLP | 2023 |
| Trompt (Chen et al., 2023a) | Supervised + Prompt learning | MLP | 2023 |
| HYTREL (Chen et al., 2023b) | Supervised + Self-supervised learning | HYTREL | 2023 |
| ReConTab (Chen et al., 2023c) | Self-supervised | Transformer | 2023 |
| XTab (Zhu et al., 2023a) | Supervised + Self-supervised learning | Transformer | 2023 |
| IATTN (Thielmann et al., 2024) | Supervised learning | Transformer | 2024 |
| MambaTab (Ahamed & Cheng, 2024) | Supervised learning | Mamba | 2024 |
| GCondNet (Margeloiu et al., 2024) | Supervised learning | MLP | 2024 |
| BiSHop (Xu et al., 2024) | Supervised learning | BiSHop | 2024 |
| LF-transformer (Na et al., 2024) | Supervised learning | Transformer | 2024 |
| TabR (Gorishniy et al., 2024) | Supervised learning | TabR | 2024 |
| TP-BERTa (Yan et al., 2024) | Supervised + Self-supervised learning | BERT | 2024 |
| CARTE (Kim et al., 2024) | Supervised + Self-supervised learning | Transformer | 2024 |
| SwitchTab (Wu et al., 2024a) | Self-supervised learning | Transformer | 2024 |
| DP-2Stage (Afonja et al., 2025) | Supervised learning | Transformer | 2025 |
| TARTE (Kim et al., 2025) | Supervised + Self-supervised learning | Transformer | 2025 |
| KumoRFM (Fey et al., 2025) | Self-supervised learning + In-context learning | Transformer | 2025 |
| TabPFN (Hollmann et al., 2025) | Self-supervised learning + Prior-data pre-training | Transformer | 2025 |
| TDColER (Kang et al., 2025) | Supervised + Unsupervised learning | Distillation | 2025 |
| LLM-driven Methods | | | |
| TAPAS (Herzig et al., 2020) | Self-supervised learning | Bert | 2020 |
| TAPEX (Liu et al., 2022c) | Supervised + Self-supervised learning | Transformer | 2022 |
| TabLLM (Hegselmann et al., 2023) | Supervised + Self-supervised learning | Transformer | 2023 |
| cTBLS (Sundar & Heck, 2023) | Supervised learning | Transformer | 2023 |
| TST-LLM (Han et al., 2025) | Self-supervised learning | Transformer | 2025 |
| Tabby (Cromp et al., 2026) | Supervised learning | Transformer | 2026 |

blocks the four families share, their differing core assumptions and trade-offs, and concrete method-selection guidance is then provided in Sec. 6.

## 4.1 Tabular Data

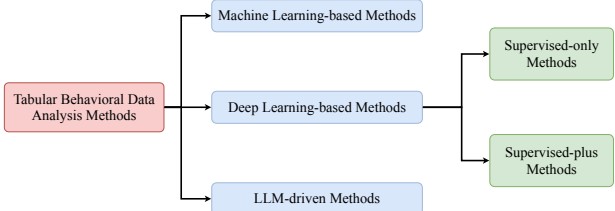

Figure 2: The dissection of the tabular data representation learning methods.

Behavioral data can be instantiated in different forms, one of which is tabular data organized in rows and columns, where each row corresponds to an individual instance and each column represents a specific feature or attribute (Shwartz-Ziv & Armon, 2022; Somvanshi et al., 2024; Ucar et al., 2021; Zhu et al., 2021). In representation learning, such data are typically modeled as static feature vectors, with each instance encoded as a fixed-dimensional vector that aggregates all attribute values. As one of the most prevalent and widely used data types in real-world information systems, tabular data are central to a broad range of domains, including medicine (Johnson et al., 2016; Ulmer et al., 2020), finance (Kumar et al., 2016; Clements et al., 2020), e-commerce (Cheng et al., 2016; Guo et al., 2017). Compared to unstructured data such as images or text, tabular data offer clear and interpretable structure, facilitating efficient storage, processing, and analysis. For example, in digital platforms, tabular data are widely used to record user behaviors such as login frequency, browsing histories, purchase transactions, or content interactions; in online education, they capture attendance records, assignment submissions, and test outcomes. With the growing emphasis on data-driven decision-making across industries, tabular behavioral data play an increasingly critical role in tasks such as user profiling, anomaly detection, risk assessment, and personalized recommendation. However, despite its seemingly regular structure, behavioral tabular data often exhibit high heterogeneity, sparsity, and domain-specific characteristics, posing significant challenges for effective modeling and analysis. Therefore, a systematic investigation into the representation, modeling strategies, and practical applications of tabular behavioral data is of both theoretical importance and practical relevance for advancing intelligent decision systems and enabling digital transformation across sectors.

From a behavioral representation perspective, tabular modeling is best understood as *aggregation-as-abstraction*. Raw behavioral histories are compressed into stable attributes, summary statistics, or engineered fields, and the learning problem becomes how to represent cross-feature interactions among these attributes (Shwartz-Ziv & Armon, 2022; Somvanshi et al., 2024). This abstraction is well suited to behaviors that are largely explained by accumulated tendencies, demographic context, historical frequency, or stable risk factors (Johnson et al., 2016; Cheng et al., 2016; Guo et al., 2017), and becomes limiting when order, interaction timing, or relational context constitutes the behavioral signal itself—precisely the regime that motivates the sequence and dynamic-graph representations that preserve event order, timestamps, and evolving neighborhoods (Kang & McAuley, 2018; Sun et al., 2019b; Li et al., 2020a; Rossi et al., 2020; Xu et al., 2020). Therefore, tabular representation learning contributes to behavioral analysis by improving how heterogeneous attributes are encoded, combined, transferred, and interpreted (Yoon et al., 2020; Bahri et al., 2022; Kim et al., 2024), while accepting a deliberate loss of fine-grained temporal and relational information (Kumar et al., 2019; Rossi et al., 2020).

Modeling methods for tabular data have progressed from traditional machine learning to deep learning and, more recently, to tabular foundation models and LLM-driven interfaces. Traditional tree-based models provide strong inductive biases for heterogeneous features and remain difficult baselines to surpass (Chen & Guestrin, 2016; Ke et al., 2017; Prokhorenkova et al., 2018; Grinsztajn et al., 2022). Deep tabular models mainly aim to learn reusable feature interactions, support multi-modal integration, or exploit unlabeled data (Cheng et al., 2016; Guo et al., 2017; Yoon et al., 2020; Bahri et al., 2022). Foundation-style models such as TabPFN (Hollmann et al., 2025) and TARTE (Kim et al., 2025) push this direction further by attempting to transfer tabular knowledge across datasets, and KumoRFM (Fey et al., 2025) extends in-context foundation modeling to *linked multi-table* behavioral records. LLM-driven methods add a different

capability: they can verbalize, augment, or reason over structured records, but they must carefully preserve numerical and relational structure during serialization (Hegselmann et al., 2023; Sui et al., 2024a).

In this section, we organize tabular behavioral representation methods by the type of representation capacity they add to aggregated profiles: **(1) Machine Learning-based Methods** provide strong partition- or rule-based baselines for heterogeneous attributes (Chen & Guestrin, 2016); **(2) Deep Learning-based Methods** learn trainable feature-interaction embeddings and pretraining objectives (Guo et al., 2017; Yoon et al., 2020); and **(3) LLM-driven Methods** introduce semantic interfaces for structured records (Hegselmann et al., 2023). For each category, we highlight the core modeling techniques, representative architectures, and recent advancements. All methods are concluded in the Tab. 1.

### 4.1.1 Machine Learning-based Methods

For aggregated tabular behavioral data, classical machine learning models are better viewed as task-specific mechanisms for modeling aggregated behavioral profiles rather than merely as competing predictors. While classical non-tree methods such as Support Vector Machines (SVMs) and linear models remain useful in specific niches—particularly small-sample regimes or high-dimensional sparse data—their sensitivity to pre-processing and scalability constraints have limited their broader adoption. Tree-based ensembles therefore became dominant because their inductive bias matches the structure of many tabular behavioral problems. Rather than learning explicit transferable embeddings, Gradient Boosting Decision Trees (GBDT) represent tabular profiles by recursively partitioning the feature space into regions defined by threshold-like and conditional rules (Friedman, 2001). Their optimized implementations, including XGBoost (Chen & Guestrin, 2016), LightGBM (Ke et al., 2017), and CatBoost (Prokhorenkova et al., 2018), inherit this partition-based modeling principle while improving scalability, efficiency, and handling of practical tabular data. This partition-based view is particularly suitable for behavioral profiles, where predictive signals are often sparse, heterogeneous, and conditional: for example, a risk score may change sharply only when several behavioral indicators co-occur, while an e-commerce profile may be determined by a small subset of high-signal attributes rather than by all recorded features (Somvanshi et al., 2024).

Beyond their empirical dominance, recent studies have investigated why such inductive biases make tree ensembles highly competitive on structured data (Shwartz-Ziv & Armon, 2022; Grinsztajn et al., 2022). These models are favored not only for accuracy and training efficiency, but also because they perform implicit feature selection and are robust to missing values, outliers, heterogeneous feature types, and limited-data regimes (Grinsztajn et al., 2022; Larionov et al., 2025). Even with the rapid development of deep tabular learning, GBDT-based models continue to serve as difficult state-of-the-art baselines for general tabular tasks (Borisov et al., 2023; Gorishniy et al., 2021).

Their limitation, however, is representational. The learned structure is usually tied to a fixed set of aggregated input features and a specific prediction objective, rather than being an explicit reusable behavioral embedding (Shwartz-Ziv & Armon, 2022; Grinsztajn et al., 2022). This makes tree ensembles highly effective when behavioral signals have already been summarized into fixed tabular profiles, but less suitable when the representation itself needs to preserve information beyond aggregated attributes, such as temporal order, evolving relational context, or transferable semantics across different behavioral tasks (Borisov et al., 2023; Gorishniy et al., 2021; Somvanshi et al., 2024). Recent interpretability-oriented studies further support this view by showing that the strength of tree-based models often lies in their internal decision partitions and feature-wise decision mechanisms, rather than in learning general-purpose embeddings (McCarter, 2025). This distinction clarifies their role in behavioral representation learning: GBDT-based models remain powerful baselines for aggregated tabular profiles, while sequential, relational, and cross-task behavioral representation challenges motivate complementary deep representation models.

**Summary and takeaways.** The machine-learning stage of tabular representation learning is not merely historical background; it explains why tree-based models remain the reference point for much of the later literature (Chen & Guestrin, 2016; Ke et al., 2017; Prokhorenkova et al., 2018; Grinsztajn et al., 2022). Classical non-tree methods such as linear models and SVMs still retain value in specific niches, especially when data are small, sparse, or strongly preprocessed, but tree ensembles became dominant because they

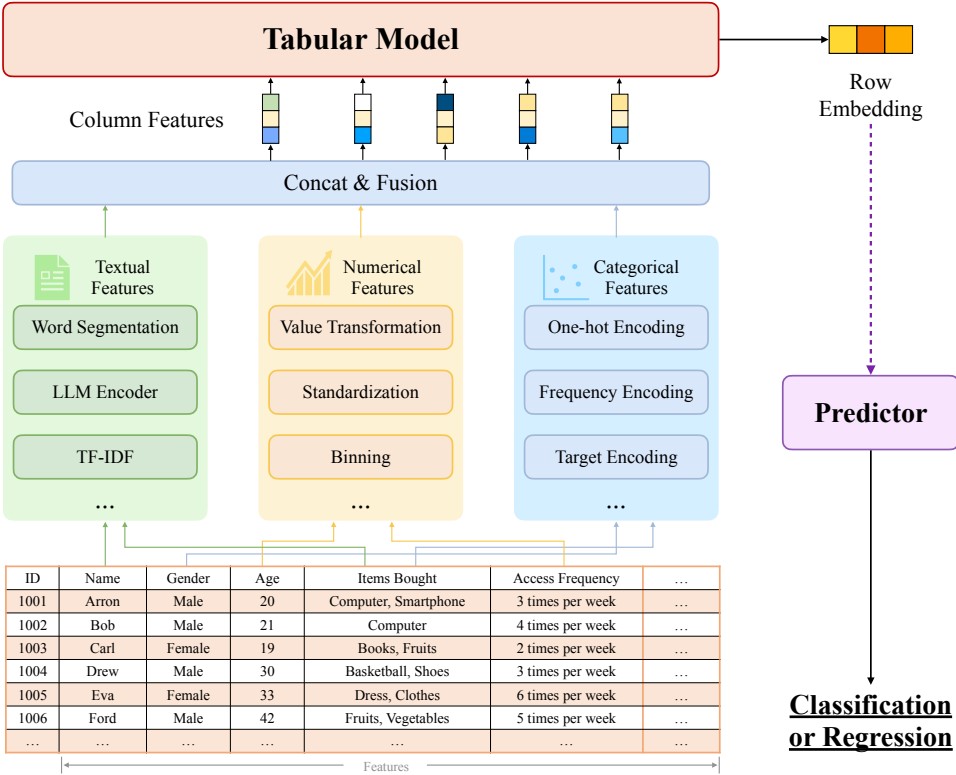

Figure 3: The pipeline of supervised tabular data representation learning methods.

better accommodate heterogeneity, missing values, nonlinear feature interactions, and small-data regimes without heavy feature scaling or architecture tuning.

### 4.1.2 Deep Learning-based Methods

While traditional machine learning methods continue to demonstrate strong performance on structured data, deep learning models are increasingly gaining traction due to their powerful representation learning capabilities. Motivated by their remarkable success in domains such as CV and NLP, deep learning approaches have been actively explored for tabular data modeling and analysis, as neural encoders can model feature interactions, support multi-modal fusion, use self-supervised objectives, and potentially transfer across related behavioral tables (Cheng et al., 2016; Guo et al., 2017; Yoon et al., 2020; Kim et al., 2024). In this section, we provide a comprehensive overview of recent advances in applying deep learning to tabular data, with a particular focus on two key methodological paradigms: **(a) Supervised-only Methods** and **(b) Supervised-plus Methods**.

**Supervised-only Methods** Supervised learning has long been the dominant paradigm in modeling behavioral tabular data. By leveraging manually annotated labels, supervised approaches are capable of learning explicit mappings between input features and target variables, and have been widely applied in tasks such as user behavior prediction, risk assessment, and recommendation systems. We show the pipeline of supervision methods in Fig. 3.

The main role of supervised deep tabular models is to turn a fixed behavioral profile into a trainable representation of feature interactions. Early hybrid architectures address this goal by combining memorization of frequent feature crosses with neural generalization. Wide & Deep (Cheng et al., 2016) keeps a linear component for reliable memorization while using a deep component to generalize beyond observed combinations. DeepFM (Guo et al., 2017) and xDeepFM (Lian et al., 2018) extend this idea by making feature interac-

tions a central modeling target, using Compressed Interaction Network (CIN)-based modules for low-order interactions and Multilayer Perceptron-(MLP) based modules for implicit or explicit high-order interactions. These interaction-oriented designs are useful when the predictive signal lies in combinations of attributes, such as the joint effect of user segment, device, time-of-day statistics, and past transaction frequency. However, their learned representations are usually optimized for a specific supervised target, and may therefore encode task-specific feature correlations rather than general behavioral structure.

A second line of work injects stronger structural priors into neural tabular encoders. Instead of treating all columns as independent raw inputs to an MLP, these methods first impose an alternative organization over the table. TabNN (Ke et al., 2018) uses GBDT-discovered feature groups as neural inputs, SuperTML (Sun et al., 2019a) converts tabular records into image-like layouts for CNN processing, and Boost-GNN (Ivanov & Prokhorenkova, 2021) combines GBDT-based feature modeling with graph neural networks for graph-attributed tabular data. This changes the role of the tabular encoder for behavioral profiles from simply fitting fixed columns to encoding assumptions about how behavioral attributes are organized: some attributes may jointly describe the same user tendency, transaction context, or entity relation (Somvanshi et al., 2024). However, the trade-off is that the induced representation depends on whether the constructed group, layout, or graph reflects meaningful behavioral structure; otherwise, the model may encode artifacts introduced by preprocessing rather than robust behavioral patterns.

Following the phenomenal success of Transformer (Vaswani et al., 2017) and BERT (Devlin et al., 2019) in NLP and representation learning, Transformer-based tabular models reinterpret columns or samples as tokens and use attention to learn context-dependent feature interactions. NPT (Kossen et al., 2021) jointly models the entire dataset using self-attention to capture inter-sample dependencies and enable cross-sample reasoning. FT-Transformer (Gorishniy et al., 2021) emphasizes the importance of standardized evaluation and proposes a simplified yet effective Transformer variant, highlighting the role of architecture and training strategy. LF-Transformer (Na et al., 2024) incorporates ideas from matrix factorization by introducing a latent attention factor matrix to enhance feature representation. Collectively, these methods demonstrate the growing potential of Transformer-based models as powerful alternatives to traditional approaches in tabular learning. Interpretable Additive Tabular Transformer Networks (IATTN) (Thielmann et al., 2024) combine self-attention with an additive modeling structure, allowing Transformer-based models to provide feature-wise contribution explanations for tabular predictions. In aggregated behavioral tables, this attention-based view is useful since the meaning of one attribute often depends on the surrounding profile: for instance, the same transaction frequency may imply different behavioral tendencies under different user segments, temporal summaries, or contextual attributes (Gorishniy et al., 2021). However, attention over columns does not automatically solve the core difficulties of tabular learning. When datasets are small, sparse, heterogeneous, or dominated by categorical irregularities, the flexibility of attention may be less effective than the stronger inductive biases of tree ensembles or carefully regularized tabular architectures (Grinsztajn et al., 2022; Shwartz-Ziv & Armon, 2022).

Beyond feature-cross and attention models, supervised tabular architectures can be organized by the inductive bias they add to fixed behavioral profiles. One group improves optimization and regularization so that neural encoders remain usable on small, sparse, or heterogeneous tables. Regularization Learning Networks (RLN) (Shavitt & Segal, 2018) assigns weight-specific regularization, Regularized DNNs (Kadra et al., 2021) search over regularization strategies, and NODE (Popov et al., 2020) makes decision-tree routing differentiable through $\alpha$-entmax transformation. A second group makes field interaction and feature selection more explicit. NON (Luo et al., 2020) separates intra-field and inter-field modeling, DNF-Net (Abutbul et al., 2020) learns disjunctive-normal-form style feature regions, ARM-Net (Cai et al., 2021) uses gated attention for high-order interactions, and GANDALF (Joseph & Raj, 2022) introduces gated feature learning units. These inductive biases are most helpful when the signal is sparse and conditional rather than attached to a single attribute, such as a risk pattern emerging only from the combination of transaction frequency, user segment, and contextual fields.

Another group changes the representation space to inject structure or interpretability. IGTD (Zhu et al., 2021) maps features into image-like layouts for CNN processing, DNN2LR (Liu et al., 2020) distills nonlinear neural interactions into explicit logistic-regression crosses, TabCBM (Zarlenga et al., 2023) exposes concept-level intermediate representations, and DeepGBM (Ke et al., 2019) combines neural modeling with GBDT-

style handling of sparse categorical and dense numerical features. This line addresses an important tension in behavioral analysis: aggregated behavioral tables often require more expressive representations than hand-crafted features provide, while downstream use cases such as risk assessment, healthcare, and monitoring still demand feature-level, rule-level, or concept-level explanations. However, its limitation is that the imposed structure must correspond to genuine behavioral organization; otherwise image layouts, logical forms, or concept layers may improve benchmark accuracy without preserving the behavioral mechanism they are intended to expose.

Recent supervised architectures further explore how adaptive representations can be learned from fixed feature profiles, i.e., tabular data. Trompt (Chen et al., 2023a) introduces prompt-like mechanisms to produce sample-specific feature importance, while GCondNet (Margeloiu et al., 2024) targets small high-dimensional tables through a tailored conditioning mechanism. MambaTab (Ahamed & Cheng, 2024) adapts structured state-space modeling, following the Mamba architecture, to supervised tabular learning, and BiSHop (Xu et al., 2024) processes columns and rows through linked directional modules to address sparsity and rotation sensitivity. TabR (Gorishniy et al., 2024) augments a Feed-forward Neural Network (FNN) with a customized k-Nearest Neighbors (k-NN) retrieval module, allowing a query profile to be represented together with similar training examples. DP-2Stage (Afonja et al., 2025) moves beyond supervised representation learning toward privacy-aware data generation, using an LLM-based two-stage pipeline to release synthetic tabular data under differential privacy.

Collectively, these models show that deep tabular learning is moving from generic MLP-style encoders toward sample-adaptive weighting, structured state modeling, retrieval-augmented prediction, and privacy-preserving data construction. They are appropriate when behavior is available only as a static table yet the analyst still requires adaptive feature selection, retrieval-based context, compact representations, efficient deployment, or privacy-preserving data sharing. Their shared limitation is representational: once behavioral histories are aggregated into fixed attributes, supervised tabular encoders cannot recover event order or evolving interactions, which motivates the sequential and temporal-graph models that encode behavioral trajectories directly (Kang & McAuley, 2018; Rossi et al., 2020).

**Supervised-plus Methods**  Supervised-plus methods integrate self-supervised pre-training with supervised fine-tuning. Supervised-plus learning addresses a different problem from supervised-only modeling for behavioral tabular data. Instead of only fitting one target, it uses unlabeled tables to learn reusable regularities among attributes: which values co-occur, which fields are semantically related, which perturbations preserve the same profile, and which compact representation can be reused under label scarcity or dataset shift (Yoon et al., 2020; Bahri et al., 2022; Kim et al., 2024; Hollmann et al., 2025). This paradigm is especially useful when downstream labels are expensive, delayed, or noisy, while unlabeled behavioral records are abundant.

Researchers have explored this idea through several complementary pretext families. Structure-aware pre-training treats a table not merely as independent columns, but as a relational object whose fields and cells form meaningful dependencies. CARTE (Kim et al., 2024) represents each row as a star-like graph and learns transferable row representations from large background data; HYTREL (Chen et al., 2023b) models cells, rows, columns, and tables through a hypergraph structure. The pre-training objective thereby shifts from reconstructing isolated cells to learning how attributes *jointly* describe an entity, action, or context. Its success hinges on whether the constructed cell-row-column structure reflects genuine behavioral relations rather than incidental table layout.

Another widely adopted strategy for self-supervised learning on tabular data is based on masking and reconstruction, inspired by masked language modeling in NLP. Masking, corruption, and reconstruction objectives learn representations by asking the model to infer missing or perturbed attributes from the remaining context. VIME (Yoon et al., 2020) estimates both feature values and mask indicators; Tabbie (Iida et al., 2021) detects swapped cells; SCARF (Bahri et al., 2022) creates corrupted views through marginal feature replacement. These objectives encourage embeddings to capture dependencies such as which actions, attributes, or outcomes tend to appear together. However, the key limitation is augmentation validity: replacing a feature with a marginally sampled value may create profiles that are statistically useful for contrastive learning but implausible under domain constraints.

Transformer-style supervised-plus methods mainly differ in how they create self-supervision from incomplete or perturbed tables. SAINT (Somepalli et al., 2021) combines contrastive views with denoising, TabNet (Arik & Pfister, 2021) masks features for reconstruction, and TabTransformer (Huang et al., 2020a) uses replacement-token detection over categorical embeddings. Recontab (Chen et al., 2023c) and XTab (Zhu et al., 2023a) further explore reconstruction-oriented or transferable tabular pre-training objectives, with ReConTab (Chen et al., 2023c) adding semi-supervised classification and contrastive losses, while SwitchTab (Wu et al., 2024a) decomposes each profile into shared and sample-specific factors by swapping mutual-information components before reconstruction. For behavioral representation learning, the advantage is label efficiency: unlabeled user, patient, transaction, or item profiles can teach the model which attributes cohere before task labels are available. The main risk is again behavioral plausibility, because masking, replacement, swapping, or denoising objectives are helpful only when the corrupted profile remains a meaningful approximation to possible behavior rather than an impossible combination of domain-constrained fields.

The same supervised-plus family also includes modality transformation and generative modeling. PTab (Liu et al., 2022b) converts tables into text and applies BERT-style pre-training, while TabDDPM (Kotelnikov et al., 2023) uses diffusion modeling to learn the distribution of heterogeneous tabular records. TP-BERTa (Yan et al., 2024) emphasizes numerical tokenization and feature semantics. Foundation-style tabular models push this trend further: TARTE (Kim et al., 2025) advocates large-scale tabular pre-training for transferable representations, while TabPFN (Hollmann et al., 2025) shows how a model trained on a broad prior over tabular tasks can support rapid adaptation through in-context prediction. KumoRFM (Fey et al., 2025) extends this foundation-model direction to linked relational tables, using in-context learning over multi-table relational graphs, which is especially relevant for behavioral records distributed across users, items, transactions, sessions, and contextual entities. These models are most relevant to behavioral analysis when the goal is not only to fit one table, but to transfer useful attribute or relational representations across related behavioral datasets or rapidly adapt when labels are limited. The open challenge is that transfer across tables can fail when column semantics, sampling mechanisms, link structures, or behavioral populations differ substantially.

Finally, compactness and transferability have become explicit design goals. TDColER (Kang et al., 2025) studies tabular representation distillation, asking how complex table structure can be transformed into compact representations for efficient downstream learning.

Together, graph-structured pre-training, feature masking, corruption, reconstruction, diffusion, text conversion, foundation-style priors, relational in-context learning, and distillation show that tabular supervised-plus learning is not simply a way to improve prediction scores. Its impact on behavioral representation learning is to make aggregated profiles and linked records more reusable under label scarcity, dataset shift, heterogeneous attribute semantics, and data generation or sharing scenarios (Kim et al., 2024; Yoon et al., 2020; Bahri et al., 2022; Kotelnikov et al., 2023; Kim et al., 2025; Fey et al., 2025; Kang et al., 2025).

**Summary and takeaways.** A consistent pattern in tabular representation learning is that strong empirical performance does not come from architectural complexity alone. Instead, successful approaches usually make one of three contributions: they model feature interactions more explicitly (Cheng et al., 2016; Guo et al., 2017; Lian et al., 2018), they exploit unlabeled data through pre-training (Yoon et al., 2020; Bahri et al., 2022; Kim et al., 2024), or they combine neural models with stronger tabular inductive biases such as tree ensembles or retrieval (Ke et al., 2019; Gorishniy et al., 2024). This also explains why deep tabular models do not uniformly displace GBDT-style methods: on small or heterogeneous datasets, optimization difficulty and weak inductive bias often outweigh representational flexibility (Grinsztajn et al., 2022; Gorishniy et al., 2021). In practice, deep tabular models appear most compelling when the task benefits from transfer, multi-modal fusion, or scalable pretraining, whereas tree-based methods remain difficult to beat in low-resource, high-heterogeneity settings (Grinsztajn et al., 2022; Shwartz-Ziv & Armon, 2022).

### 4.1.3 LLM-driven Methods

Recently, LLMs have attracted significant attention for their impressive performance across a wide range of tasks, demonstrating generalization capabilities far beyond traditional NLP applications (Fu et al., 2022;

Wei et al., 2022a). With their powerful language understanding and generation abilities, LLMs are increasingly regarded as a promising pathway toward Artificial General Intelligence (AGI) (Chang et al., 2024; Zhao et al., 2023a). Naturally, efforts have begun to investigate the potential of LLMs in tabular data processing, leveraging their language interfaces to unify the modeling of structured information. These approaches have shown promising results in tasks such as prediction, question answering, reasoning, and data augmentation (Hegselmann et al., 2023; Sui et al., 2024a; Cromp et al., 2026). What an LLM adds here is semantic, therefore, the appeal is that column names, metadata, task descriptions, retrieved evidence, and natural-language explanations can help represent not only the values in a row, but also what a behavioral profile means in context. This also enables fairness-aware in-context adaptation, where tabular foundation models are adjusted to reduce biased inference behavior (Kenfack et al., 2026). The central trade-off is representation mismatch: LLMs bring semantic priors and flexible reasoning, but behavioral tables often depend on exact numerical values, row-column structure, and constraints inherited from the underlying temporal or relational process, all of which can be weakened when tables are serialized into text (Sui et al., 2024a; Singha et al., 2023; Narayan et al., 2022; Fang et al., 2024).

LLM-driven techniques for tabular data modeling can therefore be understood as attempts to make structured behavioral records accessible to language-based representations without destroying their table semantics. The first design problem is *serialization*: tables may be represented as Pandas-like objects, JSON lines, nested lists, HTML, delimiter-separated rows, or template-based natural-language statements that expose column headers and cell values (Singha et al., 2023; Narayan et al., 2022; Yu et al., 2023a). These choices are not superficial formatting decisions. For behavioral data, serialization determines whether the LLM sees a profile as an unordered bag of fields, a set of typed attributes, or a context-rich description of a user, transaction, or event. Robustness analyses of tabular question answering further show that model behavior depends on how attention aligns questions with tabular evidence (Bhandari et al., 2025). Thus, LLM-based tabular representation is useful when semantic field names and textual context matter, but risky when numerical precision or row-column structure is lost during conversion.

The second design problem is *context compression*. Because behavioral tables can contain many rows, columns, and historical records, LLMs often need to compress or retrieve the relevant portion rather than consume the full table. TaPEx (Liu et al., 2022c) and TAPAS-style (Herzig et al., 2020) models illustrate the need to control table length and input structure, while GPT4Table (Sui et al., 2024a) imposes constraints to keep table inputs within usable context limits. Selective retrieval and Retrieval-Augmented Generation (RAG)-style pipelines further select relevant tables, rows, cells, or evidence before generation (Sundar & Heck, 2023; Gao et al., 2023b). For behavioral representation learning, this plays a role similar to task-dependent evidence selection: the model can focus on the fields most relevant to the current behavioral question. The trade-off is that retrieval mistakes become representation mistakes, especially when rare but decision-critical behavioral signals are filtered out.

The third design problem is *semantic enrichment*. Metadata such as table size, units, column types, statistical summaries, document references, and terminology can help the LLM interpret structured records, as shown by TAP4LLM (Sui et al., 2024b) and related table-QA studies (Aly et al., 2021). Prompt engineering and explicit task descriptions further guide how the serialized table should be read (Ggaliwango et al., 2024; Sui et al., 2024a). Beyond inference-time prompting, TST-LLM (Han et al., 2025) uses LLMs to generate task-aware pretext objectives for self-supervised tabular representation learning. In behavioral settings, these strategies help translate opaque fields into analyzable signals, such as converting transaction attributes into risk-relevant evidence. However, enrichment is beneficial only when it adds grounded semantics; irrelevant statistics, excessive header hierarchy, or unsupported context can increase prompt complexity without improving the behavioral representation (Sui et al., 2024b).

In contrast to prompt-based approaches, fine-tuning adapts LLMs to specific tabular tasks by updating model parameters with labeled data, often resulting in improved performance and tighter integration with external tools such as SQL and Python. Recent studies have actively explored how LLMs can interact with these tools to enhance reasoning over structured data. For instance, an iterative framework is proposed in (Zhang et al., 2023a), where execution errors from SQL queries are fed back into the LLM to enable successive refinements that significantly improve query generation accuracy. In the realm of analytical platforms, the

work of (Liu et al., 2023c) introduces a no-code solution that leverages LLMs to generate data summaries, formulate relevant analytical questions, and produce structured queries.

Furthermore, a survey by (Zhang et al., 2024f) reviews natural language interfaces for tabular data, highlighting recent advances in Text-to-SQL and Text-to-Vis techniques. Together, these fine-tuning and tool-integrated approaches showcase the growing potential of LLMs in enabling more intuitive and effective interaction with tabular data. A recent survey (Fang et al., 2024) also reviews how LLMs are adapted for tabular prediction, data generation, and reasoning, highlighting emerging trends and open challenges.

Finally, LLM-driven tabular systems increasingly combine reasoning, fine-tuning, and external tools. Chain-of-Thought (CoT) prompting (Wei et al., 2022b), program-aided reasoning (Chen et al., 2023d), and self-consistency (Wang et al., 2023a) decompose table reasoning into intermediate steps, while fine-tuning adapts LLMs to task-specific table formats. Tool-integrated systems use SQL, Python, execution feedback, or no-code analytics interfaces to query and transform structured data before answering (Zhang et al., 2023a; Liu et al., 2023c; Zhang et al., 2024f). For behavioral analysis, this makes LLMs valuable not only as encoders, but also as interfaces for inspecting, querying, and explaining structured behavioral evidence. Their limitation is that such systems may produce convincing narratives even when the underlying representation has lost exact numerical, temporal, or relational constraints—a discrepancy between fluent explanation and underlying structure that recent LLM-for-tabular surveys also emphasize (Fang et al., 2024).

**Summary and takeaways.** The main lesson from LLM-driven tabular modeling is that success depends less on simply applying an LLM and more on controlling how structured information is presented to it (Sui et al., 2024a;b; Fang et al., 2024). Methods tend to improve when they reduce irrelevant context, preserve semantic cues such as field types and units, and complement the model with retrieval or external tools (Sundar & Heck, 2023; Gao et al., 2023b; Zhang et al., 2023a; Liu et al., 2023c). At the same time, the literature suggests that LLMs are currently strongest when used for table reasoning, augmentation, or tool-mediated interaction, rather than as direct replacements for specialized predictive tabular models (Hegselmann et al., 2023; Sui et al., 2024a; Fang et al., 2024). The central bottleneck remains representation mismatch: numerical precision, token efficiency, and table structure are only imperfectly preserved after serialization (Singha et al., 2023; Narayan et al., 2022; Yu et al., 2023a; Fang et al., 2024).

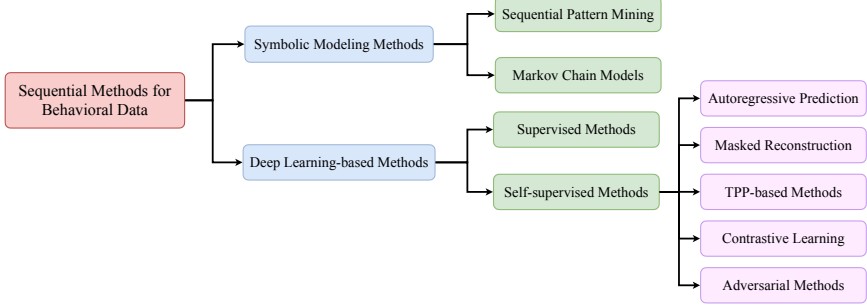

Figure 4: The dissection of the event sequence representation learning methods.

## 4.2 Event Sequences

Rooted in the foundational principles of sequential decision-making (Sutton & Barto, 1998), human behavior can be naturally modeled as a sequential process in which current actions depend on previous actions, elapsed time, and contextual state. Unlike traditional feature-based analysis, sequence-based structural modeling captures temporal dynamics, enabling more accurate behavior representation by leveraging historical patterns to inform present decision-making. Typically, a behavior sequence consists of a continuous series of ordered interactions, each associated with a timestamp and relevant contextual features. This formulation allows for the application of powerful sequential representation learning models to behavioral data analysis. Broadly, sequential modeling approaches can be categorized into **(1) Symbolic Modeling Methods**, which represent behavior through interpretable motifs or transition rules (Yap et al., 2012; Wang & Cao,

Table 2: Event sequence representation learning methods.

| Method | Training Paradigm | Model Architecture | Year |
|---|---|---|---|
| Symbolic Modeling Methods | | | |
| FPMC (Rendle et al., 2010) | Self-supervised learning | Markov Chain | 2010 |
| PSPM (Yap et al., 2012) | Self-supervised learning | Sequential pattern mining | 2012 |
| PRME (Feng et al., 2015) | Self-supervised learning | Markov Chain | 2015 |
| Deep Learning-based Methods | | | |
| GRU4Rec (Hidasi et al., 2016) | Supervised learning | GRU | 2016 |
| RMTPP (Du et al., 2016) | Self-supervised learning | RNN | 2016 |
| NHP (Mei & Eisner, 2017) | Self-supervised learning | LSTM | 2017 |
| Event2Vec (Hong et al., 2017) | Supervised learning | Word2Vec | 2017 |
| IRGAN (Wang et al., 2017) | Semi-supervised learning | GAN | 2017 |
| HRNN (Quadrana et al., 2017) | Supervised learning | RNN | 2017 |
| Caser (Tang & Wang, 2018) | Supervised learning | CNN | 2018 |
| AttRec (Zhang et al., 2018) | Supervised learning | Transformer | 2018 |
| MANN (Chen et al., 2018) | Supervised learning | Memory network | 2018 |
| RecGAN (Bharadhwaj et al., 2018) | Semi-supervised learning | GAN+RNN | 2018 |
| RPPN (Xiao et al., 2019) | Self-supervised learning | RNN | 2019 |
| BERT4Rec (Sun et al., 2019b) | Self-supervised learning | BERT | 2019 |
| DTCDR (Zhu et al., 2019a) | Supervised learning | MLP | 2019 |
| FDSA (Zhang et al., 2019b) | Supervised learning | Transformer | 2019 |
| NextItNet (Yuan et al., 2019) | Supervised learning | CNN | 2019 |
| SAHP (Zhang et al., 2020) | Self-supervised learning | Transformer | 2020 |
| THP (Zuo et al., 2020) | Self-supervised learning | Transformer | 2020 |
| BEHRT (Li et al., 2020c) | Self-supervised learning | BERT | 2020 |
| TiSASRec (Li et al., 2020a) | Supervised learning | Transformer | 2020 |
| RAPT (Ren et al., 2021) | Self-supervised learning | Transformer | 2021 |
| CoSeRec (Liu et al., 2021a) | Self-supervised learning | GAN+CL | 2021 |
| ASReP (Liu et al., 2021b) | Supervised learning | Transformer | 2021 |
| UniSRec (Hou et al., 2022) | Self-supervised learning | BERT+Transformer | 2022 |
| RecGURU (Li et al., 2022a) | Self-supervised learning | Transformer | 2022 |
| promptTPP (Xue et al., 2023) | Continual learning | Transformer+ Prompts | 2023 |
| Meta TPP (Bae et al., 2023) | Meta learning | Transformer | 2023 |
| BERT4ETH (Hu et al., 2023b) | Self-supervised learning | BERT | 2023 |
| PrimeNet (Chowdhury et al., 2023) | Self-supervised learning | Transformer | 2023 |
| ECGAN-Rec (Ni et al., 2023) | Semi-supervised learning | GAN | 2023 |
| SeqLink (Abushaqra et al., 2024) | Supervised learning | Neural ODE | 2024 |
| Chronos (Ansari et al., 2024) | Self-supervised learning | Transformer | 2024 |
| TimesFM (Das et al., 2024) | Self-supervised learning | Decoder-only Transformer | 2024 |
| player2vec (Wang et al., 2024d) | Self-supervised learning | BERT | 2024 |
| TOTEM (Talukder et al., 2024) | Self-supervised learning | VQ-VAE | 2024 |
| PhASER (Mohapatra et al., 2025) | Supervised learning | Multiple | 2025 |
| Residual TPP (Yuan & Fang, 2025) | Self-supervised learning | Hawkes + Neural TPP | 2025 |
| IOCLRec (Wang et al., 2025) | Self-supervised learning | Transformer+CL | 2025 |
| HORAE (Hu et al., 2025) | Self-supervised learning | Transformer | 2025 |

2020), and **(2) Deep Learning-based Methods**, which learn dense behavioral states from supervised or self-supervised objectives (Kang & McAuley, 2018; Sun et al., 2019b; Du et al., 2016; Xie et al., 2022).

Event-sequence methods move behavioral representation from aggregated profiles to temporally ordered trajectories. Rather than summarizing a subject by static counts or attributes, they encode *what happened before what*, how recently each event occurred, whether actions appear in bursts or long gaps, and how a subject's state evolves through repeated interactions (Kang & McAuley, 2018; Sun et al., 2019b; Li et al., 2020a). This temporal view is crucial when identical aggregate statistics correspond to different behavioral meanings: repeated purchases after long inactivity, a sudden sequence of abnormal transactions, or progressive clinical deterioration across visits (Li et al., 2020c; Hu et al., 2023b). Sequence models are therefore most suitable when the focal behavioral subject can be described by a meaningful timeline. Their limitation is that the timeline is usually centered on one subject or one interaction stream; it can preserve temporal order but may abstract away the multi-entity dependencies behind each event. When the same action is shaped by interactions among users, items, accounts, or communities, temporal graph representations provide a more suitable way to model both event order and relational context (Rossi et al., 2020; Xu et al., 2020).

### 4.2.1 Symbolic Modeling Methods

Symbolic sequence methods provide an interpretable view of behavioral trajectories before the widespread use of dense neural representations. Instead of learning continuous embeddings, they represent behavior through discrete and human-readable structures, such as frequent motifs or transition probabilities. This makes them useful when analysts need explicit rules, interpretable routines, or lightweight baselines, but it also constrains their ability to encode rare events, long-range dependencies, and heterogeneous action contexts (Yap et al., 2012; Wang & Cao, 2020; Wang et al., 2019; Feng et al., 2015). We discuss two representative forms: **(a) Sequential Pattern Mining**, which represents behavior through repeated subsequences, and **(b) Markov Chain Models**, which represent behavior through local transition dynamics.

**Sequential Pattern Mining** Sequential pattern mining first extracts frequent subsequences and then uses these motifs to represent or predict behavior (Yap et al., 2012; Wang & Cao, 2020). As a representation strategy, it captures repeated action fragments, routine trajectories, or common transition patterns. Its value for behavioral representation learning is interpretability: the mined motifs can be read directly as behavioral rules, which is well suited to routine mobility patterns, repeated purchase paths, or standardized action workflows. The reliance on frequency is also its main weakness, since rare yet decisive behaviors—such as atypical fraud patterns or early-warning clinical trajectories—tend to be underrepresented, and exhaustive mining scales poorly on long or heterogeneous histories, as summarized in the sequential-pattern-mining literature (Wang et al., 2019).

**Markov Chain Models** Markov Chain have been widely employed to model behavior sequence transitions. Existing approaches can be categorized into two main paradigms: (1) direct computation of transition probabilities from historical sequences (Cheng et al., 2013; Zhang et al., 2014), and (2) probability estimation through Euclidean distance between action embeddings (Moore et al., 2013; Feng et al., 2015). This representation is appropriate when short-term habits, recency effects, or local routines dominate the task, such as session-level recommendation or location transition modeling. However, the Markov assumption also defines its boundary: when the next action depends on long histories, delayed effects, or multiple earlier actions jointly, a local transition representation discards information needed for richer behavioral analysis (Feng et al., 2015).

**Summary and takeaways.** Symbolic sequence modeling methods remain useful because they make the assumptions of early behavioral sequence analysis explicit. Sequential pattern mining emphasizes frequent co-occurrence structures, while Markov models focus on local transition dynamics (Yap et al., 2012; Wang & Cao, 2020; Cheng et al., 2013; Zhang et al., 2014). Their practical value lies in interpretability and conceptual clarity. However, these approaches also reveal the limitations that motivated later neural sequence models: frequent-pattern pipelines can be computationally expensive and biased toward common motifs (Wang et al., 2019), whereas Markov assumptions struggle to capture long-range dependencies and complex multi-action interactions (Feng et al., 2015).

### 4.2.2 Deep Learning-based Methods

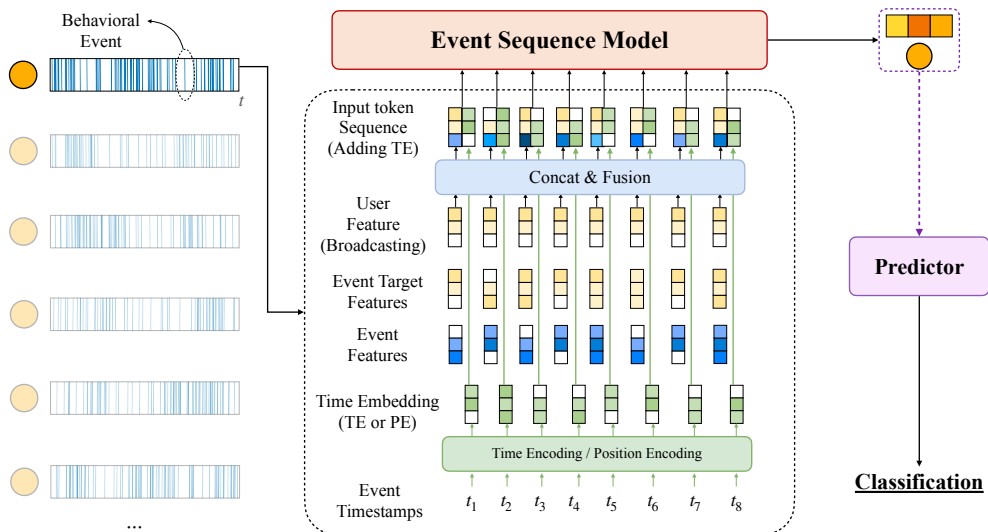

Figure 5: The pipeline of supervised event sequence representation learning methods.

Deep sequence methods move beyond symbolic motifs and local transition tables by learning trainable representations of behavioral state. Their key contribution is not only to predict the next action, but also to decide which parts of the history should dominate the current representation, how temporal order and time intervals should be encoded, and how event semantics should shape the trajectory embedding (Kang & McAuley, 2018; Sun et al., 2019b; Li et al., 2020a; Du et al., 2016). We organize these methods into **(a) Supervised Methods**, where task labels directly shape the sequence representation, and **(b) Self-supervised Methods**, where pretext objectives such as next-event prediction, masking, timing likelihood, contrastive learning, or adversarial discrimination determine what behavioral information is preserved (Sun et al., 2019b; Du et al., 2016; Xie et al., 2022).

**Supervised Methods** Supervised learning methods for behavior sequence representation utilize annotated datasets to optimize task-specific objectives such as classification or regression through direct minimization of supervised loss functions (e.g., cross-entropy or mean squared error). Although dependent on manual annotation, these methods achieve strong empirical performance when sufficient labeled data are available, with architectural flexibility across domains. The pipeline of supervised methods is shown in Fig. 5.

Rather than reviewing supervised sequence models only as a chronology of architectures, we organize them by the type of behavioral state they construct. Recurrent and memory-based models represent behavior as an evolving state; convolutional models represent behavior through local motifs; attention-based models represent behavior through global context comparison; continuous-time and phase-aware models emphasize irregular timing and non-stationarity; and LLM-driven or multi-modal models enrich event sequences with semantic context. This organization highlights the central question of supervised sequence representation learning: which part of the behavioral history should be preserved for the target task, and which temporal or semantic signals should shape the embedding.

Recurrent models encode the past as a recursively updated hidden state $(h_t)$,

$$h_t = f(h_{t-1}, x_t),$$

where each new event $(e_t)$ revises the current representation of the user or entity ($x_t$ is the embedding of the current event $e_t$, and $f(\cdot)$ denotes the state update function that integrates the previous hidden state and the current input). This makes RNNs and their gated variants, such as Long Short-Term Memory (LSTM) (Graves, 2012) and Gated Recurrent Unit (GRU) (Chung et al., 2014), natural choices for modeling short- to medium-range behavioral dependencies through sequential state updates. Memory-Augmented Neural Networks (MANN) (Chen et al., 2018) extend this idea with external memory for collaborative

filtering, while Event2Vec (Hong et al., 2017) adapts the Word2Vec (Church, 2017) embedding paradigm to neuromorphic event streams for supervised classification (Bi et al., 2019). SeqLink (Abushaqra et al., 2024) further models sparse and irregularly observed behavior sequences through continuous-time latent dynamics with Neural Ordinary Differential Equations (ODEs). These recurrent encoders maintain an explicit behavioral state that is refreshed at each event, which suits behaviors governed by order and state persistence and naturally supports online updating. Their recognized limitation is that strictly sequential updates restrict parallelism and tend to over-compress distant but informative events—a bottleneck that subsequently motivated attention-based encoders able to relate all positions in a history directly (Kang & McAuley, 2018).

Convolutional sequence encoders instead represent behavior as a composition of local temporal motifs. Caser (Tang & Wang, 2018) uses CNN filters to capture short user-item interaction patterns, while NextItNet (Yuan et al., 2019) stacks dilated convolutions to expand the receptive field without fully recurrent computation. Convolutional encoders represent a history as a composition of short, translation-invariant motifs, offering efficient and stable extraction of local routines without recurrent computation. Their limitation is restricted temporal reach, since long-range dependencies can be captured only indirectly by enlarging the receptive field, a constraint that motivated the subsequent adoption of self-attention, where distant events are compared directly (Kang & McAuley, 2018).

Transformer-based encoders shift the representation from recursive state updates or local motifs to global context comparison. SASRec (Kang & McAuley, 2018) introduced self-attention to sequential recommendation, enabling each event to attend to other events in the user's history. TiSASRec (Li et al., 2020a) incorporates personalized time intervals into attention, AttRec (Zhang et al., 2018) combines self-attention with metric learning for next-item prediction, and ASReP (Liu et al., 2021b) augments short behavior sequences through reverse training. More recent work further adapts supervised sequence representations to distributional changes: PhASER (Mohapatra et al., 2025) explicitly models non-stationarity through phase-aware representations and phase-driven augmentation. These methods are suitable when long histories, non-local dependencies, interval-aware attention, or non-stationary dynamics are decision-critical, but they also require careful control of sequence length, temporal encoding, and distribution shift.

LLM-driven and multi-modal supervised sequence models add semantic alignment to temporal modeling. AlmostRec (Wu et al., 2025c) aligns ID, text, and image signals with a large multi-modal model while retaining an ID prediction objective for controllable recommendation. This direction is useful when the same action sequence is ambiguous without item descriptions, content, or other semantic cues. Across supervised sequence encoders, the core trade-off is therefore not only architectural but representational: labels shape useful task-specific embeddings, yet strong supervised pressure can make the representation overly specific to one prediction target. Recent work on sequence-specific regularization further shows that properly regularizing such embeddings is important for improving representation quality and transferability (Butera et al., 2025).

**Self-supervised Methods** Self-supervised sequence methods are particularly important for behavioral data since event histories are often abundant while labels are sparse, delayed, or task-specific. Different pretext objectives emphasize different behavioral information: *Autoregressive Prediction* preserves directional evolution (Ansari et al., 2024; Das et al., 2024), *Masked Reconstruction* preserves bidirectional context (Sun et al., 2019b; Li et al., 2020c), *TPP-based Methods* preserve continuous-time event likelihood (Du et al., 2016; Mei & Eisner, 2017; Zuo et al., 2020), *Contrastive Learning* preserves invariances across valid augmentations (Xie et al., 2022; Liu et al., 2021a; Qiu et al., 2022), and *Adversarial Methods* improve robustness against hard or noisy behavioral alternatives (Bharadhwaj et al., 2018; Wang et al., 2017; Ni et al., 2023). These objectives should therefore be understood as choices about what the learned sequence representation is encouraged to preserve.

*i) Autoregressive Prediction:* Autoregressive Prediction is the framework that learns sequence representations by predicting the future values based on observed historical data, thus encouraging the model to learn the underlying temporal structure and behavior patterns. Such models treat the event sequence as a unidirectional process, where the future is generated step-by-step conditioned on the past. This formulation aligns naturally with many real-world tasks such as next-click prediction (Yuan et al., 2019; Cheng et al.,

2013), stock price forecasting (Hu et al., 2023b), and sequential decision-making (Wang et al., 2024d). For instance, inspired by the parallels between generative modeling and financial tasks such as churn prediction, credit default, and expenditure forecasting, NPPR (Skalski et al., 2023) employs next event prediction adapted from language modeling to handle multivariate transaction events. Chronos (Ansari et al., 2024) further formulates time-series modeling as language-model-like next-token prediction by discretizing continuous values into tokens, enabling transferable temporal representations across domains. Recent time-series foundation models extend the same logic to broader temporal data. TimesFM (Das et al., 2024) trains a decoder-only foundation model for forecasting across many time-series domains, illustrating how autoregressive pre-training can provide reusable temporal representations even when the downstream behavioral task differs from the pre-training distribution. For behavioral analysis, these models are most relevant when the key signal lies in temporal evolution and when a general temporal prior can reduce the need for task-specific sequence architectures. Their limitation is that generic time-series pre-training may not encode discrete action semantics, relational context, or domain constraints unless these are represented in the input.

*ii) Masked Reconstruction:* Masked Reconstruction learns sequence representations by reconstructing masked portions of historical records (e.g., actions or features). During training, the model processes partially masked input sequences and learns to predict the original data using the remaining unmasked tokens as context. This reconstruction objective forces the model to capture both temporal patterns and structural dependencies in the data, which enhances its effectiveness for downstream classification tasks.

This paradigm closely resembles the successful BERT model in NLP. A substantial body of research directly adapts BERT's encoder architecture and Masked Language Modeling (MLM) objective for behavior sequences, repurposing its bidirectional attention mechanism to capture complex temporal dependencies in actions and features. For example, BERT4Rec (Sun et al., 2019b) leverages a bidirectional Transformer encoder for masked item prediction, enabling robust information aggregation across entire historical sequences. Similarly, BEHRT (Li et al., 2020c) builds upon BERT's framework, adopting a two-stage pre-training and fine-tuning approach while integrating multi-modal data from five heterogeneous data sources to enhance depression prediction accuracy. Furthermore, BERT4ETH (Hu et al., 2023b) improves Ethereum transaction pattern recognition using a Transformer architecture with masked address modeling and domain-specific strategies to address data repetitiveness, skewness, and heterogeneity. Meanwhile, player2vec (Wang et al., 2024d) adapts long-context Transformer architectures to model player behavior from sessionized interaction logs, producing enriched behavior representations. These BERT-like methods share the same representation principle: behavior is treated as a partially observed trajectory, and the model learns useful embeddings by recovering missing actions, attributes, or entities from bidirectional context. This is particularly suitable when the meaning of an event depends on both preceding and surrounding behavioral context, such as medical visits, transaction histories, or game sessions (Sun et al., 2019b; Li et al., 2020c; Hu et al., 2023b; Wang et al., 2024d).

Beyond BERT-like methods, some works adopt reconstruction-based objectives without explicit masking to learn general-purpose behavioral representations. For instance, TOTEM (Talukder et al., 2024) learns discrete time series representations by training a Vector-Quantized Autoencoding Reconstruction (VQ-VAE) tokenizer to reconstruct unmasked historical sequences in a self-supervised manner, producing a fixed codebook of temporal tokens shared across domains and tasks.

Conversely, an alternative research direction argues for architectural modifications to address inherent limitations of vanilla Transformers in modeling behavioral data. Innovations such as RAPT (Ren et al., 2021) augment self-attention with explicit temporal encoding to handle irregular inter-event intervals, while SAGE (Wang et al., 2023b) introduces noisy reconstruction, injecting deletions, substitutions, and insertions into the masked sequence, leading to notable performance improvements.

Taken together, masked reconstruction methods learn behavioral representations by testing what can be recovered from context. BERT-like models emphasize bidirectional contextual coherence (Sun et al., 2019b; Li et al., 2020c; Hu et al., 2023b; Wang et al., 2024d), reconstruction-based tokenizers such as TOTEM emphasize reusable temporal discretization (Talukder et al., 2024), and specialized variants such as RAPT and SAGE adapt the objective to irregular intervals and noisy behavioral histories (Ren et al., 2021; Wang et al., 2023b). The strength of this paradigm lies in exploiting unlabeled histories to learn transferable

sequence embeddings; its boundary is a recoverability bias, where the representation may favor common or easily predicted events while rare but behaviorally decisive actions may still require task-specific supervision, contrastive objectives, or explicit temporal modeling (Xie et al., 2022; Du et al., 2016).

*iii) TPP-based Methods:* Temporal Point Processes (TPPs) are a class of probabilistic models designed to characterize the timing of events over continuous time by explicitly preserving the original timestamp of each occurrence. Unlike traditional sequence models which predict the next token in a discrete sequence, TPPs model the conditional intensity function to estimate the probability of an event occurring at any time $t$, given its event history. This unique formulation makes TPPs particularly effective for handling irregular, asynchronous event sequences as commonly found in behavioral data streams such as click logs, social media actions, or medical visit records.

While traditionally formulated in the context of generative modeling, TPPs can also be viewed as a specialized form of self-supervised learning. They optimize a likelihood-based objective by predicting the timing of future events solely based on past observations, without requiring any external labels.

Early deep learning approaches for TPPs naturally leveraged RNNs (Cho et al., 2014) to encode the event history. For instance, RMTPP (Du et al., 2016) uses an RNN to learn a representation of the event history, embedding the sequence of past event times and their associated markers into a hidden state vector which then parameterizes the intensity function. Similarly, NHP (Mei & Eisner, 2017) introduces a novel continuous-time LSTM architecture where the hidden state evolves continuously between events, allowing it to capture more complex, non-linear patterns of excitation and inhibition than traditional Hawkes models. RPPN (Xiao et al., 2019) jointly model asynchronous event sequences and synchronous time series by parameterizing the endogenous and exogenous intensity functions with separate RNNs.

These RNN-based TPPs convert irregular event histories into continuous-time state representations, allowing the intensity function to depend on both past event types and elapsed time. This formulation captures behavioral rhythms such as excitation, inhibition, delayed response, and interaction between asynchronous events and synchronous signals. However, because the historical representation is still updated sequentially, long-range dependencies and large-scale parallel training remain difficult (Du et al., 2016; Mei & Eisner, 2017; Xiao et al., 2019).

Because RNN-based TPPs struggle to capture long-term dependencies and are difficult to parallelize, subsequent research drew inspiration from the success of attention mechanisms in NLP. For example, SAHP (Zhang et al., 2020) and THP (Zuo et al., 2020) employ self-attention to summarize the influence of history events and compute the probability of the next event, which not only models long-range dependencies more effectively but also allows for parallel computation over the event sequence, breaking the sequential processing bottleneck of RNNs (Cho et al., 2014). There has also been research in creative ways of training temporal point processes, such as in continual learning framework (Xue et al., 2023) whose basis is a continuous-time retrieval prompt pool for modeling streaming event sequences and meta learning framework (Bae et al., 2023) to quickly adapt to new, heterogeneous sequences from few examples.

While attention-based TPPs improve expressiveness, they often incur high computational overhead and face difficulty in training on long sequences. To address this, the residual TPP (Yuan & Fang, 2025) proposes a hybrid framework that first employs a lightweight Hawkes process to capture dominant temporal patterns, and then delegates only the residual, irregular dynamics to a neural TPP, significantly reducing training complexity while preserving accuracy.

TPP-based methods emphasize that behavioral representation is not only about *what* happened, but also about *when* it happened (Du et al., 2016; Mei & Eisner, 2017). By modeling event likelihood over continuous time, they are particularly suitable for irregular and asynchronous behavioral streams where waiting times, bursts, delays, and repeated occurrences carry important meaning (Xiao et al., 2019; Zhang et al., 2020; Zuo et al., 2020). Their key insight is that timestamps should be treated as part of the behavioral signal rather than as auxiliary metadata (Xue et al., 2023; Bae et al., 2023). The limitation is that intensity-based modeling often captures temporal occurrence patterns more naturally than rich action semantics; when behavioral meaning depends on heterogeneous attributes, multi-entity relations, or complex event content,

TPP representations may need to be combined with sequence, feature, or graph encoders (Yuan & Fang, 2025; Rossi et al., 2020).

*iv) Contrastive Learning:* Contrastive learning learns sequence representations by pulling semantically related histories closer in embedding space while pushing unrelated histories apart. In behavioral data, the central question is not only how to construct positive and negative pairs, but also what kinds of variations should be treated as behavior-preserving. Existing methods therefore mainly differ in how they define behavioral invariance through *sample selection*, *loss design*, and *data augmentation*.

- *Sample Selection*: Sample selection determines which histories should be compared during contrastive training. For behavioral data, this is particularly important because temporal irregularity, sparse observations, and latent intent can make naive positive and negative sampling misleading. PrimeNet (Chowdhury et al., 2023) addresses irregular timing through time-sensitive stratified sampling and fixed-time masking, preserving temporal dynamics in multivariate time series. IO-CLRec (Wang et al., 2025) focuses on semantic reliability in recommendation by reducing false negatives through alignment between user behavior sequences and latent intents. These methods show that contrastive pairs should preserve not only surface similarity, but also temporal validity and behavioral intent.

- *Loss Function*: Loss design specifies which behavioral factors should be aligned in the representation space. HORAE (Hu et al., 2025) incorporates temporal dynamics into contrastive learning for multi-interest pre-training, encouraging user representations to align with evolving time-aware interests. S3-Rec (Zhou et al., 2020b) introduces multiple auxiliary self-supervised objectives to capture item-attribute, sequence, and semantic correlations in recommendation. Rather than treating contrastive learning as a generic objective, these methods adapt the loss to the behavioral structure that should remain stable across views.

- *Data Augmentation*: Data augmentation defines how alternative views of the same behavioral history are constructed. CL4SRec (Xie et al., 2022) proposes crop, mask, and reorder operations to generate self-supervised signals from interaction sequences. CoSeRec (Liu et al., 2021a) extends this idea with substitute and insert operations to create more informative augmented views. DuoRec (Qiu et al., 2022) combines model-level augmentation through Dropout with supervised positive sampling to address representation degeneration. These methods improve label efficiency by generating training signals from unlabeled histories, but their success depends on whether the augmented views still correspond to plausible behavioral variants.

Contrastive learning frames behavioral representation as an invariance problem. Sample-selection methods define which histories are semantically or temporally comparable (Chowdhury et al., 2023; Wang et al., 2025), loss-design methods specify which behavioral factors should be aligned (Hu et al., 2025; Zhou et al., 2020b), and augmentation-based methods determine which perturbations preserve the same underlying behavior (Xie et al., 2022; Liu et al., 2021a; Qiu et al., 2022). This paradigm is especially valuable under label scarcity because it extracts supervision from relationships among histories rather than from external annotations. Its main risk is invalid invariance: if cropping, masking, reordering, substitution, or sampled positives remove rare decisive actions, break temporal causality, or create unrealistic trajectories, the learned representation may become behaviorally misleading even when the contrastive loss improves (Qiu et al., 2022; Wang et al., 2025).

*v) Adversarial Methods:* Adversarial Methods employ a discriminator to distinguish between generated fake samples and real samples, while the generator improves the quality of its outputs to deceive the discriminator. This framework has been effectively adapted for behavioral data processing. For example, RecGAN (Bharadhwaj et al., 2018) integrates RNNs and GANs through adversarial training to model temporal preferences for recommendations. Similarly, IRGAN (Wang et al., 2017) unifies generative and discriminative retrieval paradigms via a minimax adversarial game, where the generative model produces challenging samples and the discriminative model refines its ranking, achieving significant improvements in search, recommendation, and QA tasks. Further advancing this approach, ECGAN-Rec (Ni et al., 2023) combines contrastive learning with adversarial training to enhance sequential recommendation by balancing data sparsity and noise,

while CoSeRec (Liu et al., 2021a) also leverages adversarial training alongside contrastive learning to boost performance of downstream tasks.

Adversarial methods frame behavioral representation learning as a robustness problem: the model learns to distinguish true histories from plausible but difficult alternatives under sparsity, noise, or ambiguous negatives (Bharadhwaj et al., 2018; Wang et al., 2017). Their value lies in sharpening decision boundaries beyond standard likelihood, reconstruction, or contrastive objectives. However, adversarial training can be unstable and hard to interpret; if generated negatives are unrealistic, the model may learn generator artifacts rather than robust behavioral structure (Ni et al., 2023).

**Summary and takeaways.** The evolution of event sequence modeling shows a clear shift from local, short-horizon transition modeling toward architectures that better preserve long-range temporal context (Cheng et al., 2013; Kang & McAuley, 2018; Sun et al., 2019b). In supervised settings, Transformers often outperform recurrent baselines when sufficient history and scale are available, largely because they model global dependencies and parallelize training more effectively (Kang & McAuley, 2018; Li et al., 2020a). However, this does not make recurrent or continuous-time methods obsolete: when event timing is highly irregular, labels are scarce, or the task depends on fine-grained temporal calibration, architectures with stronger temporal inductive bias remain competitive (Du et al., 2016; Mei & Eisner, 2017; Zuo et al., 2020; Abushaqra et al., 2024). In self-supervised settings, masked reconstruction has emerged as the most reusable pretraining signal, while contrastive methods are effective only when augmentation preserves behavioral semantics (Sun et al., 2019b; Li et al., 2020c; Xie et al., 2022; Liu et al., 2021a). The main empirical bottlenecks are therefore not only model choice, but also how time irregularity, sequence length, and augmentation validity are handled.

### 4.3 Dynamic Graphs

Human behaviors frequently manifest as evolving interactions among individuals or entities over time, naturally lending themselves to representation as dynamic graphs (Kumar et al., 2019; Rossi et al., 2020). Specifically, a dynamic graph captures temporal variations in connections between entities (e.g., people, items), thereby providing a structured framework to analyze complex, time-dependent relational patterns.

Formally, behavioral data can be modeled as a dynamic graph $G = (V, E, T)$, where $V$ is a set of nodes representing individuals or entities, $E$ denotes the set of interactions or relationships (edges) among nodes, and $T$ explicitly encodes temporal information. Based on how the temporal dimension $T$ is defined, dynamic graphs are typically classified into two distinct types:

- **Discrete-Time Dynamic Graphs**: In Discrete-Time Dynamic Graphs (DTDGs), temporal dynamics are captured via a series of graph snapshots at discrete time intervals (Chen et al., 2024d): $\mathcal{G} = \{G_1, G_2, \ldots, G_T\}, G_t = (V_t, E_t), t = 1, 2, \ldots, T$, where each snapshot $G_t$ describes the structure of interactions within a fixed time interval, representing aggregated relationships.

- **Continuous-Time Dynamic Graphs**: Continuous-Time Dynamic Graphs (CTDGs) record each interaction as individual events at precise timestamps, thus offering finer granularity (Rossi et al., 2020): $G = (V, \mathcal{E}), \mathcal{E} = \{(u, v, t) \mid u, v \in V, t \in \mathbb{R}^+\}$, where each edge $(u, v, t)$ indicates a direct interaction between nodes $u$ and $v$ occurring at a continuous timestamp $t$.

Dynamic graph methods are central to behavioral representation learning because many behaviors are not properties of isolated subjects. A purchase connects a user, an item, a seller, and a time; a financial transaction links accounts through amounts and channels; a social action is meaningful partly because of who interacts with whom (Kumar et al., 2019; Rossi et al., 2020). Graph representations therefore encode behavior through evolving relational context: a node embedding reflects not only the subject's own history but also the structure, timing, and attributes of its neighbors and interactions (Rossi et al., 2020; Xu et al., 2020). This representation is suitable for the situations when behavioral outcomes depend on social influence, collaborative patterns, fraud rings, repeated counterparties, or cascading information diffusion. The cost is that model performance depends heavily on graph construction choices, temporal granularity, and scalable neighbor retrieval (Yu et al., 2023b).

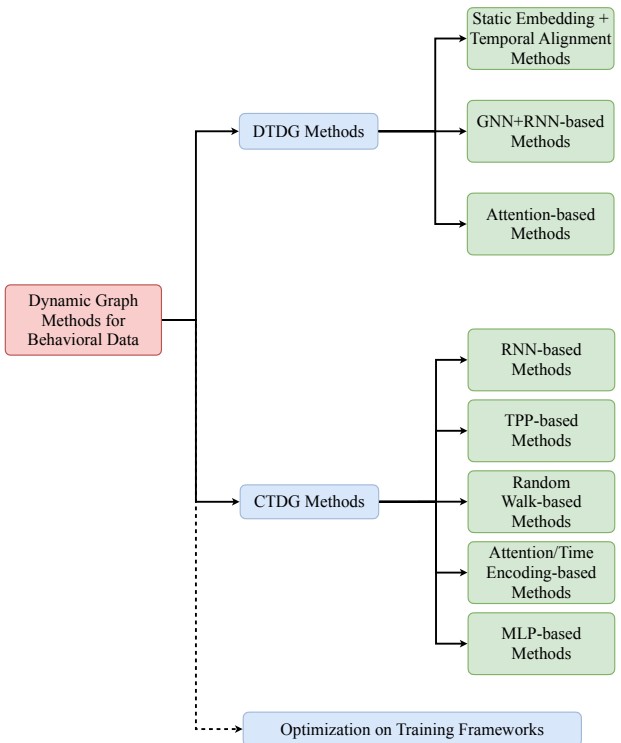

Figure 6: The dissection of dynamic graph representation learning methods.

Dynamic graphs typically facilitate two primary downstream analytical tasks: Dynamic Link Prediction and Node Classification. While a variety of other downstream tasks have also been explored, such as anomalous link detection (Postuvan et al., 2024), they are comparatively less studied and often application-specific; therefore, we focus primarily on the two dominant tasks in this paper. The primary output of representation learning methods for both DTDG and CTDG is typically dynamic node embeddings, denoted as $\mathbf{z}_v(t)$ for a node $v$ at time $t$. These embeddings encode both structural and temporal information accumulated up to time $t$, and serve as the basis for various downstream tasks.

- **Dynamic Link Prediction**: Predicting future interactions between nodes based on historical graph structures. Dynamic Link prediction is typically formulated as a self-supervised task, where the model is trained to predict future interactions based on historical data. Specifically, the future edges serve as implicit supervision signals, allowing the model to learn temporal patterns without requiring manual labels. Given a node pair $(u, v)$ and a query time $t$, the objective is to estimate the likelihood of a future edge:

$$\hat{y}_{uv}(t) = \sigma\left(f_{\text{link}}(\mathbf{z}_u(t), \mathbf{z}_v(t))\right), \tag{1}$$

where $f_{\text{link}}$ denotes a scoring function (e.g., dot product), and $\sigma(\cdot)$ is a sigmoid function. For behavioral data, link prediction corresponds to real-world tasks such as forecasting social connections, recommending potential friends, or predicting future collaborative partnerships.

- **Node Classification**: Identifying node attributes or categories based on their temporal relational context. Node classification is usually treated as a supervised task, where a set of nodes is associated with ground-truth labels. In practice, the node embeddings $\mathbf{z}_v(t)$ learned from the link prediction task are used, and an additional classification head is trained on top of these embeddings using the available label data:

$$\hat{y}_v = \text{softmax}\left(f_{\text{cls}}(\mathbf{z}_v(t))\right), \tag{2}$$

Table 3: Discrete-time dynamic graph representation learning methods.

| Method | Structural Encoding | Temporal Encoding | Year |
|---|---|---|---|
| Static Embedding + Temporal Alignment Methods | | | |
| Chakrabarti et al. (Chakrabarti et al., 2006), Chi et al. (Chi et al., 2009), Kim & Han (Kim & Han, 2009), Gupta et al. (Gupta et al., 2011), Yao et al. (Yao et al., 2016), Zhou et al. (Zhou et al., 2018) | Matrix factorization | Smoothness regularization or alignment | 2006-2018 |
| Hisano (Hisano, 2018) | Matrix factorization | Time window aggregation | 2018 |
| Sharan & Neville (Sharan & Neville, 2008) | Matrix factorization | Time-weighted adjacency matrices | 2008 |
| Ibrahim et al. (Ibrahim & Chen, 2015) | Matrix factorization | Exponential decay | 2015 |
| Ahmed et al. (Ahmed et al., 2016) | Low-rank adjacency | Temporal sampling strategies | 2016 |
| Singer et al. (Singer et al., 2019) | Random walk | Init. from previous step + fine-tuning | 2019 |
| DynGEM (Goyal et al., 2018) | Deep autoencoder | Regularization across snapshots | 2018 |
| DynamicTriad (Zhou et al., 2018) | Triadic closure | Temporal smoothness | 2018 |
| GNN+RNN-based Methods | | | |
| GCRN (Seo et al., 2018) | GCN | LSTM | 2018 |
| Narayan & Roe (Narayan & O'N Roe, 2018) | GraphSAGE | LSTM | 2018 |
| TGCN (Zhao et al., 2020) | GCN | GRU | 2020 |
| TNA (Bonner et al., 2019) | GCN | GRU | 2019 |
| VGRNN (Hajiramezanali et al., 2019) | VGAE | LSTM | 2019 |
| LRGCN (Li et al., 2019) | R-GCN | LSTM | 2019 |
| E-LSTM-D (Chen et al., 2021) | Autoencoder | LSTM | 2021 |
| EvolveGCN (Pareja et al., 2020) | GCN | GRU | 2020 |
| dyngraph2vec (Goyal et al., 2020) | graph2vec | LSTM/GRU | 2020 |
| TeMP (Wu et al., 2020) | GCN | GRU or Attention | 2020 |
| WD-GCN/CD-GCN (Manessi et al., 2020) | GCN | Modified LSTM | 2020 |
| HDGNN (Zhou et al., 2020a) | Heterogeneous random walk | Bi-RNN | 2020 |
| HTGN (Yang et al., 2021) | Hyperbolic attention-based GCN | Hyperbolic GRU | 2021 |
| GC-LSTM (Chen et al., 2022) | GCN | LSTM | 2022 |
| ROLAND (You et al., 2022) | GCN | Adaptive RNN | 2022 |
| RPC (Liang et al., 2023) | GNN | GRU | 2023 |
| SEIGN (Qin et al., 2023) | GCN-like message passing | GRU parameter adjustments | 2023 |
| RETIA (Liu et al., 2023b) | GCN | GRU + LSTM | 2023 |
| MegaCRN (Jiang et al., 2023) | Meta-graph learner | Custom GRU | 2023 |
| DEFT (Bastos et al., 2023) | GNN | RNN-based parameter evolution +Wavelet | 2023 |
| STGNPP (Jin et al., 2023) | GCN +Transformer | Continuous GRU | 2023 |
| WinGNN (Zhu et al., 2023b) | GNN | Sliding window | 2023 |
| SpikeNet (Li et al., 2023a) | GNN | SSN | 2023 |
| TTGCN (Li et al., 2024b) | Truss-based GCN | GRU | 2024 |
| Attention-based Methods | | | |
| DySAT (Sankar et al., 2020) | Graph attention | Graph attention | 2020 |
| TEDIC (Wang et al., 2021d) | Graph diffusion | Temporal convolutional network | 2021 |
| DyHATR (Xue et al., 2021) | Hierarchical attention | Temporal attentive RNN | 2021 |
| DREAM (Zheng et al., 2023) | Attention | Attention + Reinforcement learning | 2023 |
| STGNP (Hu et al., 2023a) | Dilated causal convolution | Cross-set graph convolution | 2023 |
| DTFormer (Chen et al., 2024d) | Transformer | Transformer | 2024 |

where $f_{\text{cls}}$ is typically a Multi-Layer Perceptron (MLP) or linear classifier. Practical tasks for behavioral data include detecting influential users in social networks, inferring user demographics, or identifying community roles based on interaction patterns.

In the following subsections, we systematically review and analyze representation learning methods designed for discrete-time and continuous-time dynamic graphs, respectively, followed by a discussion of the **Optimization on Training Frameworks** that are applicable across dynamic graph models. We highlight the core methodologies, recent advancements, and existing limitations of each category. It is important to note that this section focuses on introducing how the backbone models modeling behavioral data, while their associated downstream applications will be discussed in detail in Sec. 5.

### 4.3.1 Discrete-Time Dynamic Graphs

DTDGs capture evolving networks by recording the graph at successive fixed time intervals as a sequence of static snapshots (Feng et al., 2025). This turns behavioral representation into a problem of modeling transitions between network states: how the structure of users, items, accounts, or communities changes from one period to the next (You et al., 2022). The snapshot abstraction fits settings where behaviors

are naturally aggregated, observed, or acted upon at regular intervals, such as daily transaction networks, weekly communication patterns, or monthly community evolution (Chen et al., 2024d). Its strength is that it preserves coarse temporal order and relational structure; its cost is that within-window event order is collapsed. DTDGs are therefore useful when behavior is expressed through changing network states, but less suitable when exact interaction timing, burstiness, or event-level causality is the main signal (Skarding et al., 2021).

Most DTDG representation learning methods can be read as different ways of coupling within-snapshot relational encoding with across-snapshot temporal propagation (Kazemi et al., 2020; Feng et al., 2025). Given that the core distinction between DTDGs and static graphs lies in their temporal dynamics, we classify existing methods primarily based on how they model temporal dependencies, while also taking their structural modeling strategies into account. Therefore, we propose to classify existing DTDG representation learning methods into three-way taxonomy. **(a) Static Embedding + Temporal Alignment Methods**: These methods adapt static graph algorithms to each snapshot and then fuse or smooth the results over time. Often, an explicit smoothness or alignment constraint is added so that the embedding of each node changes gradually from one snapshot to the next. This "shallow" approach thus incorporates temporal smoothing but does not learn an explicit temporal model. **(b) GNN+RNN-based Methods**: These hybrid models use a GNN to encode the structure of each snapshot and an RNN to propagate information through time. Concretely, a GNN produces a node-embedding vector from each graph snapshot, yielding a time-ordered sequence of embeddings for each node. An RNN is then run on that sequence to capture temporal dependencies, so that each node's final representation reflects both its local topology (via the GNN) and its evolution history (via the RNN). This recurrent approach effectively learns dynamic updates as the graph changes. **(c) Attention-based Methods**: These models leverage self-attention mechanisms to capture both temporal and structural dependencies without relying on recurrent updates. By applying attention mechanism these models enable flexible modeling of long-range temporal interactions, addressing common limitations of RNNs such as vanishing gradients and sequential bottlenecks. This taxonomy therefore distinguishes the behavioral assumption behind the representation: smooth trajectory, recurrent state evolution, or attention-based context selection. Almost all existing methods still follow a broad two-stage pipeline in which GNNs, Transformers, or other encoders first capture structural information, and a temporal module then produces dynamic embeddings along the timeline. This pipeline is illustrated in Fig. 7a.

Note that this taxonomy is defined in a broad sense. For example, methods categorized under the GNN+RNN-based Architecture may not strictly adopt a GNN followed by a conventional RNN module; rather, we group together architectures that share a similar design principle. A comparison among DTDG representation learning methods is shown in Tab 3.

**Static Embedding + Temporal Alignment Methods** Early approaches to DTDG representation learning built upon techniques for static graphs, extending them to handle temporal evolution, as DTDGs are formed with a series of consecutive snapshots (Chen et al., 2024d). A common strategy is to apply matrix factorization or latent space models to each snapshot of the graph and impose temporal smoothness constraints between snapshots (Chakrabarti et al., 2006; Chi et al., 2009; Kim & Han, 2009; Gupta et al., 2011; Yao et al., 2016; Zhou et al., 2018; Kazemi et al., 2020). This effectively regularized the embedding trajectory of each node, assuming that node characteristics vary smoothly over time. Some methods aggregated multiple snapshots into a single decomposition: e.g., summing or averaging adjacency matrices over a time window (Liben-Nowell & Kleinberg, 2003; Hisano, 2018). However, such aggregation can lose temporal information (Kazemi et al., 2020), so later refinements introduced time-weighted sums (giving higher weight to recent snapshots) or exponential decay factors to emphasize current graph structure (Sharan & Neville, 2008; Ibrahim & Chen, 2015; Ahmed et al., 2016; Ibrahim & Chen, 2016). This line of work treats behavioral change as a gradually evolving latent trajectory. Such a formulation is well suited to stable behavioral networks, where user roles, community memberships, or relational preferences change slowly across periods. However, when the target behavior involves abrupt coordination, sudden diffusion, or rapid preference shifts, temporal smoothing and snapshot aggregation may obscure precisely the changes that should be detected (Kazemi et al., 2020).

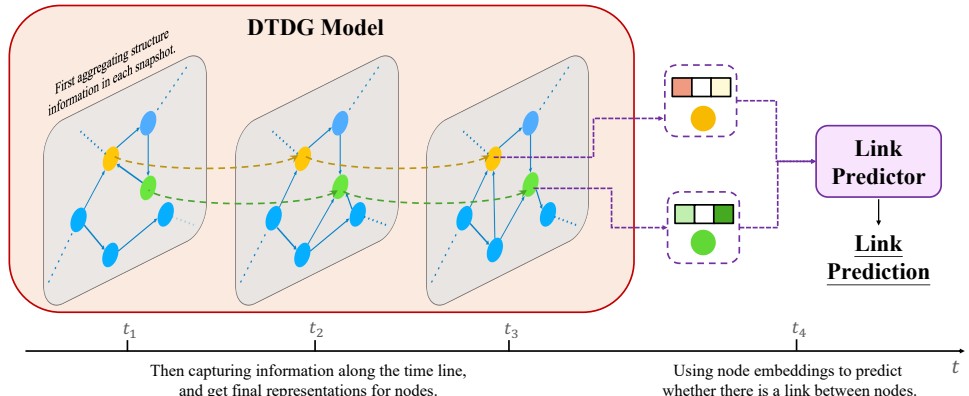

(a) The general pipeline of DTDG representation learning methods.

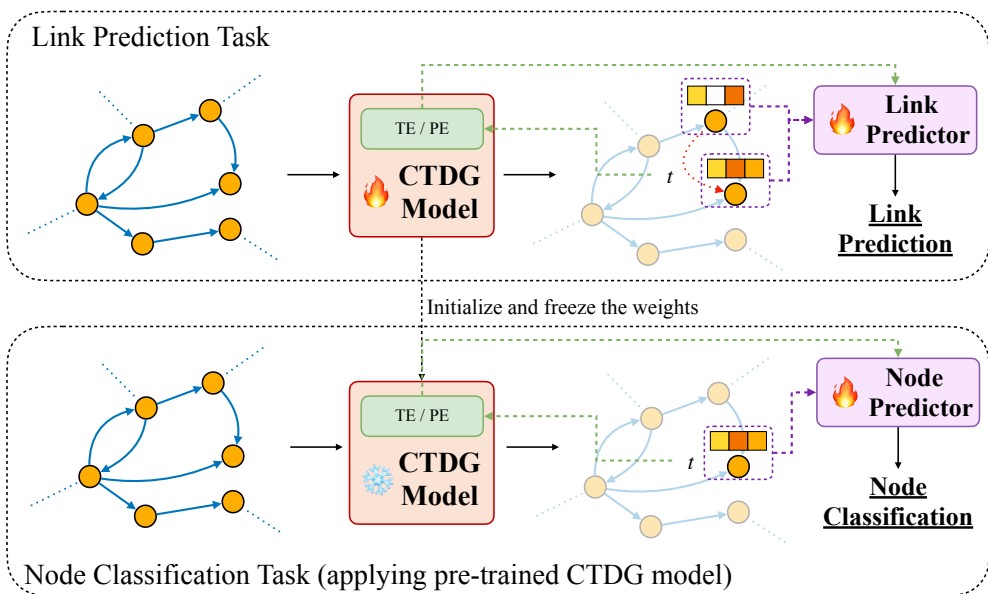

(b) The training pipeline of CTDG representation learning methods.

Figure 7: The pipelines of DTDG and CTDG methods.

Beyond matrix factorization, random walk-based embeddings (e.g., DeepWalk (Perozzi et al., 2014) and Node2Vec (Grover & Leskovec, 2016)) are also adapted: one could generate embeddings for each snapshot via truncated random walks and then align or smooth these embeddings over time. For instance, some methods initialized the embedding at time $t$ with the result from $t - 1$ and fine-tuned it, thereby leveraging past computations to speed up training (Chi et al., 2009; Singer et al., 2019). Along with the development of deep learning, DynGEM (Goyal et al., 2018) is one of the pioneering methods which utilizes deep autoencoders to learn node embeddings on each snapshot, while incorporating a regularization term to encourage the embeddings of adjacent snapshots to remain similar, thus capturing temporal smoothness. Dynamic-Triad (Zhou et al., 2018) further incorporated triadic closure process into structural modeling. Compared with matrix-factorization methods, random-walk, autoencoder, and triadic-closure approaches strengthen how each snapshot's local structure is encoded (Perozzi et al., 2014; Grover & Leskovec, 2016; Goyal et al., 2018; Zhou et al., 2018). This helps behavioral representation learning when local neighborhoods, repeated interaction routines, or closure patterns are important, such as stable communities or recurring collaboration groups. However, temporal evolution is still introduced mainly through initialization, alignment, or

smoothing, so the model may capture structural similarity better than the actual process by which behavior changes over time.

Overall, these pre-GNN methods laid the groundwork by combining static graph embeddings with temporal smoothing or forecasting. They captured evolving structure to some extent, but often in a two-step or loosely coupled manner (separate static embedding and temporal alignment), without end-to-end learning of dynamic patterns. They represent behavior as a smooth trajectory in embedding space. The underlying behavioral assumption is inertia: an entity's role, preference, or relational position usually changes gradually unless the graph structure changes substantially. This assumption is useful for stable communities or slowly evolving networks, but it can miss abrupt behavioral shifts such as sudden fraud coordination, viral diffusion, or rapid preference change.

**GNN+RNN-based Methods**  The emergence of deep learning on graphs introduced end-to-end models for DTDG representation learning that integrate GNNs (Scarselli et al., 2009; Yao et al., 2019; Zhang et al., 2019a) with sequence models (i.e., RNNs (Cho et al., 2014)). A dominant early paradigm for DTDGs is the combination of GNNs for structural encoding and RNNs for temporal encoding. We refer to this model design as the GNN+RNN-based architecture. Without loss of generality, and assuming a fixed node set $V$ across time steps for notational simplicity. Given a sequence of graph snapshots $\mathcal{G} = \{G_1, G_2, \ldots, G_T\}$, each snapshot is defined as $G_t = (V, E_t, X_t, F_t)$, where $X_t \in \mathbb{R}^{|V| \times d_x}$ is the matrix of node features, and $F_t : E_t \to \mathbb{R}^{d_e}$ provides edge features such as interaction types and weights. At each time step t, a GNN is applied to capture spatial dependencies within the snapshot. For each node $v \in V$, the GNN aggregates information from its neighborhood $\mathcal{N}_t(v)$, considering both node features and associated edge features:

$$h_v^{(t)} = \text{GNN}\left(v, \mathcal{N}_t(v); X_t, F_t\right). \tag{3}$$

The resulting structural embeddings $\{h_v^{(t)}\}v \in V$ serve as input to a RNN, which models temporal evolution by tracking each node's state across time:

$$z_v^{(t)} = \text{RNN}(z_v^{(t-1)}, h_v^{(t)}). \tag{4}$$

Here, $z_v^{(t)}$ denotes the final representation of node $v$ at time $t$, encoding both current structural context and historical dynamics. This two-stage design, which uses a GNN for intra-snapshot relational modeling and an RNN for inter-snapshot temporal modeling, enables end-to-end learning of evolving node representations in dynamic graphs.

In this architecture, each graph snapshot (at a discrete time step) is first processed by a GNN (such as a graph convolutional network (Yao et al., 2019)) to produce node embeddings that capture the structural neighborhood information at that time (Chen et al., 2024d). Then, an RNN (typically an LSTM (Graves, 2012) or GRU (Chung et al., 2014)) takes these node embeddings from each snapshot as sequential inputs to produce updated node states, thereby propagating information through time (Chen et al., 2024d).

The GNN+RNN-based architecture allows the model to learn how a node's representation should evolve based on its current neighbors and its past states.

Early examples include the model GCRN (Seo et al., 2018), which applies a GCN followed by an LSTM to capture structured sequences, and the work proposed by Narayan and Roe (Narayan & O'N Roe, 2018) using GraphSAGE (Hamilton et al., 2017) with an LSTM for representation learning. Subsequent efforts expand on this GNN+RNN paradigm with diverse architectural innovations. TGCN (Zhao et al., 2020) and TNA (Bonner et al., 2019) tightly integrate GCNs with recurrent units to learn temporally-aware node embeddings, while VGRNN (Hajiramezanali et al., 2019) brings in variational inference for modeling temporal uncertainty. LRGCN (Li et al., 2019), E-LSTM-D (Chen et al., 2021), and GC-LSTM (Chen et al., 2022) further enhance temporal modeling through stacked LSTMs or by embedding GCNs into recurrent gating mechanisms. These models convert each snapshot into relational context and then use recurrent updates to maintain an evolving node state (Seo et al., 2018; Narayan & O'N Roe, 2018). This design is useful when behavioral states depend jointly on current neighbors and historical exposure, such as repeated interactions, persistent preferences, or gradually changing risk. However, because updates occur at the

snapshot level, these models are less effective when within-window event order, irregular time gaps, or fine-grained interaction timing are central to the behavior (Zhao et al., 2020; Bonner et al., 2019; Hajiramezanali et al., 2019).

More recent advances emphasize structural adaptability and computational efficiency: EvolveGCN (Pareja et al., 2020) evolves GCN parameters over time via LSTM; dyngraph2vec (Goyal et al., 2020) combines static embeddings with RNNs for sequential modeling; ROLAND (You et al., 2022) adapts node embeddings via dynamic recurrent controllers. This subgroup extends temporal modeling from node states to the mechanisms that produce those states, such as evolving parameters, recurrent controllers, or dynamically updated embeddings. This is useful when the relationship between graph structure and behavior changes over time, for example when the same neighborhood pattern implies different risks, interests, or roles in different periods. However, the adaptation still follows the snapshot sequence, so it may miss rapid event-level transitions or behaviors that change between two observation windows (You et al., 2022). Methods such as TeMP (Wu et al., 2020), WD-GCN/CD-GCN (Manessi et al., 2020), and HDGNN (Zhou et al., 2020a) incorporate attention, hierarchical aggregation, and heterogeneity-aware modules to enhance long-term dynamics modeling. This line of work is useful for behavioral networks where different relation types, structural levels, or temporal ranges carry different meanings, such as user-item interactions, multi-type social relations, or hierarchical activity patterns. However, richer modules also introduce stronger modeling assumptions, making the representation more sensitive to how relation types, temporal ranges, or hierarchy are defined (Zhou et al., 2020a).

Additionally, models like HTGN (Yang et al., 2021), MegaCRN (Jiang et al., 2023), and DEFT (Bastos et al., 2023) explore hyperbolic spaces, meta-structure learning, and spectral wavelets respectively. Others target specific application contexts: RPC (Liang et al., 2023), RETIA (Liu et al., 2023b), and SEIGN (Qin et al., 2023) focus on temporal knowledge graphs; STGNPP (Jin et al., 2023) and WinGNN (Zhu et al., 2023b) address spatio-temporal prediction via Transformers and sliding windows; SpikeNet (Li et al., 2023a) leverages spiking neural dynamics; and TTGCN (Li et al., 2024b) exploits multi-scale truss structure for hierarchical temporal modeling. This group adapts the snapshot-sequence framework to more specialized behavioral settings, including geometric, spectral, spatio-temporal, temporal knowledge-graph, spiking, and multi-scale structures (Yang et al., 2021; Jiang et al., 2023). Such specialization helps when the behavioral problem has a clear structural prior, such as spatial locality, hierarchical organization, multi-hop reasoning, or multi-scale group formation. The drawback is reduced generality: a representation tailored to one behavioral domain may not transfer well when the relevant structure or temporal scale changes (Bastos et al., 2023; Li et al., 2024b).

Overall, these models effectively treat dynamic graph learning as a synchronous sequence modeling problem, where the RNN absorbs the node's history across snapshots. Despite architectural differences, all such GNN+RNN-based models address temporal dynamics by maintaining a state that carries information from prior snapshots. However, they also inherit limitations of RNNs: difficulty with very long-term dependencies and high memory usage for long sequences (Sherstinsky, 2020; Zhang et al., 2023d). Likewise, stacking many GNN layers per time step can cause over-smoothing of node features (Chen et al., 2020; Yang et al., 2020a). These challenges motivated the next generation of methods that seek to handle time without a standard RNN.

**Attention-based Methods**  Motivated by the success of attention mechanisms in capturing long-range dependencies across various sequence modeling tasks (Vaswani et al., 2017), researchers have started to explore their application to DTDG representation learning. This has led to the emergence of attention-based methods.

DySAT (Sankar et al., 2020) is a representative two-stage model that replaces GNNs and RNNs with structural and temporal self-attention, enabling long-range dependency modeling while retaining architectural interpretability. TEDIC (Wang et al., 2021d) uses graph diffusion and Temporal Convolutions (TCN) to model interaction dynamics, supporting permutation invariance and efficient global context capture. Dy-HATR (Xue et al., 2021) combines hierarchical attention over heterogeneous relations with temporal attentive RNNs for evolving pattern modeling. These methods use attention, diffusion, or hierarchical relation modeling to select which parts of the structural and temporal context should influence the current representation.

This is useful when behavioral signals are delayed, non-local, or relation-specific, because the model is not forced to compress all past information into a single recurrent state (Xue et al., 2021). However, since the input remains a sequence of snapshots, attention can improve cross-snapshot dependency modeling but cannot fully recover event order or causality inside each time window (Sankar et al., 2020; Wang et al., 2021d).

Later models push further toward non-recurrent and scalable designs: DREAM (Zheng et al., 2023), focusing on temporal knowledge graphs, incorporates multi-faceted attention with imitation learning to enhance reward modeling in multi-hop reasoning tasks. STGNP (Hu et al., 2023a) models extrapolation via a neural latent variable framework with causal temporal convolutions and Bayesian graph aggregation to capture uncertainty and spatial dependencies. DTFormer (Chen et al., 2024d) adopts a fully attention-based architecture, processing structural and temporal patterns jointly with multi-patch encoders and position encodings, achieving strong scalability and predictive performance. This subgroup moves DTDG modeling toward scalable non-recurrent spatio-temporal representation learning, using mechanisms such as multi-faceted attention, causal temporal convolution, latent variables, Bayesian aggregation, or patch-based encoders. These designs are useful when behavioral prediction requires access to long-range structural-temporal context without the bottleneck of recurrent propagation (Chen et al., 2024d). However, their ability to preserve meaningful behavioral dynamics depends heavily on positional encoding, patch construction, sampling strategy, or extrapolation assumptions, especially when interactions are sparse or unevenly distributed over time (Hu et al., 2023a).

Overall, these models collectively shift from recurrent temporal modeling to attention-based or convolutional paradigms, emphasizing scalability, long-term dependencies, and integration of complex spatial-temporal semantics.

**Summary and takeaways.** The DTDG literature suggests that snapshot-based dynamic graph learning is appealing largely because it reuses the machinery of static graph representation learning while introducing a lightweight temporal layer (Feng et al., 2025; Kazemi et al., 2020; Skarding et al., 2021). This design makes DTDGs comparatively accessible for benchmarking and system development, but it also introduces a clear trade-off: temporal evolution is captured coarsely through alignment, recurrence, or attention across snapshots rather than through event-level updates (Chakrabarti et al., 2006; Chi et al., 2009; Seo et al., 2018; Zhao et al., 2020; Sankar et al., 2020; Chen et al., 2024d). Consequently, existing methods differ less in whether they model time than in how tightly structural encoding is coupled with temporal propagation. Early GNN+RNN architectures improved end-to-end temporal modeling by combining intra-snapshot graph encoding with inter-snapshot recurrent updates (Seo et al., 2018; Zhao et al., 2020; Hajiramezanali et al., 2019; Pareja et al., 2020). However, long-horizon dependencies, sequential bottlenecks, and over-smoothing limited their scalability (Sherstinsky, 2020; Zhang et al., 2023d; Chen et al., 2020; Yang et al., 2020a), motivating the development of attention-based alternatives (Sankar et al., 2020; Chen et al., 2024d).

### 4.3.2 Continuous-Time Dynamic Graphs

As DTDGs discretize time into a sequence of graph snapshots, each capturing the network state within a fixed interval, this formulation simplifies modeling by leveraging established techniques for static graphs, but it may overlook fine-grained temporal information and introduce artificial boundaries between interactions. In contrast, CTDGs model data as sequences of timestamped interaction events, allowing for more precise capture of asynchronous, real-world dynamics. This finer granularity enables CTDGs to better represent time-sensitive behaviors in domains such as financial transactions (Kazemi et al., 2020; Lezmi & Xu, 2023) and e-commerce (Kang & McAuley, 2018; Rossi et al., 2020; Skarding et al., 2021). As a result, CTDGs have attracted growing attention as a complementary approach to snapshot-based modeling, particularly in scenarios where modeling temporal precision and event order is critical (Rossi et al., 2020).

Moreover, CTDGs are also commonly referred to as Temporal Interaction Graphs (TIGs) or event-based temporal graphs. In this paper, we regard these terms as equivalent and collectively refer to them as CTDGs. DTDGs can also be transformed into CTDGs (Jiao et al., 2024; Huang et al., 2025b; Chen et al., 2024d) by assigning timestamp information from each snapshot to the corresponding edges. Intuitively,

Table 4: Continuous-time dynamic graph representation learning methods.

| Method | Structure Encoding | Temporal Encoding | Memory-based | Year |
|---|---|---|---|---|
| RNN-based Methods | | | | |
| DeepCoevolve (Dai et al., 2016) | Implicit (via sequential interactions) | RNN | No | 2016 |
| JODIE (Kumar et al., 2019) | Implicit (via sequential interactions) | RNN + Projection | No | 2019 |
| Know-Evolve (Trivedi et al., 2017) | Implicit (via sequential interactions) | RNN | No | 2017 |
| RE-Net (Jin et al., 2020) | GCN | RNN | No | 2020 |
| HierTCN (You et al., 2019) | Implicit (via sequential interactions) | GRU + TCN | No | 2019 |
| DynGESN (Micheli & Tortorella, 2022) | Implicit (via sequential interactions) | Echo state network | No | 2022 |
| DyGNN (Ma et al., 2020) | GNN | LSTM | No | 2020 |
| AER-AD (Fang et al., 2023) | Local anonymous subgraph | GRU | No | 2023 |
| RTRGN (Chen et al., 2023f) | Implicit (via sequential interactions) | RNN | No | 2023 |
| TGN (Rossi et al., 2020) | Implicit (via message passing) | RNN + Memory | Yes | 2020 |
| NAT (Luo & Li, 2022) | Implicit (via sequential interactions) | RNN | Yes | 2022 |
| GDCF (Han et al., 2023) | Spatiotemporal GNN | RNN + Memory | Yes | 2023 |
| CDGP (Zhang et al., 2023f) | Community message passing | Time-aware aggregation | Yes | 2023 |
| TIGER (Zhang et al., 2023e) | Implicit (via message passing) | RNN + Dual memory | Yes | 2023 |
| RDGSL (Zhang et al., 2023c) | Implicit (via message passing) | RNN + Memory | Yes | 2023 |
| PRES (Su et al., 2024b) | Implicit (via message passing) | GMM-guided memory correction | Yes | 2024 |
| Ada-DyGNN (Li et al., 2024a) | Reinforced neighbor update | Time-based Policy | Yes | 2024 |
| SEAN (Zhang et al., 2024d) | Representative neighbor selector | RNN + Temporal-aware aggregation | Yes | 2024 |
| MemMap (Ji et al., 2024) | Latent memory-cell grid | Systematic memory routing | Yes | 2024 |
| MSPipe (Sheng et al., 2024) | Implicit (via message passing) | Staleness-aware update | Yes | 2024 |
| TPP-based Methods | | | | |
| HTNE (Zuo et al., 2018) | Historical neighbor modeling | Hawkes process | No | 2018 |
| M$^2$DNE (Lu et al., 2019) | Micro/Macro temporal co-occurrence | Hierarchical TPP | No | 2019 |
| GHN (Han et al., 2020) | Entity-level structure modeling | Hawkes process | No | 2020 |
| DyRep (Trivedi et al., 2019) | Attentive structural encoding | Multi-scale TPP | No | 2019 |
| LDG (Knyazev et al., 2021) | Edge-quality structure adaptive | Adaptive TPP | No | 2021 |
| TREND (Wen & Fang, 2022) | Implicit (via sequential interactions) | Hawkes process + Transfer function | No | 2022 |
| DynShare (Zhao et al., 2023c) | Implicit (via sequential interactions) | Personalized TPP | No | 2023 |
| EasyDGL (Chen et al., 2024a) | GAT | TPP + Correlation masking | No | 2024 |
| Random Walk-based Methods | | | | |
| CTDNE (Nguyen et al., 2018) | Timestamp-respecting walk | Skip-Gram over walks | No | 2018 |
| HNIP (Qiu et al., 2020) | Temporal random walk | Time-decay in walk sequence | No | 2020 |
| CAW (Wang et al., 2021c) | Causal Anonymous Walk | Hitting-count encoding | No | 2021 |
| NeurTWs (Jin et al., 2022) | Motif-guided random walk + ODE | ODE over walk path | No | 2022 |
| PINT (Souza et al., 2022) | Implicit (via message passing) | Provable temporal message passing | No | 2022 |
| TPNet (Lu et al., 2024) | Time-decayed walk matrix | Temporal relative encoding | No | 2024 |
| Attention/Time encoding-based Methods | | | | |
| TGAT (Xu et al., 2020) | Temporal self-attention | Functional time encoding | No | 2020 |
| TCL (Wang et al., 2021a) | Transformer | Functional time encoding | No | 2021 |
| OTGNet (Feng et al., 2023) | Open graph attention | Extended time encoding | No | 2023 |
| TGRank (Suresh et al., 2023) | Temporal attention ranking | Enhanced time encoding | No | 2023 |
| DHGAS (Ji et al., 2023) | Heterogeneous GNN + Attention | Time encoding | No | 2023 |
| DyG2Vec (Alomrani et al., 2023) | Temporal edge encoding | Time encoding | No | 2023 |
| SimpleDyG (Wu et al., 2024b) | Transformer | Time and position encoding | No | 2024 |
| Todyformer (Biparva et al., 2024) | GNN+Transformer | Position encoding | No | 2024 |
| DyGFormer (Yu et al., 2023b) | 1-hop neighbor + Co-occurrence | Time and position encoding | No | 2023 |
| APAN (Wang et al., 2021b) | Mailbox + Attention | Time encoding | Yes | 2021 |
| iLoRE (Zhang et al., 2023d) | Re-occurrence + Identity attention | Time encoding | Yes | 2023 |
| TDGNN (Qu et al., 2020) | GNN + Time-decay weighting | Exponential decay kernel | No | 2020 |
| DGEL (Tang et al., 2023) | Recent interactions | Time-aware normalization | No | 2023 |
| SUPA (Wu et al., 2023a) | Implicit (via sequential interactions) | Time modeling mechanisms | No | 2023 |
| FreeDyG (Tian et al., 2024) | Fourier-enhanced GNN | Functional time encoding | No | 2024 |
| CNE-N (Cheng et al., 2024) | Hash table-based memory | Temporal-diverse memory | Yes | 2024 |
| TG-Mixer (Zhang et al., 2024c) | Clustering patterns | Time encoding | No | 2024 |
| DyGMamba (Ding et al., 2025) | Mamba | Time encoding | No | 2025 |
| MLP-based Methods | | | | |
| GraphMixer (Cong et al., 2023) | MLP + Mean pooling | Fixed time encoding | No | 2023 |
| RepeatMixer (Zou et al., 2024) | MLP + Repeat-aware sampling | Time-aware aggregation | No | 2024 |
| BandRank (Li et al., 2025b) | Frequency-band MLP | Band-pass time filters | No | 2025 |

this conversion often results in multiple edges sharing the same timestamp, reflecting the coarser temporal granularity inherent in DTDGs.

In CTDGs, structural and temporal information are often intricately intertwined, necessitating joint modeling to accurately capture the evolution of interactions over time. As temporal dependencies critically influence both node behavior and structural dynamics (Feng et al., 2025; Jiao et al., 2025), we adopt a temporal modeling perspective as the primary taxonomy dimension, as it highlights the fundamental differences in how models encode, propagate, and reason about time-dependent interactions. Based on this criterion, we categorize existing CTDG methods into five principal classes. **(a) RNN-based Methods**: Use recurrent neural networks (e.g. LSTM/GRU) to sequentially integrate historical interactions, updating node embeddings over time. **(b) TPP-based Methods**: Employ temporal point processes (e.g., Hawkes processes) with conditional intensity functions to model event timing, capturing both historical decay and spontaneous interactions. **(c) Random Walk-based Methods**: Generate temporally constrained random walks that respect chronological order, aggregating spatio-temporal patterns from sequences of timestamped node transitions. **(d) Attention/Time Encoding-based Methods**: Apply self-attention mechanisms to sequences of event, enabling the model to capture long-range temporal dependencies without strict recurrence. However, since the attention mechanism is inherently position-invariant, time encoding is commonly introduced to explicitly incorporate temporal or positional information. In this category, we group together both methods that use attention-based architectures and those that, while not using attention mechanism, but still incorporate explicit time encoding to model temporal information. **(e) MLP-based Methods**: The necessity of complex temporal architectures has been challenged by adopting streamlined designs for temporal modeling. These methods leverage MLPs as the core building blocks for encoding temporal interactions, often combined with simple aggregation mechanisms and fixed or lightweight time encoding schemes. This taxonomy reflects different answers to the same behavioral representation question: how should an entity state be updated after each interaction? RNN-based methods update state sequentially; TPP-based methods represent event timing through intensity; random-walk methods summarize time-respecting relational paths; attention/time-encoding methods retrieve and weigh historical interactions; and MLP-based methods ask whether simpler encoders plus strong temporal priors are sufficient. These families are not merely architectural alternatives: they encode different assumptions about memory, temporal precision, repeated interactions, and scalability.

In addition to the primary temporal modeling paradigm, another important dimension in CTDG modeling is the use of memory mechanisms (ENNADIR et al., 2025). The memory mechanism introduces the idea of maintaining persistent, continuously updated representations of nodes to track their evolving states over time. Rather than recomputing node representations from scratch, memory-based models incrementally update these state vectors, enabling efficient, asynchronous updates and capturing long-term temporal dependencies. Such memory modules often incorporate temporal encoding or attention to regulate the influence of past events, making them well-suited for modeling evolving behaviors in real-world systems. Importantly, the use of memory is orthogonal to the choice of temporal modeling paradigm and can be viewed as a secondary classification dimension. Thus, by organizing models first by their temporal modeling strategy and then by their use (or absence) of memory, we obtain a two-level taxonomy that more precisely characterizes how CTDG models manage evolving temporal and structural information.

A comparison among CTDG representation learning methods is shown in Tab. 4. Nevertheless, regardless of taxonomy, most methods adopt a similar training paradigm (Chen et al., 2024e): the backbone model is first "pre-trained" via a self-supervised link prediction task, and the resulting node embeddings are then used for downstream tasks, where a projection head is trained. The pipeline is illustrated in Fig. 7b.

**RNN-based Methods** A seminal work that initiated RNN-based representation learning in CTDGs is DeepCoevolve (Dai et al., 2016). DeepCoevolve (Dai et al., 2016) introduced a co-evolutionary framework with mutually-recursive RNNs and temporal point processes, laying the groundwork for continuous-time event modeling. JODIE (Kumar et al., 2019) extends this by introducing a projection mechanism to estimate future node states given elapsed time, enabling real-time, time-aware predictions. This line of work treats behavioral dynamics as co-evolving node states that are updated after each timestamped interaction. Such representations are useful when the main behavioral question is how an entity's state changes after repeated interactions, such as evolving user preference, account activity, or partner-specific behavior. However, because the state is updated sequentially, these models can be sensitive to long event histories and may

struggle to separate persistent behavioral traits from short-term interaction noise (Dai et al., 2016; Kumar et al., 2019).

Subsequent models further enhance this paradigm in various ways: Know-Evolve (Trivedi et al., 2017) and RE-Net (Jin et al., 2020) focus on temporal knowledge graphs, combining RNNs with multivariate point processes or structural aggregation for autoregressive link prediction. HierTCN (You et al., 2019) replaces sequential updates with a hybrid RNN-TCN architecture to capture both short- and long-term user dynamics. DynGESN (Micheli & Tortorella, 2022) replaces trainable RNNs with an Echo State Network, offering a lightweight alternative for sequential modeling. These variants broaden recurrent CTDG modeling from pairwise interaction streams to knowledge-graph events, hierarchical temporal patterns, and lightweight recurrent dynamics. They are useful when behavior is expressed through typed relations, multi-scale temporal dependencies, or long-running streams where efficiency matters. However the limitation is that recurrent summarization still imposes a single evolving state, which may be insufficient when behavioral meaning depends on rich neighbor context or simultaneous multi-entity interactions (Jin et al., 2020; Micheli & Tortorella, 2022).

Recent work toward knowledge-graph foundation models further broadens this relational perspective beyond a single temporal graph. ULTRA (Galkin et al., 2024) studies transferable knowledge-graph reasoning by building relation representations conditioned on relation interactions, enabling inductive generalization across graphs with different entity and relation vocabularies. This makes it relevant for behavioral systems where user actions, items, diagnoses, accounts, or events are connected through typed relations. For behavioral representation learning, such models are promising when relational semantics transfer across domains, but they do not by themselves solve continuous-time behavioral modeling: timestamped interaction order, non-stationarity, and event intensity still require temporal encoders or CTDG-style mechanisms.

Incorporating structural information, DyGNN (Ma et al., 2020) integrates LSTM updates with GNN-based context under an event-driven scheme. AER-AD (Fang et al., 2023) focuses on inductive anomaly detection by learning anonymous edge representations and modeling temporal edge sequences via GRUs. Finally, RTRGN (Chen et al., 2023f) enhances temporal neighbor aggregation through recurrent state updates and a heterogeneous revision module, improving the completeness and accuracy of dynamic representation. By injecting structural context into recurrent updates, these models move beyond pure event-sequence modeling and begin to represent behavior as both a temporal and relational process. This is helpful for behavioral tasks where an event's meaning depends on surrounding neighbors, anonymous edge patterns, or heterogeneous relation types. The remaining challenge is scalability and memory: richer recurrent aggregation improves expressiveness, but also increases the cost of updating representations after every interaction (Fang et al., 2023; Chen et al., 2023f).

The memory mechanism in CTDG models is first proposed in the paper of TGN (Rossi et al., 2020). TGN (Rossi et al., 2020) maintains a timestamped memory vector $\mathbf{s}_i(t)$ for each node, which is updated on each interaction event. Upon an event between nodes $i$ and $j$ at time $t$, TGN computes messages $m_i(t)$ and $m_j(t)$ from their current memory states, the edge attributes, and time intervals. These messages are optionally aggregated (e.g., mean or last event) and passed through an RNN-based memory-updater (e.g., GRU) to produce updated node memories at time $t$. A node's embedding $\mathbf{z}_i(t)$ is then computed from its memory (and possibly neighbor context) for downstream tasks. This architecture enables long-term historical information retention and online embedding updates with event-level granularity. For behavioral representation learning, TGN-style memory changes the modeling unit from a recomputed sequence embedding to a persistent node state (Rossi et al., 2020). This allows entities to accumulate long-term behavioral history across asynchronous interactions, although the stored state remains sensitive to message aggregation, update frequency, and memory staleness (Zhang et al., 2023e).

The following memory-based methods largely follow the architectural framework established by TGN. These subsequent models retain this core pipeline—maintaining persistent node memories updated upon each interaction—but introduce various enhancements.

NAT (Luo & Li, 2022) introduces a neighborhood cache (N-cache) that stores multi-hop historical context per node, enabling query-time construction of temporally-aware embeddings. GDCF (Han et al., 2023) and CDGP (Zhang et al., 2023f) adopt memory modules for spatio-temporal prediction and community-

aware dynamics, respectively, by maintaining evolving node or community states. These methods expand memory from individual node histories to neighborhood, spatial, or community-level context. The resulting representations can reflect not only an entity's own past interactions but also the evolution of its local environment or group, while making memory selection and redundancy control more important (Zhang et al., 2023f).

Several works tackle memory staleness and adaptability: TIGER (Zhang et al., 2023e) employs dual memory (pre/post-event) and a restarter module to support parallel and context-aware updates; PRES (Su et al., 2024b) uses a GMM-based correction mechanism guided by gradient trends to adjust memory for robust long-term retention; Ada-DyGNN (Li et al., 2024a) introduces a learned policy for selective neighbor memory updates, enhancing efficiency and relevance. This subgroup addresses a central weakness of memory-based CTDG models: behavioral memory can become stale, inconsistent, or unnecessarily expensive to update. Restarting, correcting, or selectively updating memory improves robustness in dense and noisy streams, but also raises the risk of overlooking infrequent yet important behavioral changes (Li et al., 2024a).

Other models focus on addressing robustness and personalization: RDGSL (Zhang et al., 2023c) integrates dynamic noise filtering with attention-based edge selection; SEAN (Zhang et al., 2024d) proposes a plug-and-play neighborhood encoder with personalized neighbor selection and temporal aggregation. MemMap (Ji et al., 2024) maps nodes into latent semantic memory cells to encode high-level evolving patterns, while MSPipe (Sheng et al., 2024) decouples memory access from event order using a minimal staleness pipeline and resource-aware scheduling. These methods refine memory-based representation through noise filtering, personalized neighbor selection, semantic memory abstraction, and resource-aware scheduling. Such mechanisms make memory more robust and scalable, but they also make it less transparent which behavioral evidence is retained, filtered, or compressed (Ji et al., 2024; Sheng et al., 2024).

Overall, these methods highlight the growing trend of memory-centric architectures that move beyond traditional message passing to enable robust, asynchronous, and temporally rich dynamic graph representations.

**TPP-based Methods**  The TPP-based methods directly model event occurrence intensities based on the history of event timestamps and use learned intensity functions or personalized projection operators that condition on past events. The representation of temporal dependencies is encoded statistically through the intensity functions, thus, they do not require extra memory modules.

HTNE (Zuo et al., 2018) and GHN (Han et al., 2020) employ Hawkes processes to model the excitation effects of historical neighbors, capturing inter-event influence in dynamic graphs and TKGs. $M^2$DNE (Lu et al., 2019) introduces a hierarchical TPP framework for modeling both edge-level (micro) and network-level (macro) dynamics. DyRep (Trivedi et al., 2019) unifies communication and association dynamics into a multiscale TPP model with attention-based structural encoding. These methods represent CTDG behavior through event intensities rather than persistent memory states. They capture behavioral phenomena such as excitation after prior contacts, micro–macro network evolution, and changing association strength, but intensity-based representations often describe *when* events occur more naturally than *what* rich semantic content each event carries (Trivedi et al., 2019).

Later models extend expressiveness and interpretability: LDG (Knyazev et al., 2021) focuses on edge durability over time, while TREND (Wen & Fang, 2022) and DynShare (Zhao et al., 2023c) combine individual and collective temporal dynamics via learnable intensity functions or personalized projections. EasyDGL (Chen et al., 2024a) integrates TPP-modulated attention, a principled masking-likelihood training scheme, and graph spectral perturbations to support scalability and global interpretability. They make temporal intensity more adaptive by incorporating edge durability, individual and collective trends, personalized projections, and scalable likelihood training. These refinements better match behavioral persistence and shared temporal patterns, although they still depend on assumptions about how temporal influence should be parameterized (Chen et al., 2024a).

These models highlight the strength of TPP-based frameworks in modeling continuous-time dynamics with high temporal precision, offering strong support for event prediction and temporal reasoning in CTDGs.

**Random Walk-based Methods**   In Random Walk-based techniques, the model constructs representations by sampling time-respecting paths (walks) and aggregating information along these trajectories. This path-centric process captures spatio-temporal motifs without storing explicit node states. The node's representation is computed from sampled walks. As such, these methods avoid the complexity and overhead of maintaining per-node memory.

CTDNE (Nguyen et al., 2018) is a pioneering approach that generates timestamp-consistent walks and applies Skip-Gram for embedding learning. HNIP (Qiu et al., 2020) enhances this with high-order temporal modeling and a guided autoencoder for supervised walk reconstruction. They represent behavior through time-respecting paths, making temporal order part of the sampled structural context. They are well aligned with behaviors expressed through recurring trajectories or high-order interaction routes, but the resulting representation remains highly dependent on the walk strategy and may overlook rare yet important interactions (Nguyen et al., 2018; Qiu et al., 2020).

CAW (Wang et al., 2021c) introduces causal walks and anonymous encoding to support inductive learning and generalization to unseen nodes, achieving a good balance between expressiveness and scalability. NeurTWs (Jin et al., 2022) encodes temporal motifs from biased walks using neural ODEs and contrastive learning, capturing fine-grained temporal regularities. This subgroup strengthens temporal walks by emphasizing causality, anonymity, motif structure, and continuous-time regularity. These properties support inductive behavioral representation and recurring pattern recognition, while anonymous or motif-based encoding may discard identity-specific history needed for personalization or risk assessment (Wang et al., 2021c; Jin et al., 2022).

PINT (Souza et al., 2022) and TPNet (Lu et al., 2024) formalize temporal walk aggregation via principled message-passing frameworks, enabling provable temporal expressivity and scalable pairwise representation learning through time-decayed walk matrices or random feature propagation. By formalizing temporal walks as message passing or pairwise representation mechanisms, these methods make the path-centric view more scalable and theoretically grounded. They capture behavioral influence along multi-hop temporal paths, but remain less direct when the relevant signal is stored in node memory, rich event attributes, or non-path-based group dynamics (Souza et al., 2022; Lu et al., 2024).

These methods highlight the path-centric perspective in CTDGs, offering strong generalization, interpretability, and compatibility with transductive and inductive settings by leveraging temporal ordering and walk-based encoding.

**Attention/Time Encoding-based Methods**   TGAT (Xu et al., 2020) introduces temporal self-attention with functional time encoding based on Bochner's theorem (Xu et al., 2019), enabling translation-invariant modeling of time intervals and supporting inductive learning. TCL (Wang et al., 2021a) extends this with a two-stream Transformer and co-attention fusion, trained via contrastive learning to enhance robustness. OTGNet (Feng et al., 2023) and TGRank (Suresh et al., 2023) further extend temporal attention to handle open dynamic graphs and dynamic link ranking, respectively. These methods use attention and time encoding to determine which historical interactions shape the current representation. They capture long-range, time-sensitive, or open-world behavioral signals more flexibly than fixed recurrent updates, while remaining bounded by the sampled neighborhood or attention window (Suresh et al., 2023).

Beyond node-level temporal modeling, DHGAS (Ji et al., 2023) applies neural architecture search to discover optimal attention-based architectures for dynamic heterogeneous graphs. DyG2Vec (Alomrani et al., 2023) introduces a window-based attention-driven encoder that captures temporal motifs via relative time encoding and temporal edge features. It also adopts a non-contrastive self-supervised joint-embedding framework to learn task-agnostic representations. SimpleDyG (Wu et al., 2024b) explores the feasibility of directly applying vanilla Transformers by encoding node interaction sequences with temporal alignment tokens. Todyformer (Biparva et al., 2024) introduces a structure-aware tokenization mechanism that preserves holistic structural and temporal dependencies when converting dynamic graphs into Transformer-compatible sequences. DyGFormer (Yu et al., 2023b) enhances long-history modeling through neighbor co-occurrence encoding and a patching mechanism, effectively adapting standard Transformers for dynamic link predictions. This group adapts Transformer-style representation learning to dynamic graphs through architecture search,

temporal motifs, tokenization, alignment tokens, and long-history patching. These designs jointly model structural context and temporal dependency over long histories, but their effectiveness depends strongly on how graph events are converted into tokens, windows, or patches (Yu et al., 2023b).

While several attention-based approaches directly model temporal interaction graphs through self-attention layers—effectively capturing temporal dependencies via time encodings and long-range attention—they typically treat each interaction independently, without retaining historical context beyond attention windows. Several works propose to integrate explicit memory mechanisms into Transformer architectures.

APAN (Wang et al., 2021b) introduces a mailbox-based mechanism, where each node asynchronously receives and queues messages triggered by neighbor events, and performs attention-based summarization only when queried. iLoRE (Zhang et al., 2023d) further enhances temporal modeling via a hybrid design: an adaptive short-term updater filters noise, an attention-based long-term updater employs identity-aware attention, and a re-occurrence graph module captures repeated interactions. Memory-augmented attention models combine persistent state storage with query-time context retrieval. This pairing reduces the cost of repeatedly recomputing asynchronous histories, but it also requires careful coordination between what memory stores and what attention retrieves (Wang et al., 2021b; Zhang et al., 2023d).

Although attention-based models often employ time encodings to capture temporal ordering, time encoding is not exclusive to the Transformer architecture. A number of methods adopt explicit time encoding strategies within non-Transformer architectures. These models aim to integrate temporal signals into their representations through different time related embeddings.

TDGNN (Qu et al., 2020) applies exponential decay in its temporal aggregator to weight edges by recency. DGEL (Tang et al., 2023) updates embeddings in real time for recommendation, aligning structural evolution with normalization constraints. SUPA (Wu et al., 2023a) introduces time-aware short-term decay, time-sensitive propagation, and incremental learning tailored for multiplex heterogeneous graphs. These methods inject temporal information without relying on full recurrent or Transformer backbones, using decay, normalization, propagation, or incremental updates. They capture recency-driven behavioral change efficiently, but simple time encodings may be too limited for periodic, bursty, or multi-scale temporal patterns (Qu et al., 2020; Tang et al., 2023; Wu et al., 2023a).

To capture temporal patterns more expressively, FreeDyG (Tian et al., 2024) integrates functional time encoding with Fourier-based frequency enhancement for periodicity modeling. CNE-N (Cheng et al., 2024) compresses neighborhoods into hash-based memory and introduces temporal-diverse mechanisms for multi-scale dynamics. TG-Mixer (Zhang et al., 2024c) models interaction burstiness via silence decay and clustering-aware sampling of historical links. DyGMamba (Ding et al., 2025) leverages SSMs to capture long-term temporal dependencies by maintaining compact latent states, thereby enabling efficient and scalable temporal reasoning over long event sequences. This subgroup expands temporal encoding from simple recency to frequency, memory compression, burstiness, and long-range state-space dynamics. These mechanisms capture more complex behavioral rhythms and long-term dependencies, although temporal modules alone may not explain the structural or semantic reasons behind those patterns (Ding et al., 2025).

Overall, these methods demonstrate the versatility of temporal modeling in dynamic graphs, incorporating decay, frequency, memory compression, and burstiness to address complex, real-world temporal dynamics.

**MLP-based Methods**  Recent years have witnessed a surge in models leveraging complex architectures such as recurrent neural networks (RNNs) and Transformers to capture fine-grained temporal dependencies. While these approaches have achieved strong empirical performance, they often come at the cost of increased model complexity, slower training, and limited interpretability. As a response to this growing complexity, an emerging line of research has begun to question whether such sophisticated mechanisms are truly necessary. This has led to a re-examination of simpler alternatives—such as MLPs—as potentially sufficient for effective temporal representation learning, thereby inspiring methods like GraphMixer that demonstrate competitive performance through conceptually and technically streamlined designs.

GraphMixer (Cong et al., 2023) replaces RNNs and attention with MLP-based link encoders, mean-pooling aggregation, and fixed time encoding, offering a fast and effective alternative. RepeatMixer (Zou et al., 2024)

emphasizes repeat behavior modeling by introducing repeat-aware sampling and time-sensitive aggregation across different orders. BandRank (Li et al., 2025b) adopts a frequency-disentangled approach, decomposing temporal signals into multiple bands using adaptive filters, and employs frequency-enhanced MLPs along with a novel Harmonic Ranking loss to stabilize multi-scale supervision.

These models reflect a growing trend toward simpler yet targeted architectures that leverage signal decomposition, behavioral priors, and lightweight designs to achieve efficient and robust dynamic graph representation. However, they may be less expressive when behavior depends on complex multi-hop reasoning, heterogeneous event semantics, or long-range relational dependencies (Li et al., 2025b).

**Summary and takeaways.** The central contribution of CTDG modeling lies not merely in providing a finer taxonomy of dynamic graph methods, but in adopting a fundamentally different view of temporal behavior. In CTDG modeling, behavior is treated as timestamped events whose order and temporal spacing must be modeled with much greater precision than snapshot-based approaches allow (Kazemi et al., 2020; Skarding et al., 2021; Feng et al., 2025). Consequently, the key design axes of CTDG models shift from backbone architecture alone to temporal modeling strategies and memory mechanisms. Memory-based architectures have become influential because they support asynchronous event-level updates and long-term state tracking (Rossi et al., 2020; Kumar et al., 2019; Zhang et al., 2023e). Meanwhile, TPP-based, walk-based, attention-based, and more recent lightweight architectures illustrate different trade-offs among temporal precision, scalability, and architectural complexity (Trivedi et al., 2019; Nguyen et al., 2018; Xu et al., 2020; Cong et al., 2023). Together, these developments position CTDG models as a central paradigm for dynamic graph representation learning when fine-grained interaction order and timing are essential (Rossi et al., 2020; Xu et al., 2020; Yu et al., 2023b).

### 4.3.3 Optimization on Training Frameworks

In contrast to DTDG models, which are typically trained in an end-to-end manner for link prediction task, CTDG models commonly adopt a "pre-train, then fine-tune" training paradigm (Chen et al., 2024e), where a backbone encoder is first trained on temporal interaction data and task-specific predictors are subsequently optimized for different downstream tasks (as illustrated in Fig. 7b). Owing to the fine-grained temporal dependencies and event-driven modeling mechanisms in CTDGs, large-scale training and deployment of CTDG models are generally more complex than their DTDG counterparts. Therefore, in the following, we primarily focus on recent advances in training frameworks tailored for CTDG models.

Beyond fundamental CTDG representation learning methods, new frameworks have emerged to boost both efficiency and effectiveness. These frameworks are generally motivated by two key needs. (1) Scalability across large, evolving graphs: As CTDGs grow and update rapidly, traditional single-GPU or static training pipelines struggle to keep up. Frameworks designed for multi-GPU acceleration address this by parallelizing event processing, optimizing storage structures, and enabling distributed training. (2) Adaptability to downstream tasks under semantic and temporal shifts: CTDGs often feature complex, time-evolving relational patterns that standard pre-training setups do not fully capture. Prompt-based learning frameworks step in here: by freezing backbone parameters and injecting lightweight, task/time-conditioned prompts, they enable models to adapt dynamically across temporal domains and tasks—without heavy retraining.

Frameworks like TGL (Zhou et al., 2022), DistTGL (Zhou et al., 2023), SPEED (Chen et al., 2023e), GN-NFlow (Zhong et al., 2023), and Deep-Graph-Sprints (DGS) (Eddin et al., 2024) address the challenge of training on large-scale CTDGs by optimizing system-level bottlenecks. For instance, TGL introduces a time-sorted Temporal-CSR structure, parallel temporal sampling, and random-chunk scheduling to enable efficient neighbor retrieval and mini-batch updates, achieving large speedups on multi-GPU setups. DistTGL extends this to distributed environments, synchronizing node memory across workers to maintain throughput and convergence. SPEED contributes by offering Streaming Edge Partitioning and Parallel Acceleration Component, while GNNFlow layers on adaptive, time-indexed blocks, GPU/CPU hybrid caching, and distributed training support. DGS introduces a recurrent, sampling-free embedding architecture for CTDGs that updates node states online using vectorized forgetting factors, learnable feature embeddings, and mixed-mode automatic differentiation for efficient end-to-end learning. Collectively, these frameworks tackle storage, sampling, scheduling, caching, and parallelism to scale CTDG training reliably. These systems-level frameworks

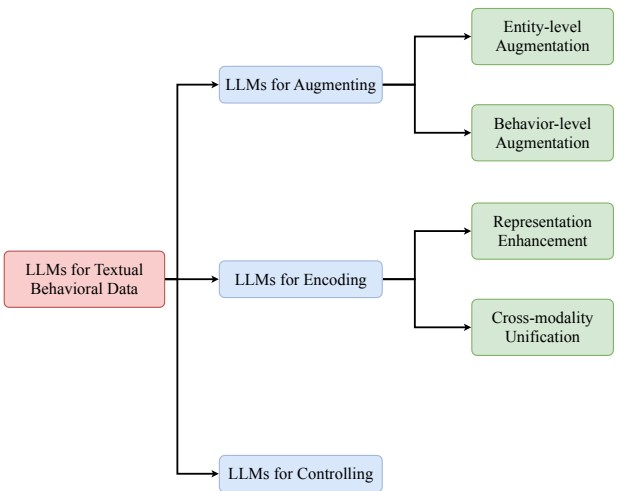

Figure 8: The dissection of the LLM methods for textual behavioral data.

address a practical constraint for behavioral representation learning: event-level models must keep pace with large and rapidly updating interaction streams. By optimizing temporal neighbor retrieval, memory synchronization, batching, caching, and parallelism, they make CTDG training more feasible at scale, although acceleration may introduce trade-offs such as stale memory, partitioned context, or approximated temporal dependencies (Eddin et al., 2024).

TIGPrompt (Chen et al., 2024e) innovatively brings the "pre-train, prompt" training paradigm to CTDG by injecting lightweight, time- and task-aware prompts, which is designed to enhance the adaptability of CTDG models by bridging the temporal and semantic gaps between pre-training and downstream tasks. It introduces three different temporal prompt generator that produces personalized, time-aware prompts for each node, enabling efficient adaptation without modifying the pre-trained model.

**Summary and takeaways.** Compared with DTDG models, CTDG learning places stronger demands on both scalable training infrastructure and flexible adaptation strategies because event-driven encoders must process large, rapidly evolving interaction streams while preserving temporal dependencies (Zhou et al., 2022; 2023; Chen et al., 2023e; Zhong et al., 2023). Recent frameworks therefore evolve along two complementary directions: system-level solutions that accelerate event-driven training on large dynamic graphs through optimized storage, sampling, scheduling, caching, and parallelism (Zhou et al., 2022; 2023; Chen et al., 2023e; Zhong et al., 2023; Eddin et al., 2024); and prompt-based approaches that enable efficient task and temporal adaptation of pre-trained CTDG encoders (Chen et al., 2024e).

### 4.4 Textual Data

Behavioral data are often expressed in textual form, arising from user-generated content and textual interactions, such as online reviews, chat logs, social media posts, and conversational records. Such textual behavioral data naturally carry rich semantic information about user intent, preferences, and contextual interactions.

LLMs, such as GPT-4o (OpenAI, 2024), Gemini (Gemini Team, 2024), PaLM (Chowdhery et al., 2023), and LLaMA (Touvron et al., 2023), have demonstrated remarkable capabilities in understanding and generating human-readable text, making them particularly well-suited for modeling textual behavioral data. Beyond traditional NLP tasks, their influence has extended to a wide range of deep learning applications, including education (Li et al., 2024c), healthcare (Nov et al., 2023), finance (Wu et al., 2023c), and recommendation systems (Bao et al., 2023).

Consequently, recent research has begun to explore the application of LLMs to behavioral data. By integrating extensive open-world knowledge and emergent abilities such as reasoning, LLMs can effectively analyze

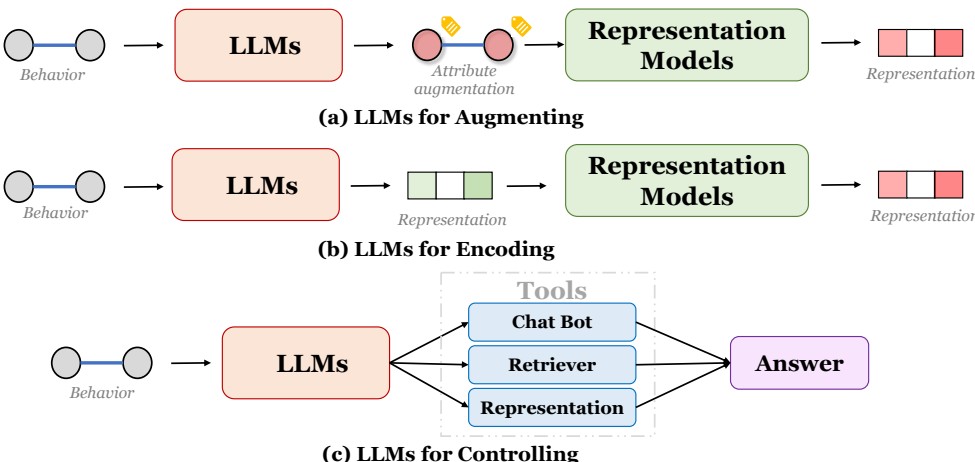

Figure 9: The pipelines of the LLM-related methods for behavioral data in textual form.

entity preferences from textual behavioral traces and enrich the semantic understanding of both entities and behaviors, thereby significantly improving the quality of behavioral data representations. Moreover, LLMs enable modeling more complex behavioral scenarios, such as conversational behaviors (Gao et al., 2023a), explainable behaviors (Chen, 2023), and real-time behaviors (Marripudugala, 2024; Gao et al., 2024a).

In this survey, the organizing axis is the LLM's role in the representation pipeline: **(1) LLMs for Augmenting** add semantic features or synthetic behavioral evidence, **(2) LLMs for Encoding** map text or serialized behavior into embeddings, and **(3) LLMs for Controlling** orchestrate retrieval, tools, reasoning, and specialist models. This role-based view clarifies when LLMs help behavioral analysis and where they introduce grounding, alignment, or auditability risks.

### 4.4.1 LLMs for Augmenting

LLMs for Augmenting refers to the process of selecting, transforming, and enriching raw behavioral data to create more expressive features/behaviors that are more readily understood by deep learning methods like time series models and dynamic graph approaches. In this process, LLMs take the original data (*e.g.*, entity descriptions, profiles, behavioral features, timestamps) as input and generate auxiliary textual data for augmentation, with various objectives such as enriching the features of entities/behaviors or synthesizing more informative behaviors for both training and evaluation. Based on the ways of augmentation mentioned above, it is easy to further classify these methods into two categories: **(a) Feature Augmentation** and **(b) Behavior Augmentation**.

From the perspective of behavioral representation learning, LLM-based augmentation changes the input space before representation learning begins. Rather than modifying the downstream encoder itself, these methods enrich what the encoder sees: feature augmentation adds semantic descriptions or external knowledge to entities and interactions (Xi et al., 2024; Liu et al., 2025; Torbati et al., 2023), while behavior augmentation expands, summarizes, or reshapes observed behavioral traces (Sidorenko et al., 2024; Chen et al., 2024c; Jiang et al., 2024). This can improve semantic coverage under sparse or noisy logs, but the added information must remain faithful to the original behavior; otherwise, augmentation may introduce plausible but unsupported signals (Carranza et al., 2024; Wu et al., 2025a).

**Feature Augmentation**   Integrated with exceptional reasoning capabilities and vast parameterized knowledge, LLMs can serve as experts to provide external knowledge during training. This enables them to generate auxiliary features that could promote the understanding of entities and their behaviors. For instance, KAR (Xi et al., 2024) leverages LLMs to produce user-side behavioral insights and item-side factual information in recommender systems, which are then incorporated as supplementary features into traditional recommendation methods. SAGCN (Liu et al., 2025) proposes a prompting mechanism to discover seman-

tic, aspect-aware behavioral interactions, providing more fine-grained interpretations of behaviors at the semantic level. In the CUP (Torbati et al., 2023), ChatGPT is used to extract and summarize each user's behavioral patterns into a set of concise keywords derived from their review history. These methods use LLMs to translate sparse behavioral evidence into semantic features that downstream models can consume. In behavioral representation learning, this helps when raw IDs, timestamps, or interaction counts do not fully express the intent, preference, or context behind an action. The main risk is semantic overreach: LLM-generated descriptions may make behavior easier to model, but they can also introduce interpretations that are not directly supported by the original logs (Torbati et al., 2023; Chen et al., 2024c).

By condensing user behavioral profiles into just 128 tokens, this method enables efficient encoding with smaller language models that have restricted context windows. In addition to the above methods that rely solely on LLM inference for feature augmentation, some methods (Cao et al., 2024a; Ren et al., 2025; Pandey & Singh, 2025) employ instruction fine-tuning with the foundational language models for various tasks across different domains, such as behavior categorization or intent prediction in e-commerce systems. Other methods also leverage LLMs for feature augmentation, including tasks such as text augmentation (Qin et al., 2025), graph completion (Ghanem & Cruz, 2024; 2025), and attribute generation (Regino et al., 2024; Gao et al., 2024c). Fine-tuning, together with the text-augmentation, graph-completion, and attribute-generation variants, extends this idea from a one-off enrichment step toward feature construction that adapts to the task and fills in what the record omits (Qin et al., 2025; Ghanem & Cruz, 2024; Regino et al., 2024). Its benefit is clearest when a missing attribute or an incomplete interaction graph is the actual bottleneck for the downstream model, and negligible when generation merely enlarges the feature space without adding behavioral signal.

**Behavior Augmentation**   Besides feature-level augmentation, LLMs are also good at generating synthetic behaviors, which can be used to empower the training datasets and subsequently boost the quality of representation generation. For example, in Precious2GPT (Sidorenko et al., 2024), generative models can be fine-tuned to produce realistic synthetic behavior logs, enriching the training datasets and improving model robustness. In (Carranza et al., 2024), privacy concerns are also being addressed by training LLMs under differential privacy constraints, enabling the generation of synthetic behavioral traces that protect identities while preserving data utility. In (Chen et al., 2024c), prompt engineering techniques are increasingly used to extract richer behavioral summaries, generate user profiles, and even simulate hypothetical interactions, thereby providing more nuanced signals for downstream models. Here the LLM acts not as an annotator but as a source of new behavioral evidence: synthetic logs, differentially private traces, and prompt-generated summaries substitute for observations that are sparse, legally restricted, or never recorded (Sidorenko et al., 2024; Carranza et al., 2024). The value of this substitution depends on realism: when the generated traces fail to preserve the timing, semantics, or population-level regularities of real behavior, the encoder may learn artifacts of the generator rather than genuine behavioral regularities (Chen et al., 2024c).

In addition, Trisum (Jiang et al., 2024) facilitates the integration of heterogeneous behavioral datasets by aligning features across domains, making it possible to leverage external sources for improvements. In TF-DCon (Wu et al., 2025a), data condensation strategies have also emerged, where LLMs generate compact yet informative behavioral samples, reducing the volume of training data required without sacrificing representativeness. In e-commerce, LLMs are fine-tuned for tasks such as query rewriting (KAG (Liang et al., 2025)), helping to bridge semantic gaps for less common or ambiguous behaviors. A complementary line directs LLM effort not toward producing more behavior but toward reconciling what already exists—aligning feature spaces across domains, condensing logs into compact yet representative samples, or rewriting queries so that rare behaviors become legible. The associated trade-off concerns fidelity: the same compression that reduces data volume and bridges vocabularies can remove the low-frequency cues that distinguish one behavior from another, so these methods are best evaluated by whether the retained signal still supports the target task rather than by accuracy alone (Jiang et al., 2024; Wu et al., 2025a; Liang et al., 2025).

**Summary and takeaways.** LLM-based augmentation expands behavioral representation learning by enriching both features and interaction signals. Feature-level methods leverage LLMs to inject external knowledge and semantic interpretations into entity and behavioral attributes (Xi et al., 2024; Liu et al., 2025; Torbati et al., 2023), while behavior-level methods generate, summarize, align, or condense interaction traces to

enhance data coverage and robustness (Sidorenko et al., 2024; Carranza et al., 2024; Chen et al., 2024c; Jiang et al., 2024; Wu et al., 2025a). Together, these approaches demonstrate how LLMs can act as data transformation interfaces, bridging raw behavioral logs and downstream representation learning models (Liang et al., 2025; Qin et al., 2025; Ghanem & Cruz, 2024; Regino et al., 2024).

### 4.4.2 LLMs for Encoding

In traditional deep learning methods, behavioral data are often formulated as the digitized features through techniques like one-hot encoding, which subsequently allows for an embedding layer to summarize and transform these features into dense vector representations. On the other hand, LLMs are built upon Transformer-based architectures, which excel at modeling contextual dependencies in textual data within behaviors. Such alignment between LLMs and embedding layers makes it intuitive to utilize LLMs as advanced feature encoders for more effective and context-aware representation generation.

Depending on the encoding strategy, existing methods that use LLMs for encoding can generally be categorized into two types: **(a) Representation Enhancement** methods, which utilize semantic information to further elevate representations; and **(b) Cross-modality Unification** methods, which exploit the generalization capabilities of LLMs to unify behavioral data from different modalities [1], thereby synthesizing more expressive and universal representations. The following sections will describe these two categories in detail.

In contrast to augmentation, LLM-based encoding changes how behavioral evidence is mapped into representations. Instead of adding new features or traces, these methods use language models to encode semantic context already present in profiles, descriptions, documents, reviews, or cross-modal signals (Bao et al., 2023; Jiang & Ferrara, 2025; Srivastava et al., 2024; Qiu et al., 2021; Li et al., 2023b). This makes LLMs useful as semantic interfaces between raw behavioral records and downstream representation learners, although their benefit depends on whether textual or cross-modal semantics are aligned with the behavioral objective (Zheng et al., 2024; Wang et al., 2024h; Zhang et al., 2025b).

**Representation Enhancement**   In this group of methods, LLMs are utilized as feature encoders for the available textual information within behavioral data, such as user profiles or item descriptions in recommendation systems (Bao et al., 2023), abstracts or introductions in citation networks (Jiang & Ferrara, 2025), and financial documents in financial systems (Srivastava et al., 2024). These approaches typically leverage LLM-based feature encoding in various applications, including news recommendation (Gao et al., 2024b), document ranking (Lu et al., 2025), tweet search (Chavinda & Thayasivam, 2025), and tag selection (Nazary et al., 2025). Treating review text, citation abstracts, or financial filings as a semantic layer over otherwise opaque entities allows a model to capture why an interaction occurred rather than only that it did. The benefit is largest where identity and structure are themselves uninformative, for instance when two items share a format but differ substantially in description (Bao et al., 2023; Srivastava et al., 2024). The corresponding limitation is that semantic richness does not guarantee behavioral relevance: text that is rich but off-target can dominate the representation, so task relevance rather than semantic abundance determines whether the added text helps (Gao et al., 2024b; Lu et al., 2025).

Additionally, beyond static textual features above, there also exist methods that utilize LLMs to capture dynamic and evolving semantic information, thereby modeling temporal dynamics and adapting to shifting preferences over time, especially in user-item behavioral data among recommendations. For instance, U-BERT (Qiu et al., 2021) enhances user representations by transforming review texts into sequences of dense vectors using BERT (Devlin et al., 2019), and then applies tailored attention mechanisms to effectively model user interests. Similarly, LLM4ARec (Li et al., 2023b) leverages GPT-2 to extract personalized aspect terms and latent features from user profiles and reviews, thereby improving the quality of recommendations. Unlike the aforementioned methods that directly use the textual features as LLM input, some methods leverage encoded embedding tokens to represent behavioral semantics. For example, LC-Re (Zheng et al., 2024) proposes to integrate collaborative semantic representation information to generate items from the entire item set for recommendation. Similarly, EAGER (Wang et al., 2024h) introduces a two-stream

---

[1]Different modalities refer to behavioral data with different distributions, such as series/graphs, different scenarios, different datasets, different systems, etc.

behavior-collaborative architecture that extracts semantic information into embedding tokens, thereby enhancing overall performance in recommendation systems. Extracting aspect-level terms, latent features, or collaborative semantic tokens advances encoding from a static text embedding toward a representation that tracks how interests shift over time (Zheng et al., 2024; Wang et al., 2024h). These methods also expose a fusion challenge between the semantic and interaction streams: if the two are not balanced, the model may overfit textual descriptions and underuse the interaction evidence that ultimately determines behavior (Qiu et al., 2021; Li et al., 2023b).

**Cross-modality Unification**    In addition to representation improvement methods, the generalization capability of LLMs offers a promising solution for transfer learning and cross-modality unification, where natural language serves as a bridge to align distribution differences across various modalities. For instance, OneLLM (Han et al., 2024) proposes a multi-modal LLM that aligns eight types of behavioral data to language within a unified framework, utilizing a multi-modal encoder and a progressive multi-modal alignment pipeline. RUNSRec (Zhang et al., 2025a) comprehends universal user behavioral patterns across different domains but also captures their inherent preferences to make recommendations. SocialMind (Yang et al., 2025a) employs human-like perception leveraging multi-modal sensors to extract both verbal and nonverbal behaviors, social factors, and implicit personas, incorporating these social behaviors into LLM reasoning for social suggestion generation. Brooks et al. (Brooks et al., 2023) present two models available from an API-based suite of emotional expression models that measure nuanced facial and vocal signals, providing rich, high-dimensional emotional expression estimates among human-computer interactions. By projecting actions, text, vision, speech, sensor, and affective signals into a single language space, these systems employ natural language as a coordination medium rather than a mere text encoder (Yang et al., 2025a; Brooks et al., 2023). Their central difficulty is alignment across channels that differ in temporal resolution, noise, and behavioral meaning: if alignment is too coarse, the unified representation suppresses the modality-specific evidence that motivated including each channel (Han et al., 2024; Zhang et al., 2025a).

Uni-CTR (Fu et al., 2025) utilizes a shared LLM to extract semantic representations at multiple layers, effectively capturing shared features across diverse domains. This approach enhances the model's ability to perform multi-domain recommendation by leveraging the underlying commonalities among different domains. Some works in recommendation systems (Tian et al., 2025; Lin et al., 2025) leverage unified cross-domain textual embeddings from a fixed LLM (e.g., Sheared-LLaMA (Touvron et al., 2023)) to tackle scenarios with cold-start users/items or low-frequency long-tail features. CROSS (Zhang et al., 2025b) further explores layerwise mixer to obtain better representations over textual features between the LLM-generated texts and behavior structures. For cross-domain and long-tail settings, a shared language-model embedding provides otherwise incompatible domains with a common semantic basis, and CROSS extends this by mixing LLM-generated text with behavioral structure in a layer-wise manner (Zhang et al., 2025b). Such grounding is an effective remedy for cold-start and sparsity, but it relies on an assumption that does not always hold—that similar wording implies similar behavior—which fails when the same text corresponds to different behavioral patterns across domains (Tian et al., 2025; Lin et al., 2025).

**Summary and takeaways.** LLM-based encoding enhances behavioral representation learning by introducing a semantic encoding layer. Representation enhancement methods leverage LLMs to extract contextual and semantic information from textual features, improving entity and interaction embeddings (Bao et al., 2023; Jiang & Ferrara, 2025; Srivastava et al., 2024; Qiu et al., 2021; Li et al., 2023b; Zheng et al., 2024; Wang et al., 2024h). Cross-modality unification methods further use language as a shared representation space to align heterogeneous behavioral data across domains and modalities (Han et al., 2024; Zhang et al., 2025a; Yang et al., 2025a; Brooks et al., 2023; Fu et al., 2025; Tian et al., 2025; Lin et al., 2025; Zhang et al., 2025b). Together, these approaches position LLMs as semantic interfaces that bridge diverse behavioral signals and downstream representation learning models.

### 4.4.3   LLMs for Controlling

As the number of model parameters continues to grow, LLMs have demonstrated remarkable *emergent abilities* that smaller models cannot achieve (Wei et al., 2022a), such as in-context learning, instruction following, self-reflection, and effective tool use. With these capabilities, LLMs can move beyond serving as components within a traditional pipeline. They can take on the role of pipeline controllers or decision-makers across

modules in behavioral data representation learning, thereby enhancing interactivity and interpretability of models. Compared with augmentation and encoding, controlling methods place LLMs above the representation pipeline rather than inside a single encoder. The LLM decides which tools to call, which behavioral evidence to inspect, how to decompose the task, and how to coordinate specialized modules (Wang et al., 2024g; Zhang et al., 2024g; Piao et al., 2025). This can make behavioral analysis more interactive and adaptive, but it also shifts part of the representation process into natural-language reasoning, where faithfulness, controllability, and error propagation become central concerns (Friedman et al., 2023; Cao et al., 2024b; Zhou et al., 2025c).

In recommendation systems, conversational agents like Chat-REC (Gao et al., 2023a) and RecLLM (Friedman et al., 2023) leverage LLMs to interact with users in a dialogue format. These models can infer user preferences from purchase histories, determine when to invoke recommendation APIs, and dynamically filter or rerank candidate items, ultimately delivering more accurate and personalized recommendations. Rather than relying on LLMs to manage the entire recommendation workflow, a more granular approach involves breaking down the recommendation task into smaller, manageable components. RecMind (Wang et al., 2024g) processes behavioral data using a self-inspiring prompting strategy and multi-step reasoning, incorporating tools like expert models, SQL databases, and search engines to enrich the analysis and interpretation of user behaviors. Placing an LLM in control reframes recommendation from a fixed embedding lookup into an interactive process that reasons over history, candidates, and external tools, enabling preference clarification and dynamic reranking (Gao et al., 2023a; Wang et al., 2024g). This flexibility, however, introduces additional points of failure: the final recommendation now also depends on prompt design, tool reliability, and whether the model interprets the behavioral evidence correctly (Friedman et al., 2023).

In finance systems, FinCon (Yu et al., 2024) mimics an investment firm's structure with "Manager" and specialist "Analyst" agents collaborating via natural language. A Manager LLM consolidates insights from news, reports, and quantitative analysts, while dual-level risk-control agents monitor market risk and update investment beliefs through self-reflection. As for multi-modal behavioral data, FinAgent (Zhang et al., 2024g) integrates a GPT-based core with specialized tools: it processes news text, stock charts (vision), and APIs for pricing and analytics. Its pipeline has modules for market intelligence (news summarization), trading behavior generation (via Chain-of-Thought), and portfolio execution. These finance-domain systems (Xiao et al., 2024; Ye et al., 2024c; Cao et al., 2024b) position LLMs not as stand-alone predictors but as decision-making cores in modular architectures. They integrate chain-of-thought reasoning, retrieval or vision tools, and multiple specialized agents to tackle complex tasks like trading, forecasting, QA, and fraud detection. Finance-oriented controllers illustrate how LLMs can coordinate heterogeneous behavioral evidence, including news, reports, market signals, charts, and quantitative tools. This helps when representation requires both structured numerical signals and narrative context, but it also raises stronger requirements for calibration, risk control, and auditability, since reasoning errors can directly affect high-stakes decisions (Yu et al., 2024; Zhang et al., 2024g).

In social networks, LLMs are central agents or modules in simulations of human behavior. They exhibit multi-turn reasoning and context-awareness, utilize tools, and even self-reflect. For example, AgentSociety (Piao et al., 2025) integrates environmental data with LLM agents that plan activities, interact, and communicate, where LLMs function as autonomous social actors whose emergent behaviors shed light on community phenomena. ProSim (Zhou et al., 2025c) is a modular framework for simulating prosocial behavior, where LLM agents autonomously decide whether to cooperate, punish unfairness, etc. It shows agents naturally exhibit human-like prosocial actions and adjust to policy changes. This illustrates LLMs as modular social actors capable of nuanced, multi-step social reasoning. The research suggests that LLMs can mirror collective dynamics across modules through techniques such as cooperation, tool usage, or reasoning capabilities, making them powerful for controlling the model pipeline in behavioral data processing. Social simulation systems extend LLM controlling from individual decision support to multi-agent behavioral dynamics. They provide a way to explore emergent social behavior, cooperation, and policy response, but simulated behaviors should be interpreted carefully because agent interactions may reflect model priors or prompt artifacts rather than real population dynamics (Piao et al., 2025; Zhou et al., 2025c).

**Summary and takeaways.** LLM-based controlling frameworks elevate language models from individual modeling components to orchestration layers in behavioral learning systems (Wang et al., 2024g; Zhang et al.,

Table 5: LLM-related methods for behavioral data.

| Methods | LLM Function | Modality Transformation | LLM Fine-tuning | Year |
|---|---|---|---|---|
| LLMs for Augmenting | | | | |
| Carranza et al. (Carranza et al., 2024) | Behavior augmentation | Language | No | 2024 |
| TF-DCon (Wu et al., 2025a) | Behavior augmentation | Language | No | 2025 |
| CUP (Torbati et al., 2023) | Feature augmentation | Language | No | 2023 |
| Precious2GPT (Sidorenko et al., 2024) | Behavior augmentation | Multi-omics → Language | Yes | 2024 |
| TriSum (Jiang et al., 2024) | Behavior augmentation | Sequence → Language | Yes | 2024 |
| Chen et al. (Chen et al., 2024c) | Behavior augmentation | Sequence → Language | No | 2024 |
| KAR (Xi et al., 2024) | Feature augmentation | Language | No | 2024 |
| Ghanem et al. (Ghanem & Cruz, 2024) | Feature augmentation | Graph → Language | Yes | 2024 |
| Ghanem et al. (Ghanem & Cruz, 2025) | Feature augmentation | Graph → Language | No | 2025 |
| APLe (Cao et al., 2024a) | Feature augmentation | Sequence (multi-modal) → Language | Yes | 2024 |
| KAG (Liang et al., 2025) | Behavior augmentation | Language | Yes | 2025 |
| SAGCN (Liu et al., 2025) | Feature augmentation | Sequence → Language | No | 2025 |
| Pandey et al. (Pandey & Singh, 2025) | Feature augmentation | Sequence → Language | Yes | 2025 |
| DyLas (Ren et al., 2025) | Feature augmentation | Language | Yes | 2025 |
| LLMs for Encoding | | | | |
| U-BERT (Qiu et al., 2021) | Representation enhancement | Language | Yes | 2021 |
| Social-LLM (Jiang & Ferrara, 2025) | Representation enhancement | Graph → Language | No | 2025 |
| Brooks et al. (Brooks et al., 2023) | Cross-modality unification | Sequence → Language | No | 2023 |
| UFIN (Tian et al., 2025) | Cross-modality unification | Sequence → Language | No | 2025 |
| Uni-CTR (Fu et al., 2025) | Cross-modality unification | Sequence → Language | No | 2025 |
| GNR (Gao et al., 2024b) | Representation enhancement | Language | Yes | 2024 |
| EAGER (Wang et al., 2024h) | Representation enhancement | Sequence → Language | Yes | 2024 |
| LC-Re (Zheng et al., 2024) | Representation enhancement | Language | Yes | 2024 |
| OneLLM (Han et al., 2024) | Cross-modality unification | Graph/Sequence → Language | Yes | 2024 |
| Lu et al. (Lu et al., 2025) | Representation enhancement | Language | Yes | 2025 |
| Chavinda et al. (Chavinda & Thayasivam, 2025) | Representation enhancement | Language | Yes | 2025 |
| Poison-RAG (Nazary et al., 2025) | Representation enhancement | Sequence → Language | No | 2025 |
| RUNSRec (Zhang et al., 2025a) | Cross-modality unification | Sequence → Language | Yes | 2025 |
| SocialMind (Yang et al., 2025a) | Cross-modality unification | Graph/Sequence → Language | No | 2025 |
| CROSS (Zhang et al., 2025b) | Cross-modality unification | Graph → Language | Yes | 2025 |
| Lin et al. (Lin et al., 2025) | Cross-modality unification | Sequence → Language | No | 2025 |
| LLMs for Controlling | | | | |
| RecLLM (Friedman et al., 2023) | Pipeline control | Language | No | 2023 |
| RecMind (Wang et al., 2024g) | Pipeline control | Language | No | 2024 |
| FinCon (Yu et al., 2024) | Pipeline control | Language | Yes | 2024 |
| FinAgent (Zhang et al., 2024g) | Pipeline control | Language | Yes | 2024 |
| TradingAgents (Xiao et al., 2024) | Pipeline control | Language | Yes | 2024 |
| TS-Reasoner (Ye et al., 2024c) | Pipeline control | Language | Yes | 2024 |
| RiskLabs (Cao et al., 2024b) | Pipeline control | Language | Yes | 2024 |
| AgentSociety (Piao et al., 2025) | Pipeline control | Language | Yes | 2025 |
| ProSim (Zhou et al., 2025c) | Pipeline control | Language | Yes | 2025 |

2024g; Piao et al., 2025). By leveraging reasoning, tool usage, and multi-agent coordination, LLMs can dynamically manage task decomposition, module interaction, and decision-making across complex pipelines. This paradigm demonstrates how LLMs can integrate heterogeneous models and behavioral signals into interactive and interpretable systems that support more flexible behavioral analysis and decision processes (Friedman et al., 2023; Cao et al., 2024b; Zhou et al., 2025c).

# 5 Application Scenarios

Having reviewed the foundational principles and empirical characteristics of major behavioral data representation methods, we now shift focus to their practical applicability across real-world tasks. This section examines the application of various data representations—including tabular, event sequence, dynamic graph, and textual—to key application scenarios across five prominent domains: **E-commerce**, **Healthcare**, **Social Media**, **Finance**, and **Gaming**.

## 5.1 E-commerce

In e-commerce, user interactions generate large-scale behavioral data, including clickstreams, search queries, and purchase records. Effectively learning representations from these data is crucial for powering intelligent downstream applications. We will now explore the application of behavioral data representations in two pivotal application scenarios in e-commerce: **(1) Personalized Search and Recommendation**, **(2) Enhanced User Understanding and Profiling**.

### 5.1.1 Personalized Search and Recommendation

As the core of e-commerce platforms, personalized search and recommendation aim to match users with the most relevant products by leveraging their behavioral data and contextual signals. Tabular data has long served as the backbone in e-commerce. Early studies leveraged clickstream logs and transaction tables to build aggregate user profiles for personalization (Mobasher et al., 2000). With the growth of large-scale e-commerce platforms, structured tabular features have become central to collaborative filtering (e.g., NCF (He et al., 2017)) and hybrid deep learning models for recommendation (e.g., DeepFM (Guo et al., 2017), xDeepFM (Lian et al., 2018), AutoInt (Song et al., 2019)). These tabular and hybrid feature models are useful when relevance can be inferred from stable user-item attributes and cross-feature interactions, but they provide limited access to the order in which user intent emerges.

By modeling a user's behavior as a time-stamped list of events, event sequence modeling has become a cornerstone of personalized recommendation. The foundation was laid by methods combining traditional matrix factorization with sequential signals, such as FPMC (Rendle et al., 2010), which integrated first-order Markov chains to capture dependencies on the user's last action. The paradigm shifted decisively with the introduction of deep learning. GRU4Rec (Hidasi et al., 2016) was a seminal work that first applied RNNs to model session-based clickstreams, demonstrating the power of learning complex sequential patterns.

To overcome the limitations of RNNs in capturing long-range dependencies, the field then embraced attention mechanisms, leading to the rise of attention-based models. SASRec (Kang & McAuley, 2018), for instance, pioneered the use of a unidirectional Transformer, allowing the model to weigh the importance of all past items to make more context-aware predictions. Building on this architectural foundation, further research has explored innovations in other dimensions. One key direction is enhancing item representations. MARN (Li et al., 2020b), for example, disentangles item features into modality-specific and modality-invariant components to create a richer representation before feeding it into a sequential model. Similarly, Moreira et al. (Moreira et al., 2021) combine tabular data with textual and image vectors for multi-modal feature learning within a Transformer ensemble. Another direction focuses on optimizing the sequence structure itself. SAGE (Wang et al., 2023b), inspired by DNA expression, organizes user behavior into critical "snapshots" to preserve essential signals while reducing sequence length and computational overhead. Recently, the fusion of tabular data with event sequences has become a popular paradigm in modern e-commerce recommendation. For example, IU4Rec (Wu et al., 2025b) leverages tabular product attributes to construct stable "Interest Units" (IUs) and jointly models fine-grained item sequences with higher-level IU sequences for improved recommendation. Sequence representations are therefore valuable when recommendation depends on intent drift, recency, or local behavioral context. Multi-modal and hybrid variants extend this temporal view by enriching each event with item content or higher-level attributes, but the core representation still centers on the user's ordered trajectory.

However, representing user-item interactions purely as sequences can overlook latent structural connections that evolve over time. To address this, dynamic graph modeling has been introduced to capture both temporal and structural patterns. There are two key applications of dynamic graph modeling in personalized search and recommendation: First, edge prediction. DGSR (Zhang et al., 2023b) connects different user sequences through a dynamic graph structure, and converts the next-item prediction task in sequential recommendation into a link prediction between the user node and the item node. While existing Temporal Graph Neural Networks (T-GNNs) often suffer from scalability and efficiency challenges due to high computational overhead, to address this limitation, a lightweight framework is proposed, named EAGLE (Li et al., 2025a), which integrates short-term temporal recency and long-term global structural patterns. Second, recommendation and ranking. SRJGraph (Zhao et al., 2022) jointly models search and recommendation

with a unified graph neural network to improve ranking performance by alleviating user behavior sparsity. This graph-based perspective is useful when collaborative signals, shared item transitions, or user-item-user dependencies are predictive, but it introduces higher modeling and training costs than purely sequential methods.

Nowadays, representing behavioral data with LLMs has initiated a paradigm shift, reconceptualizing user-item interaction sequences as a semantic language of consumer intent. This approach enables a more nuanced, context-aware understanding of user journeys, moving beyond traditional ID-based methods. This paradigm has evolved significantly: Initial work, such as BERT4Rec (Sun et al., 2019b), adapted the masked language model objective from BERT to behavioral data, demonstrating the power of deep bidirectional context. The modern approach, however, directly leverages generative LLMs. A seminal work, P5 (Geng et al., 2022), established a text-to-text framework using prompts to unify diverse recommendation tasks (e.g., sequential prediction, rating, explanation) within a single T5-based model. Following this, the field of LLM4Rec (Wu et al., 2023b) has burgeoned, exploring the use of powerful foundation models. Chat-REC (Gao et al., 2023a), which demonstrates how models like ChatGPT can be effectively prompted and guided to act as high-quality conversational recommender agents with minimal fine-tuning. Furthermore, methods like M6-Rec (Cui et al., 2022) and V-RECS (Podo et al., 2024) integrate multi-modal information (textual descriptions, images) into the sequence, creating richer item representations that allow the LLM to capture complex semantic and visual preferences, thus mitigating cold-start issues and improving recommendation quality. LLM-based recommendation further expands the representation space by translating user behavior, item content, and task intent into natural language. These methods are especially valuable under cold-start, explanation, or conversational settings, but their effectiveness depends on whether semantic representations remain grounded in observed user behavior.

Taken together, personalized search and recommendation illustrate a gradual expansion of what is considered informative user behavior. Tabular and hybrid feature models capture stable user-item attributes and cross features (He et al., 2017; Guo et al., 2017; Lian et al., 2018), sequential models capture order-sensitive preference evolution (Hidasi et al., 2016; Kang & McAuley, 2018; Sun et al., 2019b), dynamic graphs capture collaborative and structural dependencies beyond a single user timeline (Zhang et al., 2023b; Zhao et al., 2022; Li et al., 2025a), and LLM-based methods add semantic intent, conversational interaction, and multi-modal item understanding (Geng et al., 2022; Wu et al., 2023b; Gao et al., 2023a; Cui et al., 2022). The practical lesson is that recommendation quality often depends not on one universally superior representation, but on whether the model captures the most decision-critical signal: stable attributes, recent actions, collaborative structure, or semantic preference.

### 5.1.2 Enhanced User Understanding and Profiling

Enhanced user understanding and profiling differs from recommendation in that the goal is not only to predict the next item, but to build a persistent and interpretable representation of the user. Different modalities correspond to different profile granularities: tabular features summarize stable behavioral statistics, sequences capture evolving interests, graphs expose relational roles, and LLMs translate sparse histories into human-readable profiles.

Initially, this was achieved using tabular data processed through extensive feature engineering. This paradigm was dominant in critical industrial tasks such as Click-Through Rate (CTR) prediction and customer modeling, often employing models like logistic regression or factorization machines on massive, sparse feature sets (McMahan et al., 2013; Rendle, 2010; He et al., 2014). For example, the Repeat Buyer Prediction model (Liu et al., 2016) perfectly illustrates this approach by constructing static profiles for users and merchants from transactional statistics to identify potential loyal customers.

To overcome the limitations of static profiles, later research began leveraging deep learning to model user behavior sequences as event sequences. DUPN (Ni et al., 2018) demonstrated that a universal user representation could be learned from these sequences and shared across multiple tasks like search and recommendation. Further advancing this direction, HUP (Gu et al., 2020) introduced a hierarchical profiling method, which captures users' multi-granularity interests by modeling not only item-level sequences but also fine-grained

micro-behaviors. These methods improve the temporal sensitivity of user profiles, but they still primarily represent the user through observed actions rather than through broader relational or semantic context.

In dynamic graph modeling, enhanced user understanding and profiling is commonly formulated as a node classification problem. For instance, by constructing a tripartite graph connecting users, products, and personas, TriPer (Gupta et al., 2025) reformulates the multi-label persona identification task into a link prediction problem, allowing the model to generalize to new personas not seen during training.

Besides, LLMs excel at generating deep, qualitative insights into user behavior, creating rich, human-readable user profiles from raw interaction logs. This moves analysis from "what" a user will do next to "who" the user is. This paradigm is exemplified by architectures like the Language-based Factorization Model (LFM) (Zhou et al., 2024), which employs an LLM encoder-decoder to distill a user's interaction history into a compact, natural-language summary, thereby enhancing interpretability and addressing cold-start challenges. To refine this process, the GPG (Zhang, 2024) method demonstrates that adding a crucial intermediate summarization step is vital; this guides the LLM to effectively parse sparse data and extract salient personal features, significantly boosting personalization accuracy. Furthermore, this approach has evolved to capture the dynamic nature of user preferences. For instance, the framework from Sabouri et al. (Sabouri et al., 2025), explicitly models temporal dynamics by generating distinct profiles for a user's short-term versus long-term interests, offering enhanced explainability. Complementing this, the PURE (Bang & Song, 2025) framework focuses on the continuous evolution of profiles by systematically extracting information from new user-generated text, such as reviews, to maintain an up-to-date understanding of the user. Collectively, these innovations in generating, refining, and dynamically managing textual profiles are paving the way for e-commerce systems that are not only more accurate but also fundamentally more transparent and adaptive.

Compared with recommendation, user profiling places greater emphasis on producing a persistent and interpretable representation of the user. Tabular profiles summarize stable behavioral statistics (McMahan et al., 2013; Rendle, 2010; Liu et al., 2016), sequence models capture multi-granularity interest evolution (Ni et al., 2018; Gu et al., 2020), graph formulations introduce relational persona signals (Gupta et al., 2025), and LLM-based profiles translate sparse histories into human-readable summaries that can be updated over time (Zhou et al., 2024; Zhang, 2024; Sabouri et al., 2025; Bang & Song, 2025). The limitation is that increasingly semantic profiles may improve interpretability, but they must remain grounded in observed behavior rather than becoming plausible narratives detached from user actions.

## 5.2 Healthcare

Hospitals and clinics record a continuous flow of patient information, such as admissions, diagnoses, lab results, medications, and treatments. These clinical events form Electronic Health Records (EHRs), where representation learning must capture both longitudinal patient trajectories and clinically meaningful covariates. Compared with e-commerce or social media, healthcare applications place stronger emphasis on risk sensitivity, interpretability, and treatment relevance. We therefore organize this subsection around two primary scenarios: **(1) Disease Prediction and Risk Stratification** and **(2) Personalized Treatment Recommendation**.

### 5.2.1 Disease Prediction and Risk Stratification

Disease prediction and risk stratification aim to forecast future health events, such as the onset of sepsis, ICU transfer, or hospital readmission, based on a patient's historical clinical behaviors. Behavioral representation learning, particularly from event sequences and tabular EHR data, has enabled significant advancements in this field. For example, models like RETAIN (Choi et al., 2016a) and more recent attention-based frameworks (e.g., BEHRT (Li et al., 2020c)) learn temporal dependencies from sequences of visits, diagnoses, and medications to produce informative patient embeddings. ProtoEHR (Cai et al., 2025) further advances event sequence modeling by introducing a hierarchical prototype-based representation framework that captures relationships across medical codes, visits, and patients.

In addition, dynamic graph-based representations are increasingly used to model evolving patient states. MHGRL (Liu et al., 2024b) utilizes heterogeneous graphs to model patient data, integrating multi-modal

attributes and structural relationships for enhanced clinical prediction tasks such as disease prediction and EHR clustering.

Recently, the application of LLMs has expanded this field beyond direct sequence modeling into more interactive and reasoning-driven frameworks. EHRAgent (Shi et al., 2024) is an LLM-powered agent designed for natural language interaction with EHR systems, capable of multi-tabular reasoning and autonomous code execution to retrieve complex patient information.

These studies show that clinical risk prediction benefits from representations that preserve different levels of patient history. Sequence models encode visit-level temporal progression (Choi et al., 2016a; Li et al., 2020c; Cai et al., 2025), graph-based models capture heterogeneous patient-code-attribute relations (Liu et al., 2024b), and LLM agents provide interactive access to complex EHR tables and clinical reasoning workflows (Shi et al., 2024). The main challenge is that stronger representations must remain clinically interpretable and robust to missing, irregular, and institution-specific records.

### 5.2.2 Personalized Treatment Recommendation

Treatment recommendation systems rely heavily on accurately modeling patient behaviors and responses to previous interventions.

Tabular EHR data, which typically include demographic profiles, laboratory test results, and vital signs, remain a key source for personalized treatment modeling. For example, DeepSurv (Katzman et al., 2018) extends the classical Cox proportional hazards model with deep neural networks to capture nonlinear interactions between patient covariates and treatment effectiveness, thereby enabling individualized survival-based treatment recommendations.

Patient care histories are naturally represented as clinical event sequences, capturing diagnoses, medications, and procedures over time. To address the patient-specific variability inherent in such sequences, Lee and Hauskrecht (Lee & Hauskrecht, 2023) proposed adaptive event sequence prediction models that refine population-wide models into subpopulation- and patient-specific dynamics, enabling more accurate and individualized clinical predictions. Building on this line, SRL-RNN (Wang et al., 2018) introduces a supervised reinforcement learning framework with recurrent neural networks that leverages both clinician prescription signals and long-term survival rewards from EHR sequences, achieving more effective and safer dynamic treatment recommendations.

Knowledge graphs and LLMs introduce external reasoning into treatment recommendation. KARE (Jiang et al., 2025) combines LLMs with community-level knowledge graph retrieval to support reasoning-enhanced clinical predictions. GAME (Zhou et al., 2025a) uses knowledge graph integration and federated learning to harmonize multi-institutional EHR data while preserving privacy. More generally, medical LLMs such as Med-PaLM 2 (Singhal et al., 2025) demonstrate strong medical question answering capabilities and suggest the potential of language-based clinical reasoning systems.

Personalized treatment recommendation therefore requires representations that support intervention reasoning, not only risk prediction. Tabular survival models capture nonlinear covariate-treatment interactions (Katzman et al., 2018), sequence models represent evolving patient responses to prior interventions (Lee & Hauskrecht, 2023; Wang et al., 2018), and knowledge-graph or LLM-enhanced methods introduce external medical reasoning and cross-site harmonization (Jiang et al., 2025; Zhou et al., 2025a; Singhal et al., 2025). The main limitation is that treatment representations must be evaluated not only by predictive accuracy, but also by causal validity, safety, and clinical interpretability.

### 5.3 Social Media

The social media domain contains multi-modal, high-velocity behavioral traces (i.e., posts, likes, shares, comments, follows, clicks). To make these signals actionable, researchers have applied diverse representation paradigms. Below we summarize how they are used across key application scenarios and categorize them into four classes: **(1) User Profiling and Interest Modeling**, **(2) Content Recommendation and Feed Ranking**, **(3) Misinformation and Rumor Detection**, and **(4)Toxicity and Safety Moderation**.

### 5.3.1 User Profiling and Interest Modeling

In social media platforms, user profiling and interest modeling aim to infer latent characteristics—such as personality traits, preferences, and evolving interests—from rich behavioral traces. Natural language–based approaches focus on extracting signals directly from user-generated text. For instance, Peters demonstrated that GPT-3.5/4 can predict Big Five personality traits from Facebook status updates in a zero-shot setting, achieving moderate correlations with self-reports (Peters & Matz, 2024). More recently, Sabouri et al. (Sabouri et al., 2025) introduced LLM-TUP, which summarizes user histories into natural-language descriptions and encodes them with pre-trained language models, effectively capturing both short- and long-term preferences. Graph-based representations, by contrast, leverage the relational context of users. GREENER (Panayotov et al., 2022) modeled news outlets as nodes in a user–media bipartite graph, applying GNNs to predict factuality and bias, while HL-MRFs (Farnadi et al., 2020) integrated textual content, profile images, and social ties through a structured relational model to jointly infer user demographics and traits. Sequential modeling provides yet another perspective by emphasizing temporal dynamics of user behavior. Liu et al. (Liu et al., 2019) proposed SA-LSTM to jointly model user interaction sequences and the influence of friends, while Yang et al. (Yang et al., 2025b) used Bi-LSTMs with a specialized optimizer to capture temporal dependencies in ad click streams for interest prediction. Together, these methods show that user profiling in social media often requires both semantic self-expression and relational exposure signals.

### 5.3.2 Content Recommendation and Feed Ranking

In social platforms, content recommendation and feed ranking aim to surface the most relevant items from massive content streams by learning expressive user and item representations. Natural language–based methods leverage the semantic richness of text to enhance recommendation. For example, SPAR (Zhang et al., 2024a) uses pre-trained language models to encode user histories and candidate item text, introducing sparse poly-attention to effectively handle long engagement sequences and improve personalization. Sequential modeling approaches emphasize temporal dynamics in user behavior. Yang (Yang et al., 2024a) augments user interaction sequences with latent item relations mined from language models, thereby enriching sequential dependencies beyond simple co-occurrence. Meanwhile, Mamba4Rec (Liu et al., 2024a) replaces attention-heavy Transformers with selective state space models to capture long-term dependencies more efficiently, achieving both scalability and accuracy on large datasets. Graph-based approaches bring a structural perspective. ContextGNN (Yuan et al., 2025) integrates pair-wise subgraph representations with global two-tower embeddings, addressing the pair-agnostic limitation of traditional architectures. Similarly, CGAT (Yang et al., 2020b) exploits item knowledge graphs with attention over local and non-local neighbors, enhancing item embeddings and ultimately boosting recommendation accuracy. These methods indicate that feed ranking is not only a semantic matching problem, but also a temporal and relational filtering problem over rapidly changing user-content ecosystems.

### 5.3.3 Misinformation and Rumor Detection

Detecting misinformation and rumors on social media is a critical task, requiring models that can capture both the semantic content of claims and the structural and temporal dynamics of their spread. Natural language–based approaches exploit textual cues to directly judge veracity. Chen et al. (Chen et al., 2024b) conducted a broad empirical study showing that large language models can match strong baselines in content-only classification, while Ruiz et al. (Ruiz-Dolz & Lawrence, 2025) went further by integrating argumentation schemes and critical question answering to provide interpretable justifications for misinformation detection. Graph-based methods instead emphasize propagation and relational signals. For example, Liu et al. (Liu et al., 2024c) jointly modeled user correlations and diffusion trees, while GSMA (Ma et al., 2024a) improved GraphSAGE by weighting neighbors according to their structural positions in propagation graphs, achieving stronger accuracy on Weibo datasets. Sequential modeling brings temporal order into focus: Dong et al. (Dong & Qian, 2022) captured word- and sentence-level dependencies to improve detection with limited labels, and GETAE (Truică et al., 2024) combined Bi-RNN text encoders with propagation dynamics for robust ensemble predictions. Hybrid approaches such as MAGIC (Xu, 2024) further integrate multi-modal content and adaptive graph attention to capture both textual semantics and relational context. The main

lesson is that misinformation detection rarely depends on content alone; propagation structure and temporal response often determine whether a claim becomes harmful or credible.

### 5.3.4 Toxicity and Safety Moderation

Ensuring safe and respectful online communities requires robust moderation systems capable of detecting toxic or harmful behavior across diverse contexts. Natural language–based approaches remain the backbone of this task: lightweight yet effective transformer models such as Tiny-toxic-detector (Kamphuis, 2024) achieve competitive accuracy with minimal resources, while a recent survey (Shahi & Majchrzak, 2025) synthesize progress across multilingual datasets, highlighting the challenges of implicit language and cultural variation. Graph-enhanced models enrich textual detection with structured knowledge or relational context. MetaTox (Zhao et al., 2025) integrates a meta-toxic knowledge graph with LLMs to provide external toxic concept grounding, while ShieldVLM (Cui et al., 2025) extends moderation to multi-modal implicit toxicity, using deliberative reasoning across vision–language graphs to capture harmful meanings hidden in text–image combinations. Sequential methods bring temporal and contextual signals into play: Cheng et al. (Cheng et al., 2022) formulates moderation as a sequential decision problem to reduce bias in detection over comment streams, and Oyetayo et al. (Oyetayo et al., 2024) leverages dialogue context with RNN-based encoders to classify multiple types of toxicity simultaneously. Hybrid multi-modal frameworks further expand the scope, such as MHSDF (Prabhu & Seethalakshmi, 2025), which fuses textual, visual, and audio/video modalities through attention mechanisms to detect complex hate speech embedded in memes or multi-modal content. The challenge is that toxicity is highly context-sensitive: a representation must capture not only the utterance itself, but also conversational history, cultural cues, and multi-modal implication.

Across social media applications, the choice of representation is largely determined by whether the task depends more on user semantics, diffusion structure, temporal interaction, or multi-modal context. User profiling and content ranking often combine textual semantics with sequence or graph signals (Sabouri et al., 2025; Zhang et al., 2024a; Yuan et al., 2025); misinformation detection requires both claim semantics and propagation dynamics (Chen et al., 2024b; Liu et al., 2024c; Xu, 2024); and safety moderation increasingly depends on multi-modal and context-aware representations (Kamphuis, 2024; Zhao et al., 2025; Cui et al., 2025). The common challenge is that social behaviors are highly contextual: the same post, comment, or interaction can have different meanings depending on user history, network position, and surrounding discourse.

## 5.4 Finance

The data in the financial sector are inherently heterogeneous, temporally structured, and often high-stakes. Different behavioral data representations are prominent in specific financial applications. For risk management and fraud detection, event sequences, dynamic graphs, and LLMs are the primary modalities. In contrast, stock forecasting and portfolio optimization frequently rely on tabular data and LLMs. Consequently, we structure our review of these applications around two critical domains: **(1) Risk Management and Fraud Detection**, **(2) Stock Forecasting and Portfolio Optimization**.

### 5.4.1 Risk Management and Fraud Detection

In financial services, user behavioral event sequences have gained increasing attention in recent years for applications in risk management and fraud detection. These sequences effectively capture the distinctive digital behavioral signature of individual users, which is inherently more difficult to forge or manipulate compared to static profile data. By learning expressive representations from these event streams, models can capture subtle temporal patterns and behavioral deviations, thereby offering enhanced capabilities for proactive risk assessment and anomaly detection. Since SimpleRNN networks are prone to the vanishing gradient problem, Xavier et al. (Xavier et al., 2024) find that GRU outperforms both SimpleRNN and LSTM models. Branco et al. (Branco et al., 2020) treats user behavior as a sequence composed of many interleaved, unbounded sub-sequences, and use GRUs to build fraud detection models. FinDeepBehaviorCluster (Min et al., 2021) combines RNNs with intuitive features generated by risk experts to form a hybrid feature representation, which models the sequence of user clicks for fraud detection.UB-PTM (Liu et al., 2022a)

designs three agent tasks at different granularities, such as action, intention, and sequence levels, to detect fraud activities from unlabeled data. CLeAR (Huang et al., 2024b) devise an Intensity-Aware Transformer to capture dramatic changes in the intensity of transaction amounts and timing, while design two representation learning tasks that utilizes varying levels of contrastive learning to improve representation robustness.

Node representation learning on discrete dynamic graphs divides data into static snapshots at discrete time intervals, then each snapshot is processed with graph network models. Researches use time-related models to get the user (node) representations. DyHDGE (Wang et al., 2024f) jointly captures temporal transaction dependencies and dynamic heterogeneous graph structures to enable accurate real-time fraud detection in financial scenarios via integrated risk assessment and contrastive learning. EvolveGCN (Pareja et al., 2020) adapts the GCN model over time without relying on node embeddings. Instead, it captures the evolving nature of graph sequences by using a RNN to update the GCN parameters dynamically. FiFrauD (Khoda-bandehlou & Golpayegani, 2024) turns these real-time transactions into a stream of graphs, utilizing density signals in graphs in unsupervised training paradigm. Unlike discrete dynamic graphs, continuous dynamic graphs capture changes in nodes and edges as they occur continuously over time, rather than at discrete intervals. GADY (Lou et al., 2023) proposes a continuous dynamic graph model to capture complex time information, and uses a message-passing framework integrated with positional features to generate edge embeddings, which are subsequently decoded to detect anomalies. $EL^2$-DGAD (Chen et al., 2025a) integrates continuous-time embeddings into an attention-based graph encoder model. SSH-$T^3$ (Gu et al., 2025) then uses a self-supervised hierarchical two-tower Transformer framework that models long, multi-scenario payment behavior sequences via robust day-level pre-training and amount-aware scenario modeling to significantly improve financial risk assessment under scarce labels and complex real-world settings.

LLMs are fundamentally reshaping risk management and fraud detection, moving beyond static rules to analyze dynamic, context-rich data. They are notably applied to detect scams and fraud by analyzing textual and conversational data, where models like GPT-3.5 have proven effective at identifying phishing and other deceptive language (Jiang, 2024), leading to higher precision and recall than traditional methods (Korkanti, 2024). This capability is exemplified in advanced real-time systems, such as a RAG framework that analyzes phone calls to prevent impersonation and solicitation fraud as they occur (Singh et al., 2025). Beyond immediate threats, LLMs enable more comprehensive financial risk assessment by integrating diverse, multi-modal data sources. The RiskLabs framework pioneers this approach by fusing textual and vocal data from earnings calls with market time-series data to forecast financial volatility and variance, demonstrating a more holistic risk prediction capability (Cao et al., 2024b). Collectively, these applications showcase a shift towards a more predictive and integrative approach to managing financial risk.

Financial risk and fraud detection highlight the value of combining temporal, relational, and semantic representations. Sequence models capture user-specific behavioral signatures and abrupt deviations in action or transaction patterns (Xavier et al., 2024; Branco et al., 2020; Min et al., 2021; Liu et al., 2022a; Huang et al., 2024b); dynamic graphs expose coordinated or network-level fraud structures (Wang et al., 2024f; Pareja et al., 2020; Khodabandehlou & Golpayegani, 2024; Lou et al., 2023; Chen et al., 2025a); and LLMs bring textual, conversational, and multi-modal evidence into risk reasoning (Jiang, 2024; Singh et al., 2025; Cao et al., 2024b). The limitation is that high-stakes financial decisions require not only strong detection performance but also robustness, calibration, and auditability.

### 5.4.2 Stock Forecasting and Portfolio Optimization

In financial modeling, tabular data are advancing stock forecasting and portfolio optimization by leveraging heterogeneous numerical and categorical features to support accurate predictions and interpretable investment strategies.

For stock price forecasting, a common approach is to treat tabular data as a time series. Within this paradigm, the LSTM network has been widely adopted due to its ability to capture long-range dependencies and address the vanishing gradient problem (Ta et al., 2020; Han & Fu, 2023; Wang et al., 2024a). Building on this, recent works have explored the Transformer model, which leverages the attention mechanism not only for long-range dependencies but also to explicitly model complex interactions across multiple assets (Lezmi & Xu, 2023). In contrast to these sequence-focused methods, another line of research modeling the problem

by leveraging the inherent attribute of tabular data. For instance, Zhang et al. (Zhang et al., 2024e) propose a hybrid model combining a deep neural network with TabNet, a model specifically designed for tabular data, which achieves instance-wise feature selection through its unique attention mechanism. In the context of portfolio optimization, the dominant paradigm is a two-stage, "predict-then-optimize" framework. In this approach, the primary challenge is framed as a time-series forecasting problem as mentioned above, these forecasts are fed into classical optimization algorithms, most commonly the Mean-Variance Optimization (MVO) model, to determine optimal asset allocation (Ta et al., 2020; Wysocki & Sakowski, 2022; Singh et al., 2023). In addition, some end-to-end methods have also been proposed. SARL (Ye et al., 2020) employs a reinforcement learning agent to make end-to-end portfolio decisions, it explicitly addresses the challenge of information-rich environments by augmenting the agent's state representation to tackle the challenges of data heterogeneity and market non-stationarity.

At the same time, LLMs are revolutionizing stock forecasting and portfolio optimization by synthesizing diverse data sources for sophisticated investment strategies. A key innovation is their ability to enhance predictive models by integrating unstructured text with traditional market data. For instance, the Stock-Time (Wang et al., 2024c) architecture treats stock price series as a language sequence for prediction, while LED-GNN (Xu et al., 2025) uses an LLM to extract inter-stock relationships from news to build dynamic graphs, improving trading volume forecasts. Beyond pure prediction, LLMs are enabling more transparent and customized investment frameworks. The SEP (Koa et al., 2024) framework trains an LLM to generate human-readable explanations for its stock predictions through a self-reflective process, addressing the "black box" problem. Concurrently, research on Persona-Based Ensembles (Abe et al., 2024) leverages LLMs to simulate different institutional investor personas, creating adaptive strategies that outperform traditional methods in specific market conditions. These advancements collectively push towards more accurate, explainable, and personalized portfolio management.

Stock forecasting and portfolio optimization differ from fraud detection in that the central behavioral object is often the market trajectory rather than an individual user. Time-series and tabular models capture numerical market dynamics and covariate interactions (Ta et al., 2020; Han & Fu, 2023; Wang et al., 2024a; Zhang et al., 2024e), reinforcement learning supports end-to-end allocation decisions (Ye et al., 2020), and LLM-based methods integrate news, explanations, inter-stock relations, or investor personas into forecasting and strategy design (Wang et al., 2024c; Xu et al., 2025; Koa et al., 2024; Abe et al., 2024). The main challenge is non-stationarity: representations that perform well under one market regime may fail when temporal correlations, narratives, or investor behavior shift.

## 5.5 Gaming

The gaming domain provides an exceptionally rich environment for behavioral data analysis due to the high volume, velocity, and complexity of player interactions. To decipher player intent, skill, and engagement from these intricate data streams, researchers have developed diverse representation learning approaches, which have been applied to a range of application scenarios. In this section, we review representative tasks and highlight how different representation paradigms—such as event-sequence modeling and natural language–based approaches—are leveraged within them. We organize the discussion from three perspectives: **(1) Player Goal and Strategy Recognition**, **(2) Non-Player Character (NPC) Generation and Control**, and **(3) Toxicity Detection and Community Management**.

### 5.5.1 Player Goal and Strategy Recognition

Understanding player intent and strategy is a foundational task in game analytics. Sequence-based models have shown strong effectiveness here: early work used RNNs and LSTMs to frame goal recognition as sequence labeling, significantly improving prediction performance (Min et al., 2016). In the Real-Time Strategy (RTS) game domain, unsupervised replay embeddings learned from event sequences enabled automatic discovery and labeling of player strategies without manual annotation (Kantharaju & Ontañón, 2020). More recently, attention-based methods such as player2vec (Wang et al., 2024d) have tokenized in-game events and used self-supervised learning to capture long-range dependencies, offering robust player representations for intent recognition. Moreover, innovations like Masking by Token Confidence (MTC) further refined pre-training by

adaptively masking task-relevant actions, improving recognition accuracy in Massively Multiplayer Online Role-Playing Game (MMORPGs) (Wang et al., 2024e).

### 5.5.2 NPC Generation and Control

A second key task is generating believable and adaptive NPCs. Sequence learning approaches demonstrated early promise by using LSTMs with attention mechanisms to generate human-like action sequences, enhancing NPC realism and engagement (Zhao et al., 2021). With the advent of LLMs, this direction has expanded substantially: LLMs now serve as the reasoning "brains" of game agents. For example, MineDojo provides an internet-scale knowledge base to support embodied agents trained via LLMs (Fan et al., 2022), and Voyager showcases lifelong skill acquisition through automatic curriculum generation (Wang et al., 2024b). Other frameworks integrate LLMs with reinforcement learning, where the LLM constructs a skill graph and plan that the RL agent executes to solve long-horizon tasks (Yuan et al., 2023). Beyond planning, code-tuned LLMs empower NPCs to not only generate free-form dialogue but also produce executable, game-state-altering code, enabling deeply interactive experiences (Volum et al., 2022).

### 5.5.3 Toxicity Detection and Community Management

Ensuring healthy in-game communities is another important application scenario. Early linguistic analyses characterized the communication patterns of toxic players (Kwak & Blackburn, 2015), measured the prevalence of disruptive behaviors (Aguerri et al., 2023), and established links between toxicity and game outcomes (Märtens et al., 2015). Modern approaches extend this line by tackling real-time, multi-modal toxicity detection challenges (Ng et al., 2024). Here, LLM-based language representations have been especially valuable in understanding complex chat dynamics. Furthermore, reinforcement learning techniques, such as contextual bandits, have been introduced to allocate monitoring resources efficiently toward sessions with the highest predicted toxicity, improving both detection efficiency and player safety (Morrier et al., 2025).

Gaming applications show how behavioral representation learning extends from passive prediction to interactive control. Sequence representations capture player goals, strategies, and evolving action patterns (Min et al., 2016; Kantharaju & Ontañón, 2020; Wang et al., 2024d;e); LLM- and agent-based systems support open-ended NPC planning, skill acquisition, and interactive behavior generation (Fan et al., 2022; Wang et al., 2024b; Yuan et al., 2023; Volum et al., 2022); and toxicity moderation combines linguistic, contextual, and sequential signals to maintain healthy communities (Ng et al., 2024; Morrier et al., 2025). The distinctive challenge in gaming is that behavior is both strategic and environment-dependent, so representations must capture not only what players do, but also the goals, constraints, and feedback loops created by the game world.

### 5.6 Summary and Takeaways

Across application domains, the most effective behavioral representation is usually determined by which aspect of behavior is most decision-critical. Tabular representations remain competitive when the signal is largely aggregated and the prediction target is stable, as seen in CTR prediction, survival modeling, and portfolio forecasting (He et al., 2017; Guo et al., 2017; Katzman et al., 2018; Zhang et al., 2024e). Event sequences dominate when order, recency, and evolving user or patient state matter, such as sequential recommendation, EHR prediction, fraud detection, and player modeling (Kang & McAuley, 2018; Sun et al., 2019b; Li et al., 2020c; Huang et al., 2024b; Wang et al., 2024d). Dynamic graphs become advantageous when interactions among entities are themselves predictive, such as user-item graphs, fraud networks, social propagation, or knowledge-enhanced clinical reasoning (Zhang et al., 2023b; Wang et al., 2024f; Liu et al., 2024c; Jiang et al., 2025). Textual and LLM-based representations are most useful when latent intent, semantic context, explanation, or interactive decision-making is central (Geng et al., 2022; Gao et al., 2023a; Shi et al., 2024; Cao et al., 2024b; Wang et al., 2024b). This suggests that modality choice should be treated as a modeling decision rather than a data-format convenience: the same behavioral system may reasonably be represented in multiple ways depending on whether the task prioritizes prediction, retrieval, reasoning, intervention, or control.

Table 6: Cross-modal comparison of behavioral data representation learning paradigms.

| Dimension | Tabular Data | Event Sequences | Dynamic Graphs | Textual Data |
|---|---|---|---|---|
| Entity representation | Constructed from aggregated static features across behavioral attributes | Derived from encoding a chronologically ordered sequence of timestamped interactions | Learned from evolving relational structure and temporal interaction context | Derived from semantic encoding of behavioral text via pre-trained language models |
| Core assumption | Behavior is captured by aggregated features; temporal and relational structure is secondary | Chronological order and timing of events are the primary behavioral signals | Inter-entity relationships and their temporal evolution are the dominant predictive signal | Behavioral semantics can be expressed or enriched through natural language |
| Information preserved | Feature values, cross-feature interactions | Event order, inter-event intervals, sequential dependencies | Structural topology, relational dynamics, multi-hop context | Semantic meaning, intent, sentiment, world knowledge |
| Information lost | Temporal ordering, relational structure, fine-grained dynamics | Multi-entity relational context, global structural patterns | Non-relational entity attributes, aggregated statistics | Numerical precision, exact temporal intervals, structural topology |
| Typical supervision | Supervised classification/ regression; growing self-supervised pre-training | Supervised (classification); self-supervised (masking, autoregressive, TPP) | Self-supervised link prediction as pre-training; supervised node classification | Self-supervised LM pre-training; supervised fine-tuning; zero/few-shot prompting |
| Key strengths | Simplicity; interpretability (tree models); efficiency; robust to missing values | Captures temporal dynamics; mature pre-training ecosystem; scalable | Models relational context and structural evolution; strong for interaction-heavy data | Flexibility; zero-shot generalization; rich semantic understanding; world knowledge |
| Key limitations | Loses temporal/structural information; deep models often underperform trees on small data | Cannot model multi-entity structure; sensitive to sequence length; cold-start for new users | High computational cost; complex training pipelines; poor for isolated nodes | Token inefficiency for structured data; numerical imprecision; high inference cost; opacity |
| Best-fit applications | Risk scoring, user profiling, CTR prediction, tabular classification | Sequential recommendation, session prediction, clinical event forecasting, fraud sequence detection | Social network analysis, financial transaction monitoring, collaborative filtering, link prediction | Conversational agents, semantic user profiling, cross-domain transfer, explanation generation |

# 6 Cross-Modal Synthesis and Comparative Analysis

The preceding sections reviewed behavioral data representation learning methods organized by data modality and their downstream applications. While a modality-based taxonomy enables focused comparisons within each paradigm, it can also obscure insights that emerge only when methods are juxtaposed across modalities. In this section we provide an integrated cross-modal analysis. We **(1) examine shared technical building blocks**, **(2) compare the core assumptions and trade-offs of each modality**, and **(3) offer practical guidance on when to prefer one method over another given specific application requirements**. Table 6 provides a consolidated overview.

## 6.1 Shared Technical Building Blocks

Despite apparent differences in data structure, representation learning methods across modalities draw on a surprisingly convergent set of technical components. Identifying these shared building blocks clarifies both the deep connections among modalities and the modality-specific adaptations that differentiate them.

### 6.1.1 Attention Mechanisms and Transformers

Self-attention has become a near-universal backbone. In tabular learning, FT-Transformer (Gorishniy et al., 2021) and SAINT (Somepalli et al., 2021) treat columns as tokens and model inter-feature dependencies via self-attention. In event sequences, SASRec (Kang & McAuley, 2018) and BERT4Rec (Sun et al., 2019b) apply unidirectional or bidirectional attention over historical item interactions. In dynamic graphs, TGAT (Xu et al., 2020) and DyGFormer (Yu et al., 2023b) use temporal self-attention to aggregate neighbor interactions, while DySAT (Sankar et al., 2020) and DTFormer (Chen et al., 2024d) employ attention jointly over structural and temporal dimensions. In textual behavioral modeling, LLMs such as GPT (OpenAI, 2023) and BERT (Devlin et al., 2019) are inherently attention-based. The common motivation is that attention can capture long-range dependencies without the sequential bottleneck of recurrent models; however, each modality adapts the attention formulation differently: tabular methods attend across features within a single sample, sequential methods attend over time-ordered events, graph methods attend over temporally filtered neighborhoods, and textual methods attend over token sequences enriched with behavioral semantics.

### 6.1.2 Self-supervised Pre-training

Pre-training with self-supervised objectives is another cross-cutting theme that spans all four modalities, reflecting a shared philosophical commitment: learning general-purpose behavioral representations from the structure of the data itself before adapting to specific downstream tasks.

The dominant strategy varies by modality but follows analogous logic. Masking and reconstruction (inspired by BERT's MLM) appears in tabular learning (TabTransformer (Huang et al., 2020a), VIME (Yoon et al., 2020)), event sequences (BERT4Rec (Sun et al., 2019b), BEHRT (Li et al., 2020c)), and dynamic graphs (DyG2Vec (Alomrani et al., 2023)). Contrastive learning is used in tabular settings (SCARF (Bahri et al., 2022), CARTE (Kim et al., 2024)), sequential modeling (CL4SRec (Xie et al., 2022), CoSeRec (Liu et al., 2021a)), and graph methods (TCL (Wang et al., 2021a)). In dynamic graphs, the most prevalent form of self-supervised pre-training is temporal link prediction: the model learns to predict future interactions based on historical ones, using the natural temporal ordering of edges as an implicit supervision signal. This "future-supervises-past" formulation, adopted by nearly all CTDG methods (e.g., TGN, TGAT, DyGFormer) as the default pre-training task (Rossi et al., 2020; Xu et al., 2020; Yu et al., 2023b), is conceptually parallel to autoregressive prediction in sequences—both exploit the arrow of time to generate labels without manual annotation. For LLMs, self-supervised pre-training is foundational rather than optional: models like GPT are trained via next-token prediction and BERT via masked token prediction on massive text corpora. When applied to behavioral data, this pre-trained knowledge provides a semantic and reasoning substrate that can be adapted through prompting or fine-tuning, effectively making every LLM-based behavioral method a beneficiary of self-supervised pre-training, even when the downstream adaptation itself is supervised.

Despite these parallels, the augmentation and pretext task strategies required to generate effective training signals diverge significantly across modalities: tabular methods corrupt feature values, sequential methods crop or reorder sub-sequences, graph methods leverage temporal edge ordering, and LLMs rely on token-level masking or autoregressive generation over linguistic corpora. This suggests that while the pre-training philosophy, i.e., exploit the intrinsic structure of unlabeled data to learn transferable representations, is genuinely universal across behavioral data modalities, effective implementation remains modality-specific and constitutes a key engineering challenge in each domain.

### 6.1.3 Time and Position Encoding

Encoding "when" an event occurs is critical for all modalities except static tabular data. Functional time encodings (Xu et al., 2019; Kazemi et al., 2019) are shared between sequential models and CTDG methods, with recent learnable variants (e.g., LeTE (Chen et al., 2025b)) extending their expressiveness. Positional encodings from the Transformer literature (sinusoidal, RoPE (Su et al., 2024a), ALiBi (Press et al., 2022)) are applied in both sequential recommendation and dynamic graph Transformers. The key distinction is that event sequences typically encode a single timeline per user, whereas dynamic graphs must encode time for every edge in a multi-entity interaction history, making scalability a more pressing concern in graph settings. More detailed discussions about time and position encoding are provided in Sec. 7.

### 6.1.4 Memory and State-tracking Mechanisms

Maintaining persistent node or user states that evolve with each new interaction is a shared design pattern. In sequential modeling, RNN hidden states and memory-augmented networks (MANN (Chen et al., 2018)) serve this purpose. In CTDG methods, TGN (Rossi et al., 2020) formalized the memory module for graph nodes, spawning a large family of memory-based models. In LLM-based behavioral analysis, context windows and RAG play an analogous role by bringing past behavioral context into the model's reasoning scope. These mechanisms share the goal of long-term information retention but differ in update granularity: sequential models update per event, graph models update per interaction involving a node, and LLMs re-read or retrieve entire behavioral histories.

### 6.2    Core Assumptions and Trade-offs

Each modality encodes a different set of assumptions about which aspects of behavior are most important for downstream tasks. These assumptions introduce distinct trade-offs.

#### 6.2.1    Tabular Data: Aggregation-as-Abstraction

Tabular methods assume that behavior can be meaningfully summarized into a fixed-dimensional feature vector. This abstraction discards temporal ordering and relational structure but gains simplicity, interpretability, and computational efficiency. Tree-based models further benefit from natural handling of heterogeneous feature types and missing values. The main limitation is that aggregated features cannot distinguish behavioral patterns (e.g., binge purchasing vs. steady purchasing) from behavioral volumes, leading to information loss when temporal dynamics or interaction structures are decision-relevant.

#### 6.2.2    Event Sequences: Order as the Primary Signal

Sequential methods assume that the chronological order of events (and, in more recent models, their exact timestamps) is the most informative behavioral signal. This makes them powerful for next-action prediction but introduces challenges when events are sparse, irregularly timed, or involve complex multi-party interactions that a linear timeline cannot capture. Another practical limitation is sensitivity to sequence length: very short histories provide insufficient context, while very long histories increase computational cost and may introduce noise from outdated behavior.

#### 6.2.3    Dynamic Graphs: Structure as Context

Graph-based methods assume that behavior is best understood through the evolving relational context among entities. This is advantageous when inter-entity dependencies (e.g., social influence, collaborative patterns, cascading interactions) are predictive. However, graph methods introduce higher computational overhead, more complex training pipelines (especially for CTDGs), and greater sensitivity to graph construction choices (e.g., how to define edges, what temporal granularity to use for snapshots). The reliance on graph structure also means that isolated entities with few connections receive poor representations.

#### 6.2.4    Textual Data: Semantics as a Bridge

LLM-based methods assume that behavioral data can be meaningfully translated into or enriched through natural language. This assumption is strongest for inherently textual behaviors (reviews, conversations) and weakest for purely numerical or structural interactions. The main advantages are flexibility, zero-shot generalization, and the ability to inject world knowledge. The main limitations are token efficiency (serializing large tables or long sequences is expensive), numerical precision (LLMs struggle with exact quantitative reasoning), and the opacity of how the model weighs behavioral versus linguistic signals.

#### 6.2.5    Cross-cutting Trade-offs

Beyond the modality-specific assumptions discussed above, several fundamental trade-offs cut across all four modalities and recur whenever practitioners must choose among representation learning paradigms. These tensions are not fully resolvable as they reflect inherent design compromises, but making them explicit helps frame modality selection as a deliberate engineering decision rather than a default choice driven by data format alone.

- **Expressiveness vs. Efficiency.** Richer representations (graphs, long sequences) capture more behavioral nuance but require more computation and data.

- **Temporal Precision vs. Scalability.** CTDG and TPP-based methods offer fine-grained temporal modeling but are harder to train at scale than snapshot-based or tabular approaches.

- **Generalization vs. Inductive Bias.** LLMs generalize broadly but lack task-specific inductive biases; tree models and GNNs embed strong domain priors but transfer less easily.

- **Interpretability vs. Performance.** Tabular (especially tree-based) and symbolic sequential methods offer interpretable decisions, whereas deep graph and LLM methods often trade interpretability for predictive power.

## 6.3 Method Selection Guidance

A central practical question for researchers and practitioners is: given a behavioral dataset and a target application, which modality and method family should one choose? While no universal rule exists, the following guidelines synthesize the patterns observed across the literature reviewed in this survey.

**Task's Decision-critical Signal**  As noted in Sec. 5, the most effective representation is usually determined by which aspect of behavior is most decision-critical:

- If the task depends primarily on **aggregated user/entity profiles** (e.g., risk scoring, demographic classification), tabular representations are often sufficient and provide the best efficiency-accuracy trade-off, especially when tree-based models can be applied.

- If the task requires understanding **temporal order and recency** (e.g., next-item prediction, session-based recommendation, clinical event forecasting), event sequence modeling is the natural choice.

- If the task is sensitive to **inter-entity interactions and structural evolution** (e.g., fraud detection in transaction networks, social influence modeling, link prediction), dynamic graph methods are most appropriate.

- If the task involves **semantic reasoning, intent understanding, or explanation generation** (e.g., conversational recommendation, user profiling from text, financial report analysis), LLM-based approaches offer the richest representational capacity.

**Data Characteristics**  Beyond the task, several data properties influence method and modality choice:

- **Label scarcity:** Self-supervised pre-training is most mature for sequential (BERT4Rec-style masking) and graph (link prediction as pretext) settings. LLMs offer zero-shot or few-shot alternatives. Tabular self-supervised methods exist but remain less established.

- **Data scale:** Tabular models and shallow sequential models scale well. Deep graph methods, especially CTDGs, face scalability challenges on billion-edge graphs despite recent system-level optimizations (TGL (Zhou et al., 2022), SPEED (Chen et al., 2023e)). LLMs are constrained by context length and inference cost.

- **Feature heterogeneity:** When features include a mix of numerical, categorical, and textual attributes, tabular deep learning (e.g., FT-Transformer (Gorishniy et al., 2021)) or LLM serialization can handle heterogeneity naturally, whereas pure sequential or graph models may require careful feature engineering.

- **Temporal irregularity:** For highly irregular event timing, continuous-time models (Neural TPPs, CTDG methods) are preferable to discrete-time methods.

**Application-modality Alignment**  Drawing on the application analysis in Sec. 5, we observe recurring patterns of modality preference across domains:

- **E-commerce recommendation:** Dominated by event sequence methods (SASRec (Kang & McAuley, 2018), BERT4Rec (Sun et al., 2019b)), with growing adoption of LLM-augmented approaches (P5 (Geng et al., 2022), Chat-REC (Gao et al., 2023a)). Dynamic graphs are used when collaborative signals or user-item-user interactions are important (DGSR (Zhang et al., 2023b), EAGLE (Li et al., 2025a)).

- **Healthcare prediction:** EHR data naturally map to event sequences (BEHRT (Li et al., 2020c), RETAIN (Choi et al., 2016a)) and tabular features (DeepSurv (Katzman et al., 2018)). Knowledge graphs and LLMs (EHRAgent (Shi et al., 2024), KARE (Jiang et al., 2025)) are emerging for treatment reasoning.

- **Financial fraud detection:** Transaction networks favor dynamic graph modeling (EvolveGCN (Pareja et al., 2020), GADY (Lou et al., 2023), DyHDGE (Wang et al., 2024f)) and behavioral sequence analysis (CLeAR (Huang et al., 2024b), UB-PTM (Liu et al., 2022a)). Tabular features remain important for aggregated risk scores.

- **Social media analysis:** User profiling leverages textual/LLM methods (LLM-TUP (Sabouri et al., 2025), GPT-based personality inference) and graph methods (GREENER (Panayotov et al., 2022), HL-MRFs (Farnadi et al., 2020)). Content recommendation blends sequential and graph approaches.

- **Gaming:** Player behavior modeling primarily uses event sequences (player2vec (Wang et al., 2024d), RNN-based goal recognition). LLMs are increasingly adopted for NPC control (Voyager (Wang et al., 2024b), MineDojo (Fan et al., 2022)).

**Multi-modal Combinations.** In practice, many real-world systems combine multiple modalities. For example, tabular user features are often concatenated with sequential embeddings in recommendation systems (Wu et al., 2025b), and LLM-generated features are fed into graph or sequence models (Xi et al., 2024; Liu et al., 2025). The most robust strategy is often to match the primary modality to the task's decision-critical signal and augment with secondary modalities to address coverage gaps (e.g., using LLMs to enrich cold-start items with textual descriptions, or using graph structure to model collaborative signals that sequences alone cannot capture).

**Cross-Modal Comparison** Tab. 6 provides a systematic comparison across the four behavioral data modalities along key dimensions including representation type, core assumptions, typical supervision, evaluation protocols, strengths, limitations, and application alignment. We also include Tab. 7 to compare the dominant method families within each modality along practical axes.

Specifically, we evaluate each method family along five practically oriented dimensions that we consider most relevant for guiding modality and model selection. These ratings are qualitative syntheses of the representative mechanisms and empirical findings discussed in Secs. 4.1-4.4, rather than absolute benchmark scores. **Scalability** reflects the method's ability to handle large-scale behavioral data, a primary deployment constraint given that real-world systems routinely involve millions of users and billions of interactions; the assessment is informed by the efficiency of tree ensembles (Chen & Guestrin, 2016; Ke et al., 2017), sequence Transformers (Kang & McAuley, 2018; Sun et al., 2019b), temporal graph systems (Zhou et al., 2022; 2023), and lightweight CTDG encoders (Cong et al., 2023). **Label Efficiency** captures how well a method can learn useful representations when labels are scarce, drawing on self-supervised tabular learning (Yoon et al., 2020; Bahri et al., 2022; Kim et al., 2024), sequence pre-training (Sun et al., 2019b; Li et al., 2020c; Ansari et al., 2024; Das et al., 2024), and temporal link prediction in CTDGs (Rossi et al., 2020; Xu et al., 2020; Yu et al., 2023b). **Temporal Precision** measures the granularity at which a method models the timing of behavioral events: tabular methods typically discard event order, sequence models preserve order and intervals, TPPs explicitly model event timing (Du et al., 2016; Mei & Eisner, 2017; Zuo et al., 2020), and CTDG methods retain timestamped interactions (Rossi et al., 2020; Xu et al., 2020). **Interpretability** reflects whether the representation or decision process can be inspected, with tree-based models, symbolic sequence models, and some TPP-based methods offering clearer mechanisms than deep graph or LLM-based encoders (Prokhorenkova et al., 2018; Grinsztajn et al., 2022; Yap et al., 2012). Finally, **Cold-start Handling** assesses whether a method can support new users, items, patients, nodes, or scenarios with little behavioral history. This dimension favors methods that exploit semantic priors, side information, augmentation, or transferable pre-training, while purely ID-based supervised sequence and graph models usually remain weak when historical interactions are absent (Bao et al., 2023; Gao et al., 2023a; Han et al., 2024; Yoon et al., 2020; Bahri et al., 2022; Kim et al., 2024).

Table 7: Cross-modal comparison of method families along practical axes. Ratings are relative within each row: **H** = High, **M** = Medium, **L** = Low. The rating rationale for this table is shown in Appendix A.

| Method Family | Scalability | Label Efficiency | Temporal Precision | Interpret-ability | Cold-start Handling |
|---|---|---|---|---|---|
| *Tabular Data (Sec. 4.1)* | | | | | |
| Machine Learning-based | H | M | L | H | L |
| DL-based: Supervised-only | M | L | L | M | L |
| DL-based: Supervised-plus | M | M–H | L | M | M |
| LLM-driven | L | H | L | M | H |
| *Event Sequences (Sec. 4.2)* | | | | | |
| Symbolic: Sequential Pattern Mining | M | L | L | H | L |
| Symbolic: Markov Chain Models | H | M | L | H | L |
| DL-based: Supervised (RNN/CNN/Transformer) | H | L | M | L | L |
| DL-based: Self-supervised (Autoregressive) | H | M–H | M | L | L–M |
| DL-based: Self-supervised (Masked Reconstruction) | H | H | M | L | M |
| DL-based: Self-supervised (TPP-based) | M | M–H | H | M | L |
| DL-based: Self-supervised (Contrastive) | M–H | H | M | L | M |
| DL-based: Self-supervised (Adversarial) | M | M–H | M | L | L–M |
| *Dynamic Graphs (Sec. 4.3)* | | | | | |
| DTDG: Static Embed. + Temporal Alignment | M–H | M | L | M | L |
| DTDG: GNN+RNN-based | M | M | M | L | L |
| DTDG: Attention-based | M–H | M | M | L–M | L |
| CTDG: RNN-based | M | M | H | L | L |
| CTDG: TPP-based | M | M–H | H | M | L |
| CTDG: Random Walk-based | M–H | M–H | M–H | M | M |
| CTDG: Attention/Time Encoding-based | M | M–H | H | L–M | L–M |
| CTDG: MLP-based | H | M | M | M | L |
| *Textual Data (Sec. 4.4)* | | | | | |
| LLMs for Augmenting: Feature Augmentation | M | H | L | M–H | H |
| LLMs for Augmenting: Behavior Augmentation | M | H | L–M | M | H |
| LLMs for Encoding: Representation Enhancement | L–M | H | L–M | L–M | M–H |
| LLMs for Encoding: Cross-modality Unification | L–M | H | L–M | L | H |
| LLMs for Controlling | L | H | L | M–H | H |

# 7 Key Modules

Although foundational modeling approaches such as tabular modeling, sequential modeling, and dynamic graph modeling have achieved significant progress in behavioral data analysis, the inherent complexity and diversity of behavioral data often render these basic methods insufficient for fully capturing critical factors such as temporal dependencies, structural patterns, and contextual information. To address these limitations, researchers have proposed a variety of enhanced modeling strategies designed to further improve the representational capacity and predictive performance of behavior models. These enhancement methods are not standalone modeling paradigms, but rather modular components or mechanisms that can be flexibly integrated into different types of models. They include, but are not limited to, time encoding, position encoding, context modeling, structure-aware mechanisms, contrastive learning, and pre-training. These strategies have demonstrated strong generalizability and effectiveness across a wide range of behavioral modeling tasks in recent years, becoming key drivers in the advancement of behavioral data modeling.

This section provides a systematic overview of these enhanced modeling methods, analyzing their underlying design principles, applicable scenarios, and integration approaches with primary modeling techniques.

## 7.1 Time Encoding

In behavioral data modeling, temporal information often contains critical dynamic patterns such as periodicity, time-delay dependencies, and triggering sequences. Whether in user behavior, medical events, or

financial transactions, the timing or intervals of actions play a crucial role in understanding their under-
lying evolution. Early studies typically relied on handcrafted temporal features specifically designed for
downstream tasks (Choi et al., 2016b; Baytas et al., 2017; Kwon et al., 2019). These methods involved
decomposing timestamps into components (e.g., year, month, day), assigning embeddings to each compo-
nent, and summing them to form the final time representation. While effective in certain scenarios, such
approaches were labor-intensive, domain-specific, and limited in their ability to capture only predefined
temporal patterns (Kazemi et al., 2019).

With the rapid development of attention mechanisms, researchers began exploring more flexible ways to
model temporal information, particularly in capturing long-term dependencies and dynamic changes. This
led to the emergence of Functional Time Encoding (FTE) methods, which aimed to provide time rep-
resentations compatible with self-attention architectures. Representative works include Functional Time
Representation (Xu et al., 2019) and Time2Vec (Kazemi et al., 2019). These methods apply multiple linear
transformations to the raw time input, followed by pre-defined nonlinear functions (e.g., trigonometric func-
tions) to generate multi-dimensional time embeddings. Their ability to effectively encode periodic patterns
made them widely adopted in dynamic graph representation learning and time series modeling (Yu et al.,
2023b; Chen et al., 2025b).

Despite the success of functional time encoding in capturing periodic behaviors, these methods inherently rely
on strong inductive biases rooted in predefined periodic assumptions (Li et al., 2018; Xu et al., 2019; Kazemi
et al., 2019). As a result, they often struggle to model more complex temporal patterns such as non-periodic
and mixed behaviors. To address these limitations, recent studies proposed more expressive approaches like
LeTE (Chen et al., 2025b), which aim to capture a broader range of temporal dynamics. LeTE introduces
neural transformations that allow the model to automatically learn meaningful temporal features directly
from raw time values, without relying on handcrafted components or fixed periodic functions. This enables
the model to better adapt to the diverse and complex nature of real-world temporal behaviors. In contrast,
Chung et al. propose using simpler linear time encodings and show that self-attention can extract temporal
patterns from these features while reducing model complexity (Chung et al., 2025).

The evolution of time encoding techniques has significantly advanced the field of behavioral data analysis.
In many real-world scenarios, such as user interactions, clinical events, or financial activities, behaviors are
not only determined by what happens, but also by when and in what order they occur. Accurate temporal
representations help uncover latent behavioral patterns such as periodicity, recency effects, burstiness, or
long-term dependencies. Modern time encoding methods, especially those based on neural transformations,
enable models to better capture these complex temporal dynamics without relying on rigid assumptions.
This has led to more faithful representations of behavioral sequences, improved interpretability of behavioral
trends, and enhanced generalization across diverse behavioral contexts. As research continues, there is a
growing need for time encoding techniques that are not only expressive but also tailored to the unique chal-
lenges of behavioral data, such as irregular sampling, multi-scale temporal patterns, and context-dependent
timing effects.

## 7.2  Position Encoding

In early sequence modeling, positional information was either implicitly captured through the use of se-
quential architectures such as RNNs, or explicitly introduced using integer-based position indices. These
approaches allowed models to process tokens in a temporal order, thereby encoding positional dependencies
through recurrence or positional tags. However, the emergence of attention-based models—particularly the
Transformer—marked a paradigm shift, as these models lack an inherent notion of sequence order. This
limitation sparked a surge of research interest in position encoding, aiming to explicitly inject positional
information into attention mechanisms. Among the proposed solutions, two main categories have emerged:
absolute position encoding, which assigns unique representations to fixed positions, and relative position en-
coding, which focuses on modeling the distance or offset between tokens. The development of more effective
position encoding schemes has since become a central theme in advancing the performance and generalization
of attention-based models.

### 7.2.1 Absolute Position Encoding

Along with the attention mechanism, the sinusoidal position encoding is proposed (Vaswani et al., 2017), a form of absolute position encoding that maps each position to a unique, analytically defined vector. While this encoding is theoretically extrapolatable to arbitrarily long sequences, subsequent studies (Dai et al., 2019; Neishi & Yoshinaga, 2019) revealed practical limitations. Specifically, since the model is trained on sequences of limited length, the attention weights may fail to generalize to longer sequences during inference, leading to unstable outputs. This highlights a fundamental limitation of absolute position encoding in terms of extrapolation capacity in real-world applications.

In response to the extrapolation challenges of absolute position encoding, researchers have proposed various enhancement strategies. Ideally, position encodings should possess specific properties; notably, shift invariance—the property that a function's output remains unchanged when its input is uniformly shifted—was emphasized in (Wang et al., 2020) as a key factor for achieving robust extrapolation. a method involving random positional shifts during training was introduced by (Kiyono et al., 2021), where the original position index $pos$ with $pos + k$, where $k$ is sampled from a discrete uniform distribution $u(0, K)$. This approach discourages the model from relying on absolute positions and instead promotes learning of relative positional patterns. Building on this idea, additional perturbations, including global shifts, local jitters, and global scaling, were proposed in (Likhomanenko et al., 2021) to model position encodings as continuous signals with controlled variability to improve generalization. Additionally, representing word embeddings as continuous functions of position has been suggested (Wang et al., 2020), allowing word representations to evolve smoothly with increasing position indices. Finally, a dynamic systems perspective was explored by (Liu et al., 2023a), directly modeling the temporal dynamics between positional representations. Collectively, these efforts represent significant progress in enhancing absolute position encoding, aiming to improve its generalization and robustness in long-sequence modeling tasks.

### 7.2.2 Relative Position Encoding

While absolute position encodings provide a straightforward way to inject position information into Transformer models, they are inherently limited in their ability to generalize to sequences of unseen lengths or shifted patterns. More importantly, in many sequence modeling tasks, it is not the absolute position of a token that matters most, but rather its relative position with respect to other tokens. For instance, syntactic dependencies or temporal correlations are often defined by how far apart two elements are, rather than their fixed positions in the sequence (Huang et al., 2020b; Sinha et al., 2022).

Motivated by this observation, relative position encoding was first formalized in (Shaw et al., 2018) by explicitly incorporating pairwise positional relationships into the self-attention mechanism. This approach enhanced the model's ability to capture local structures and long-range dependencies.

Building upon this foundation, subsequent works extended relative position encoding in various directions. For instance, Transformer-XL (Dai et al., 2019) introduced a segment-level recurrence mechanism to enable longer context modeling during inference. Similarly, a simplified form of relative position encoding was adopted in T5 (Raffel et al., 2020), where each relative position is mapped to a learnable scalar that is directly added to the attention logits, improving computational efficiency. The concept was further advanced by DeBERTa (He et al., 2021), which decoupled content and positional information and applied the embeddings in a Transformer-XL-style architecture. Additionally, a log-bucket relative position bucketing strategy has been utilized (Chi et al., 2022); this method discriminates nearby positions more finely while coarsely representing distant ones. Such a logarithmic approach reduces the number of parameters and improves extrapolation capacity.

To further enhance length generalization, strategies such as adding bias terms to the attention scores (Wennberg & Henter, 2021) and incorporating actual distance information (Wu et al., 2021) have been explored. Notably, ALiBi (Press et al., 2022) emerged as the first relative position encoding method specifically designed for length extrapolation. It combines simplicity with strong performance by applying fixed, position-dependent biases to the attention weights.

Inspired by sinusoidal encodings, Rotary Position Embedding (RoPE) is introduced (Su et al., 2024a), which encodes relative position by rotating the query and key vectors in a position-dependent manner, rather than adding explicit position vectors. This continuous and extrapolatable encoding has been widely adopted in large-scale language models such as LLaMA and ChatGLM, making it one of the most prominent relative position encoding techniques today.

In conclusion, enhancing temporal and position encoding is essential for effective behavioral data analysis. Behavioral sequences often involve complex temporal patterns where the relative timing and order of actions are more informative than their absolute positions. Advanced encoding strategies—such as relative position embeddings and time-aware attention—enable models to better capture dependencies, generalize across varying sequence lengths, and improve both performance and interpretability. These improvements highlight the critical role of encoding-aware designs in modeling real-world behavioral data.

### 7.3 Summary and Takeaways

The discussion about key modules also reveals that time and position encoding are not isolated technical details, but recurring design choices that distinguish stronger behavioral representations across modalities. In event sequences, they determine whether irregular intervals and long histories remain usable; in dynamic graphs, they regulate how event order and temporal distance affect message passing; and in textual or LLM-based settings, they shape the model's ability to align semantics with behavioral chronology. The broader lesson is that many empirical gains attributed to larger architectures are in fact mediated by better encoding of temporal and positional structure.

## 8 Datasets and Benchmarks

High-quality datasets play a crucial role in advancing research on behavioral data representation learning, as they provide a solid foundation for model development and evaluation. Corresponding benchmarks further enable fair and consistent comparisons across different methods. In this section, we discuss commonly used behavioral data datasets along with benchmarks, and present relevant statistical information for each.

### 8.1 Tabular Data

Table 8: Behavioral datasets for tabular representation learning

| Dataset | # Numeric | # Categories | # Samples | # Tasks |
|---|---|---|---|---|
| California Housing (CA) | 8 | 0 | 20,640 | Regression |
| Adult (AD) | 6 | 8 | 48,842 | Classification |
| Helena (HE) | 27 | 0 | 65,196 | Classification |
| Jannis (JA) | 54 | 0 | 83,733 | Classification |
| Higgs (HI) | 28 | 0 | 1,000,000 | Classification |
| ALOI(AL) | 128 | 0 | 108,000 | Classification |
| Epsilon (EP) | 2,000 | 0 | 500,000 | Classification |
| Year (YE) | 90 | 0 | 515,345 | Regression |
| Covertype(CO) | 54 | 0 | 518,012 | Classification |
| Bank(BK) | 7 | 9 | 45,211 | Classification |
| Blastchar (BC) | 3 | 17 | 7,043 | Classification |
| Shoppers (SH) | 4 | 14 | 12,330 | Classification |
| Volkert (VO) | 180 | 1 | 58,310 | Classification |
| Income (IC) | 6 | 8 | 32,561 | Classification |
| Yahoo (YA) | 699 | 0 | 709,877 | Regression |
| Microsoft (MI) | 136 | 0 | 1,200,192 | Regression |

Table 9: Behavioral benchmarks for tabular representation learning.

| Benchmark | Paper | Repository | Year |
|---|---|---|---|
| OpenML-CC18 | (Bischl et al., 2021) | OpenML | 2017 |
| Regularized DNNs | (Kadra et al., 2021) | GitHub | 2021 |
| TabularBench | (Grinsztajn et al., 2022) | GitHub | 2022 |
| TabZilla | (McElfresh et al., 2023) | GitHub | 2023 |
| OpenTabs | (Ye et al., 2024a) | GitHub | 2024 |
| TALENT | (Ye et al., 2024b) | GitHub | 2024 |
| TDBench | (Kang et al., 2025) | GitHub | 2025 |

**Datasets**  To support research on behavior modeling in tabular data formats, both academia and industry have developed a wide range of representative datasets that are widely used for evaluating and comparing model performance. Among them, datasets such as California Housing (CA), Adult (AD), Helena (HE), and Jannis (JA) have been repeatedly adopted in numerous studies (Wu et al., 2024a; Na et al., 2024; Ahamed & Cheng, 2024; Xu et al., 2024; Joseph & Raj, 2022), and have gradually become standard benchmarks for tabular behavior modeling tasks. These datasets span a variety of prediction objectives—from regression to multi-class classification—and cover diverse application domains, including financial risk assessment, socioeconomic analysis, and local lifestyle services.

Some of these datasets originate from public data science competitions, such as those hosted on the UCI Machine Learning Repository and Kaggle, ensuring a high degree of standardization and broad generality. This contributes to the reproducibility of research and facilitates fair comparison across methods. Moreover, behaviors in these datasets are typically recorded in a structured form—such as samples composed of user attributes and behavioral outcomes—naturally framing them as classification or regression problems, and providing a solid foundation for building more complex behavior modeling frameworks. We summarize the most commonly used datasets in Tab. 8.

**Benchmarks**  To systematically evaluate the performance of various behavior modeling approaches, researchers have proposed a range of benchmark settings to standardize experimental protocols and enable fair comparisons across models. In the context of behavior modeling with tabular data, most datasets are sourced from the open platform OpenML, and many well-established benchmarks in this domain have been built upon it. Depending on their specific evaluation goals, researchers selectively construct benchmark suites by sampling from the OpenML repository.

One of the most widely adopted benchmarks is OpenML-CC18 (Bischl et al., 2021), which was constructed using eight filtering criteria and includes 72 high-quality classification datasets. This benchmark has become a standard reference for evaluating machine learning models on tabular data. In addition, the influential work (Grinsztajn et al., 2022) selected 45 representative datasets from OpenML based on criteria such as heterogeneity, feature richness, data realism, and task difficulty, creating a challenging and insightful benchmark for comparing deep learning and traditional models.

Building upon this, TaZilla (McElfresh et al., 2023) conducted a comprehensive analysis of 176 classification datasets from OpenML. It introduced additional filtering criteria that consider not only task diversity but also practical constraints such as runtime. Furthermore, it ensured broad coverage of datasets commonly used in popular studies, contributing to a more representative and reproducible benchmark foundation. Alongside TDColER (Kang et al., 2025), the authors also propose TDBench, a tabular distillation benchmark covering 23 datasets, 7 model classes, and 11 distillation strategies. We summarize the collected benchmarks in Tab. 9.

## 8.2  Event Sequences

**Datasets**  The datasets utilized in event sequence modeling encompass a broad spectrum of domains, such as E-commerce, Local Services, Media Content, and specialized verticals including Healthcare, Art,

Table 10: Sequential behavioral datasets

| Dataset | # Users | # Items/Businesses | # Samples | # Domain |
|---|---|---|---|---|
| Amazon Product Reviews 2023 | 54.51M | 48.19M | 571.54M | E-commerce |
| Amazon Q&A | – | 191K | 1.48M Q, 4.02M A | E-commerce |
| ModCloth Marketing Bias | 44.78K | 1.02K | 99.89K | E-commerce |
| Google Local Reviews 2021 | 113.64M | 4.96M | 666.32M | Local services |
| Google Restaurants | 1.01M | 65K | 1.77M | Local services |
| Twitch | 15.5M | 465K | 124M | Media content |
| Food.com Recipe & Review | 226.57K | 231.64K | 1,13M | Media content |
| EndoMondo Fitness Tracking Data | 1.10K | – | 253.02K | Healthcare |
| Behance Community Art Data | 63.50K | 178.79K | 1M | Art |
| Taobao UserBehavior | 987.99K | 4.16M | 100.15M | E-commerce |
| MovieLens 32M | 200.95K | 87.59K | 32.00M Ratings | Media content |
| Steam Video Game and Bundle Data | 2.57M | 15.47K Items, 615 Bundles | 7.79M | Gaming |

Table 11: Behavioral sequential representation learning benchmarks.

| Benchmark | Paper | Repository | Year |
|---|---|---|---|
| GRU | (Hidasi et al., 2016) | GitHub | 2015 |
| BERT | (Devlin et al., 2019) | GitHub | 2017 |
| CPC | (van den Oord et al., 2018) | GitHub | 2018 |
| Transformer | (Vaswani et al., 2017) | GitHub | 2017 |
| PrimeNet | (Chowdhury et al., 2023) | GitHub | 2023 |
| RMTPP | (Du et al., 2016) | GitHub | 2016 |

and Gaming. In these datasets, an event is typically defined as a timestamped user interaction, such as a purchase, rating, review, view, or content engagement.

A significant portion of research focuses on the E-commerce domain. Prominent examples include the Amazon Product Reviews 2023 (Hou et al., 2024) and Amazon Q&A (Wan & McAuley, 2016; McAuley & Yang, 2016), which capture rich user feedback and interaction histories. In the fashion and retail sector, ModCloth Marketing Bias (Wan et al., 2020) and the widely used Taobao UserBehavior (Zhu et al., 2018; 2019b; Zhuo et al., 2020) are frequently employed to benchmark models on purchasing and browsing sequences.

In the realm of Local Services, event sequences are extensively studied to understand user preferences for physical establishments. Datasets such as Google Local Reviews 2021 (Li et al., 2022b; Yan et al., 2023) and Google Restaurants (Yan et al., 2023) offer valuable insights into location-based user activities and service interactions. For Media Content, large-scale datasets are used to evaluate recommendation and sequence prediction capabilities across different media types. This category includes Twitch (Rappaz et al., 2021) for live streaming interactions, the massive MovieLens 32M (Harper & Konstan, 2015) for movie ratings, and Food.com Recipe & Review (Majumder et al., 2019) for culinary content exploration.

Furthermore, several datasets target specific vertical domains. These include EndoMondo Fitness Tracking Data (Ni et al., 2019) for Healthcare, Behance Community Art Data (He et al., 2016) for Art appreciation, and Steam Video Game and Bundle Data (Kang & McAuley, 2018; Wan & McAuley, 2018; Pathak et al., 2017) for Gaming behavior analysis. We summarize the statistics and characteristics of these event sequence datasets in Tab. 10.

**Benchmarks** To evaluate the performance of various approaches, a set of classical and widely effective sequential modeling methods have been selected as benchmarks. These methods are representative works within the categories of time series modeling discussed previously. They are general-purpose and applicable to diverse downstream domains, such as item recommendation, financial forecasting, and healthcare prediction. While domain-specific methods exist, they often build upon these general paradigms (e.g., GRU4Rec

Table 12: Behavioral datasets for dynamic graph representation learning.

| | Dataset | # Nodes | # Edges | # Snapshots / Unique Steps | Domain | Time Granularity |
|---|---|---|---|---|---|---|
| DTDG | BSI-ZK | 1,744,561 | 56,194,191 | 257 | Finance | Days |
| | BSI-SVT | 89,564 | 190,133 | 49 | Finance | Weeks |
| | Bitcoin-OTC | 5,881 | 35,592 | 279 | Finance | Weeks |
| | Bitcoin-Alpha | 3,783 | 24,186 | 274 | Finance | Weeks |
| | Reddit-Title | 54,075 | 571,927 | 178 | Interaction | Weeks |
| | Reddit-Body | 35,776 | 286,561 | 178 | Interaction | Weeks |
| | AS-733 | 7,716 | 11,965,533 | 733 | Traffic | Days |
| | UCI-Message | 1,899 | 59,835 | 49 | Social | Weeks |
| | Flights | 13,169 | 1,927,145 | 122 | Transport | Days |
| | Can. Parl. | 734 | 74,478 | 14 | Politics | Years |
| | US Legis. | 225 | 60,396 | 12 | Politics | Congresses |
| | UN Trade | 255 | 507,497 | 32 | Finance | Years |
| | UN Vote | 201 | 1,035,742 | 72 | Politics | Years |
| | Contact | 692 | 2,426,279 | 8,064 | Proximity | 5 Minutes |
| CTDG | Wikipedia | 9,227 | 157,474 | 152,757 | Social | Unix timestamps |
| | Reddit | 10,984 | 672,447 | 669,065 | Social | Unix timestamps |
| | MOOC | 7,144 | 411,749 | 345,600 | Interaction | Unix timestamps |
| | LastFM | 1,980 | 1,293,103 | 1,283,614 | Interaction | Unix timestamps |
| | Enron | 184 | 125,235 | 22,632 | Social | Unix timestamps |
| | Social Evo. | 74 | 2,099,519 | 565,932 | Proximity | Unix timestamps |
| | ML25M | 221,588 | 25,000,095 | 20,115,267 | Interactions | Unix timestamps |
| | DGraphFin | 4,889,537 | 4,300,999 | 821 | Finance | Unix timestamps |
| | Taobao | 5,149,747 | 100,135,088 | 815,859 | E-commerce | Unix timestamps |
| | tgbl-review-v2 | 352,637 | 4,873,540 | 6,865 | E-commerce | Unix timestamps |
| | tgbl-coin | 638,486 | 22,809,486 | 1,295,720 | Finance | Unix timestamps |
| | tgbl-comment | 994,790 | 44,314,507 | 30,998,030 | Interaction | Unix timestamps |
| | tgbl-flight | 18,143 | 67,169,570 | 1,385 | Transport | Unix timestamps |

in recommendation is based on GRU, BEHRT in healthcare on BERT, and player2vec in gaming on the Transformer architecture). These general benchmark methods are summarized in Tab. 11.

### 8.3 Dynamic Graphs

**Datasets** The datasets used in dynamic graph representation learning span a wide range of domains, including finance, social media, and e-commerce, providing researchers with diverse resources to study behavioral data across different application scenarios. Based on the intrinsic characteristics of the datasets, we categorize them into DTDG and CTDG datasets. As discussed in Sec. 4.3.2, CTDG representation learning models can also handle DTDG data. For example, in DyGFormer (Yu et al., 2023b), the authors evaluate their model using several DTDG datasets, such as UCI-Message, Flights, and Can.Parl.

Commonly used DTDG datasets include BSI-ZK, BSI-SVT, Bitcoin-OTC, Bitcoin-Alpha, Reddit-Title, Reddit-Body, and AS-733 (You et al., 2022). For CTDG datasets, Wikipedia, Reddit, MOOC, and LastFM are among the most widely adopted. These four datasets are originally introduced in the JODIE paper (Kumar et al., 2019) and have since been extensively used in subsequent research. The authors of JODIE continue to maintain and update the Stanford Network Analysis Project (SNAP)[2], which hosts many other dynamic graph datasets.

---

[2]https://snap.stanford.edu/index.html

Table 13: Benchmarks for dynamic graph representation learning.

| Benchmark | Paper | Repository | Specialize | Year |
|---|---|---|---|---|
| PyG-Temporal | (Rozemberczki et al., 2021) | GitHub | DTDG | 2021 |
| TGL | (Zhou et al., 2022) | GitHub | Large-scale CTDG | 2022 |
| SPEED | (Chen et al., 2023e) | GitHub | Large-scale CTDG | 2023 |
| DYGL | (Ma et al., 2024b) | GitHub | DTDG and CTDG | 2023 |
| DyGLib | (Yu et al., 2023b) | GitHub | CTDG | 2024 |
| TGB | (Huang et al., 2023b; Gastinger et al., 2024) | GitHub | CTDG | 2024 |
| BenchTemp | (Huang et al., 2024a) | GitHub | CTDG | 2024 |
| DGB | (Gravina & Bacciu, 2024) | GitHub | DTDG and CTDG | 2024 |
| BenchTGNN | (Yang et al., 2024b) | GitHub | CTDG | 2024 |
| TGX | (Shirzadkhani et al., 2024) | GitHub | CTDG | 2024 |
| DGNN | (Feng et al., 2025) | GitHub | DTDG and CTDG | 2024 |
| UTG | (Huang et al., 2025b) | GitHub | DTDG and CTDG | 2024 |

Since the publication of DyGFormer, broader evaluations on CTDG representation learning models have emerged. For instance, in addition to several DTDG datasets, DyGFormer also uses Enron and Social Evo. datasets for evaluation. Large-scale datasets such as DGraphFin (Huang et al., 2022), ML25M (Harper & Konstan, 2015), and Taobao (Zhuo et al., 2020) are often used to benchmark scalable training frameworks. Recently, the release of the Temporal Graph Benchmark (TGB) (Huang et al., 2023b; Gastinger et al., 2024) has introduced a series of larger-scale datasets, including tgbl-review-v2, tgbl-coin, tgbl-comment and tgbl-flight, which have further enriched the evaluation landscape. We summarize the commonly used dynamic graph datasets in Tab. 12.

**Benchmarks** The early stage of dynamic graph benchmark development, exemplified by PyTorch Geometric Temporal (PyG-Temporal) (Rozemberczki et al., 2021), focuses primarily on DTDGs. This early framework provides essential support for snapshot-based modeling but remains limited in scalability and task diversity.

With the increasing interest in modeling fine-grained temporal dynamics, the next phase sees the emergence of CTDG benchmarks. TGL (Zhou et al., 2022) and SPEED (Chen et al., 2023e) emphasize large-scale learning and efficient training on billion-edge graphs, marking a shift toward practical deployment scenarios and industrial applicability. Recent years have witnessed rapid expansion of the ecosystem. The Dynamic Graph Library (DyGLib) (Yu et al., 2023b), released alongside DyGFormer, as well as the Temporal Graph Benchmark (TGB) (Huang et al., 2023b) and its extended version TGB 2.0 (Gastinger et al., 2024), set new standards by offering large-scale, domain-diverse CTDG datasets alongside automated evaluation pipelines and leaderboards, significantly improving reproducibility and comparability. Parallel efforts—such as BenchTemp (Huang et al., 2024a), BenchTGNN (Yang et al., 2024b), Temporal Graph Analysis with TGX (TGX) (Shirzadkhani et al., 2024), DGNN (Feng et al., 2025) have further enriched the landscape with modular toolkits that cover an even wider spectrum of models.

In addition to CTDG-focused efforts, several benchmark libraries aim to bridge the DTDG and CTDG paradigms. DYGL-library (DYGL) (Ma et al., 2024b) and the Dynamic Graph Benchmark (DGB) (Gravina & Bacciu, 2024) support both DTDG and CTDG models and tasks under a unified framework. Similarly, Unifying Temporal Graph (UTG) (Huang et al., 2025b) attempts to provide a general infrastructure that accommodates both DTDG and CTDG representation learning methods, further promoting methodological consistency across different temporal settings. We summarize representative benchmarks in Tab. 13.

## 8.4 Textual Data

**Datasets** Finally, we introduce the *language-based behavioral* datasets. These datasets typically consist of entities and the behavioral interactions occurring among them. Such interactions are often timestamped, and both entities and interactions are associated with rich textual attributes. We summarize these datasets

Table 14: Datasets for textual behavioral data.

| Datasets | # Entity | # Behavior | # Behavior Category | # Timestamp | Domain |
|---|---|---|---|---|---|
| Enron | 42,711 | 797,907 | 10 | 1,006 | E-mail |
| GDELT | 6,786 | 1,339,245 | 237 | 2,591 | Knowledge graph |
| ICEWS1819 | 31,796 | 1,100,071 | 266 | 730 | Knowledge graph |
| Stack elec | 397,702 | 1,262,225 | 2 | 5,224 | Multi-round dialogue |
| Stack ubuntu | 674,248 | 1,497,006 | 2 | 4,972 | Multi-round dialogue |
| Googlemap_CT | 111,168 | 1,380,623 | 5 | 55,521 | E-commerce |
| Amazon movies | 293,566 | 3,217,324 | 5 | 7,287 | E-commerce |
| Yelp | 2,138,242 | 6,990,189 | 5 | 6,036 | E-commerce |

in Tab. 14, which collectively span a broad spectrum of domains, ranging from *E-mail communication* and *Knowledge graphs* to *Multi-round dialogues* and large-scale *E-commerce platforms*. For example, the Enron dataset models organizational email exchanges: entities correspond to individual users annotated with descriptive metadata such as email addresses, while interactions represent message transmissions that are further accompanied by the full textual content of the emails. These datasets not only mirror real-world communication and decision-making processes, but also provide fertile ground for advancing research at the nexus of natural language processing and behavioral modeling. They offer a robust empirical foundation for harnessing LLMs to capture fine-grained behavioral regularities and to drive performance gains across a diverse set of downstream applications.

**Benchmarks** The benchmarking practice of applying LLMs to behavioral data is still in its early stages. For instance, DTGB (Zhang et al., 2024b) has recently introduced a benchmark that models user behaviors through dynamic text-attributed graphs, thereby providing one of the first systematic attempts to bridge behavioral dynamics with textual semantics in an LLM-friendly manner. Nevertheless, beyond this effort, we find relatively few benchmarks or frameworks that explicitly focus on modeling language-based behavioral data with LLMs. One possible reason is the inherent complexity of such data: interactions are not only temporally ordered but also linguistically rich, which poses challenges in aligning heterogeneous modalities, designing scalable representations, and ensuring efficient training. Moreover, the absence of standardized datasets and evaluation protocols further hinders the community from conducting consistent and reproducible studies. This gap highlights a promising research direction for future work. Establishing comprehensive benchmarks, unified evaluation metrics, and diverse datasets would significantly accelerate progress, allowing LLMs to better capture the interplay between textual semantics and behavioral regularities.

### 8.5 Benchmark maturity and evaluation gaps

The benchmark landscape is highly uneven across modalities. Tabular learning already benefits from relatively mature suites built on OpenML-style protocols, and dynamic graph learning has recently converged toward more standardized toolkits and leaderboards such as TGB and DyGLib. By contrast, event-sequence evaluation remains more fragmented and application-specific, while textual behavioral modeling still lacks broadly adopted benchmark infrastructure. A more fundamental limitation is that current benchmarks are almost entirely modality-specific: they rarely allow the same underlying behavioral data to be modeled and evaluated as tabular features, sequences, graphs, or text under a unified protocol. As a result, many reported improvements are convincing within a modality, but much less informative for cross-modality model selection.

## 9 Comparison with Existing Surveys

Tab. 15 summarizes representative surveys adjacent to our work. Existing surveys generally remain limited along at least one important dimension. Some are modality-specific, focusing on only one form of behavioral data, such as tabular representation learning (Jiang et al., 2026), sequential modeling (Shchur et al., 2021; Zhou et al., 2025b), dynamic graph representation learning (Jiao et al., 2025; Feng et al., 2025), or textual

Table 15: Comparison with representative adjacent surveys. ∘ denotes partial coverage.

| Survey | Year | Main organizing idea | Coverage | | | | | | |
|---|---|---|---|---|---|---|---|---|---|
| | | | Tab. | Event Seq. | Dyn. Graph | Text/LLM | Cross-modal view | Tasks | Data/Bench. |
| *Neural Temporal Point Processes: A Review* (Shchur et al., 2021) | 2021 | Intensity / encoder / objective design | × | ✓ | × | × | × | ∘ | ∘ |
| *A Survey on User Behavior Modeling in Recommender Systems* (He et al., 2023) | 2023 | Behavior modeling paradigms in RecSys | × | ∘ | × | × | × | ✓ | ∘ |
| *Embedding in Recommender Systems: A Survey* (Zhao et al., 2023b) | 2023 | Matrix / sequential / graph embeddings | ∘ | ✓ | ∘ | ∘ | × | ✓ | ∘ |
| *User Modeling and User Profiling: A Comprehensive Survey* (Purificato et al., 2024) | 2024 | Historical evolution of user modeling | × | × | ∘ | ∘ | × | ✓ | × |
| *A Survey on Deep Tabular Learning* (Somvanshi et al., 2024) | 2024 | Historical evolution of tabular deep learning | ✓ | × | × | × | × | ✓ | ∘ |
| *A Survey on Temporal Interaction Graph Representation Learning* (Jiao et al., 2025) | 2025 | Structural / temporal / application taxonomy | × | ∘ | ✓ | × | × | ✓ | ✓ |
| *Using NLP to Analyse Text Data in Behavioural Science* (Feuerriegel et al., 2025) | 2025 | NLP methods for textual behavioral data | × | × | × | ✓ | × | ✓ | ∘ |
| *A Survey of Foundation Model-Powered Recommender Systems* (Huang et al., 2025a) | 2025 | Feature-based / generative / agentic FM RecSys | ∘ | ∘ | × | ✓ | × | ✓ | ∘ |
| *Advances in Temporal Point Processes: Bayesian, Neural, and LLM Approaches* (Zhou et al., 2025b) | 2025 | Bayesian → neural → LLM TPP taxonomy | × | ✓ | × | ∘ | × | ∘ | ✓ |
| *A Comprehensive Survey of Dynamic Graph Neural Networks* (Feng et al., 2025) | 2025 | DTDG / CTDG models and benchmarks | × | ∘ | ✓ | × | × | ✓ | ✓ |
| *Representation Learning for Tabular Data* (Jiang et al., 2026) | 2026 | Specialized / transferable / general tabular RL | ✓ | × | × | ∘ | ∘ | ∘ | ✓ |
| *A Survey on Sequential Recommendation* (Pan et al., 2026) | 2026 | Model families in sequential recommendation | × | ✓ | × | ∘ | ∘ | ✓ | ∘ |
| **Ours: A Survey on Behavioral Data Representation Learning** | 2026 | Unified modality taxonomy + within-modality sub-taxonomies | ✓ | ✓ | ✓ | ✓ | ✓ | ✓ | ✓ |

behavioral data analysis (Somvanshi et al., 2024; Feuerriegel et al., 2025). Others are organized around a particular application domain, especially recommender systems, including user behavior modeling (He et al., 2023), sequential recommendation (Pan et al., 2026), foundation-model-powered recommendation (Huang et al., 2025a), and recommendation embeddings (Zhao et al., 2023b). More broadly scoped surveys such as user modeling and profiling (Purificato et al., 2024) cover related themes, but do not organize the literature through a representation-learning-centered, cross-modality taxonomy. As a result, prior surveys typically provide either deep coverage of a single modality or broad discussion within a single application domain, but not a unified comparison across major behavioral data forms together with their associated tasks, datasets, and benchmarks.

In comparison, our survey adopts a unified perspective that covers four major behavioral data forms within a single framework. This organization reflects the observation that the same underlying behavioral phenomenon can be represented in different data forms depending on modeling choices, and that this representational choice fundamentally determines the appropriate learning paradigm. Beyond the top-level categorization, we further introduce finer-grained sub-taxonomies within each modality. The survey also systematically connects these method taxonomies to downstream application scenarios, shared modeling modules, and modality-specific datasets and benchmarks.

# 10 Conclusion and Future Directions

In conclusion, behavioral data representation learning plays a critical role in extracting meaningful patterns from complex, dynamic behaviors, supporting diverse real-world applications. This survey systematically categorizes existing methods, clarifies their functionalities and application scenarios, and provides an insightful framework to guide future developments.

Looking forward, several critical areas demand further research:

**(1) Principled multi-modal fusion.** The cross-modal comparison in Tab. 6 shows that every modality is lossy in a different way: tabular models discard order and relations, sequence models discard multi-entity structure, graph models discard non-relational attributes, and textual models discard numerical precision.

Because most real behavioral systems are inherently multi-modal, the central question is how to combine these representational stances without collapsing them into a single lossy format. Cross-modality LLM unification (Sec. 4.4) is one route, but aligning channels that differ in temporal resolution, sparsity, and behavioral meaning remains largely open.

**(2) Foundation models for behavior.** Modality-specific foundation models are emerging rapidly—TabPFN (Hollmann et al., 2025), TARTE (Kim et al., 2025), and the relational KumoRFM (Fey et al., 2025) for structured and multi-table records; Chronos (Ansari et al., 2024) and TimesFM (Das et al., 2024) for temporal data; and ULTRA (Galkin et al., 2024) for transferable knowledge-graph reasoning. The open problem is what actually transfers for *behavioral* data, where populations, schemas, and intent shift across domains, and whether a genuinely cross-modal behavioral foundation model is attainable.

**(3) Non-stationarity and distribution shift.** Behavior is non-stationary by nature: preferences, market regimes, health status, and social roles drift over time. Phase- and frequency-aware encoders (Mohapatra et al., 2025; Li et al., 2025b) are early steps, but representations that stay valid under concept drift—and evaluation protocols that explicitly test for it—are still scarce.

**(4) Cold-start and cross-domain transfer.** New users, items, accounts, and nodes with little history remain a pervasive failure mode for ID-based sequence and graph models. Semantic priors and side information help, but they assume that similar descriptions imply similar behavior, which does not always hold across domains.

**(5) Representation-aware evaluation.** Most methods are still judged by downstream accuracy alone. Behavioral analysis would benefit from evaluation that scores what a representation encodes and omits—temporal calibration, relational fidelity, robustness to augmentation, and interpretability—so that model selection reflects the modeling assumptions of Sec. 3.6 rather than leaderboard numbers alone.

**(6) Interpretability, fairness, and privacy.** Because behavioral data describe people, high-stakes deployment in finance, healthcare, and social systems demands representations that are auditable, fair across subpopulations, and learnable under privacy constraints (Kenfack et al., 2026; Afonja et al., 2025; Carranza et al., 2024).

# A   Rating rationale for Tab. 7

Building on the above criteria, the ratings reflect the characteristic strengths and weaknesses of each modality rather than isolated benchmark results.

For **tabular methods**, the main pattern is the trade-off between efficiency and representational richness. Traditional machine learning methods, especially GBDT-style models, are rated highly in scalability and interpretability because they train efficiently on structured records and expose split- or feature-based decision mechanisms (Chen & Guestrin, 2016; Ke et al., 2017; Prokhorenkova et al., 2018; Grinsztajn et al., 2022). Their low temporal-precision and cold-start ratings follow from the fixed-profile assumption: event order is usually aggregated away, and new entities lack reliable historical features. Supervised-only deep tabular models improve feature interaction modeling but remain label-dependent (Cheng et al., 2016; Guo et al., 2017; Lian et al., 2018; Arik & Pfister, 2021; Gorishniy et al., 2021). Supervised-plus and foundation-style tabular methods improve label efficiency through corruption, contrastive learning, row/table structure, or cross-table pre-training (Yoon et al., 2020; Bahri et al., 2022; Kim et al., 2024; Hollmann et al., 2025; Kim et al., 2025). LLM-driven tabular methods obtain stronger label efficiency and cold-start handling from semantic priors and task descriptions, but serialization, context length, and inference cost limit scalability, and fine-grained behavioral timing is not preserved by default (Hegselmann et al., 2023; Sui et al., 2024a; Han et al., 2025; Fang et al., 2024).

For **event-sequence methods**, the ratings depend on whether the model merely records local order or learns transferable temporal representations from unlabeled histories. Symbolic methods such as sequential pattern mining and Markov chains are relatively interpretable because they expose frequent motifs or transition probabilities, but they struggle with rare behaviors and new entities (Yap et al., 2012; Wang & Cao, 2020; Wang et al., 2019; Cheng et al., 2013; Zhang et al., 2014; Feng et al., 2015). Supervised neural encoders scale

well and preserve ordered histories, but their label efficiency is low when representations are shaped mainly by task-specific supervision (Tang & Wang, 2018; Yuan et al., 2019; Kang & McAuley, 2018; Li et al., 2020a). Self-supervised sequence models improve label efficiency by constructing supervision from the sequence itself: autoregressive models use next-event or next-token prediction (Skalski et al., 2023; Ansari et al., 2024; Das et al., 2024), masked reconstruction learns bidirectional behavioral context (Sun et al., 2019b; Li et al., 2020c; Hu et al., 2023b; Talukder et al., 2024), contrastive methods learn invariances across augmented views (Xie et al., 2022; Liu et al., 2021a; Qiu et al., 2022), and adversarial methods improve robustness through hard generated or discriminative alternatives (Bharadhwaj et al., 2018; Wang et al., 2017; Ni et al., 2023). TPP-based methods receive the highest temporal-precision rating because they explicitly optimize event-time likelihoods and model continuous-time intensities (Du et al., 2016; Mei & Eisner, 2017; Zhang et al., 2020; Zuo et al., 2020). Across this modality, cold-start handling remains limited unless side features or semantic information are introduced, because most methods still require at least some behavioral prefix.

For **dynamic graph methods**, the ratings reflect the additional benefit and cost of modeling evolving relational context. DTDG methods reuse static graph encoders over snapshots, which supports moderate scalability but limits temporal precision because event-level timing is aggregated into windows (Goyal et al., 2018; Zhou et al., 2018; Kazemi et al., 2020). GNN+RNN-based DTDG models jointly encode snapshot structure and temporal evolution, but recurrent propagation and stacked graph layers introduce scalability and over-smoothing concerns (Seo et al., 2018; Pareja et al., 2020; You et al., 2022). Attention-based DTDG methods improve long-range snapshot comparison and parallelism, while still inheriting the temporal coarseness of snapshot histories (Sankar et al., 2020; Chen et al., 2024d). CTDG methods are rated higher in temporal precision because they preserve timestamped interactions. RNN- and memory-based CTDG models update entity states at event time (Dai et al., 2016; Kumar et al., 2019; Rossi et al., 2020), TPP-based CTDG models represent event occurrence intensities (Zuo et al., 2018; Trivedi et al., 2019; Chen et al., 2024a), and attention/time-encoding methods retrieve temporal neighborhoods with explicit time encodings (Xu et al., 2020; Wang et al., 2021a; Alomrani et al., 2023; Yu et al., 2023b). Random-walk-based CTDG models receive moderate cold-start handling because time-respecting walks and anonymous encodings can sometimes exploit local neighborhood evidence for sparse or unseen nodes (Nguyen et al., 2018; Wang et al., 2021c; Souza et al., 2022). MLP-based CTDG models receive high scalability because they replace heavier recurrent or attention mechanisms with lightweight aggregation and temporal priors (Cong et al., 2023; Zou et al., 2024; Li et al., 2025b). The main limitation across graph methods is that richer relational context usually comes with higher sampling, memory, and training complexity.

For **textual and LLM-based methods**, the ratings are dominated by the benefits and costs of pre-trained semantic knowledge. Feature augmentation receives high label efficiency and cold-start handling because LLM-generated user insights, aspect labels, attributes, and compact summaries can enrich sparse behavioral records before downstream representation learning (Xi et al., 2024; Liu et al., 2025; Torbati et al., 2023). Behavior augmentation improves data coverage through synthetic logs, privacy-preserving traces, prompt-generated summaries, or condensed samples, although its temporal precision depends on whether generation preserves realistic event dynamics (Sidorenko et al., 2024; Carranza et al., 2024; Wu et al., 2025a; Chen et al., 2024c). Representation enhancement improves sparse-case transfer by encoding reviews, profiles, documents, or semantic tokens, but inference cost, opacity, and weak native timestamp handling keep scalability, interpretability, and temporal precision below specialized sequence or graph encoders (Bao et al., 2023; Qiu et al., 2021; Li et al., 2023b; Zheng et al., 2024; Wang et al., 2024h). Cross-modality unification methods use language as a shared semantic space to align user histories, item metadata, sensor signals, and graph/text attributes across domains, leading to high label efficiency and cold-start handling when side information is available (Han et al., 2024; Zhang et al., 2025a; Fu et al., 2025; Zhang et al., 2025b). LLM controlling methods also receive high label efficiency and cold-start handling because they exploit pre-trained reasoning, tool use, and multi-agent orchestration, but their scalability is limited by large-model inference and their temporal precision often depends on external sequence, graph, or retrieval modules rather than native event modeling (Gao et al., 2023a; Wang et al., 2024g; Yu et al., 2024; Piao et al., 2025).

Overall, Tab. 7 should be read as guidance for method selection under deployment constraints. Methods with strong inductive biases, such as tree ensembles, Markov models, TPPs, and temporal GNNs, tend to perform

well along the dimensions they are designed for, but transfer less easily to cold-start or cross-domain settings. Self-supervised and LLM-based methods improve label efficiency by exploiting unlabeled structure or pre-trained semantic knowledge, but may sacrifice scalability, temporal precision, or transparency. Dynamic graph methods provide the richest relational context, especially in CTDG settings, but incur greater memory, sampling, and training complexity.

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
