# OpenReview forum: "A Survey on Behavioral Data Representation Learning"
_TMLR — Decision pending for TMLR_

### Review · Reviewer_QAmS · 2026-03-11

**Summary Of Contributions:**

This paper surveys behavioral data representation learning methods, organizing them into a taxonomy based on four data modalities: tabular data, event sequences, dynamic graphs, and textual data (primarily via LLMs). Within each modality, the authors further classify methods by modeling paradigm — for instance, tabular methods are split into ML-based, deep learning-based (supervised-only vs. supervised-plus), and LLM-driven; dynamic graph methods are split by DTDG (static+temporal alignment, GNN+RNN, attention-based) and CTDG (RNN, TPP, random walk, attention/time encoding, MLP). The paper also reviews five application domains (e-commerce, healthcare, social media, finance, gaming), key modules (time encoding, position encoding), and datasets/benchmarks for each modality. Three contributions are claimed: (1) a novel modality-based taxonomy, (2) comprehensive method reviews within each category, and (3) analysis of emerging technologies (particularly LLMs) applied to behavioral data.

**Audience:**

Yes

**Audience Explanation:**

A survey unifying representation learning across tabular, sequential, graph, and textual behavioral data is timely and broadly relevant. The dynamic graph sections and the dataset/benchmark compilations are particularly valuable. Adding cross-modality comparison guidance (when to choose sequences vs. graphs vs. tabular features for the same task) would further increase the paper's practical impact.

**Broader Impact Concerns:**

None.

**Claims And Evidence:**

No

**Claims Explanation:**

- The paper is a survey, so the "claims" here are primarily about the comprehensiveness and organization of the taxonomy, and the accuracy of the method descriptions. However, there are multiple factual inconsistencies:
  - Page 15: "Mohapatra et al. (Butera et al., 2025)" — Butera et al. is by Butera/De Felice/Cini/Alippi, not Mohapatra.
  - "LLMREC" in Sec. 3.4.3 vs. "RecLLM" in Table 5 for the same Friedman et al. 2023 paper.
  - "LLM4ARec [98]" on p.31 uses numeric style inconsistent with the paper's author-year format.
  - Duplicate reference keys for identical papers (Cho et al. 2014a/b, Cheng et al. 2016a/b, Guo et al. 2017a/b).
  - **I did not thoroughly verify every citation among the ~300 references, as that is impractical to do manually. However, the number of errors found in a partial spot-check should raise a flag about the overall bibliography quality.**
- The paper presents a “novel taxonomy,” but the core top-level split by modality is fairly natural and not clearly justified against prior survey structures. The current paper does not establish what is genuinely new about the taxonomy beyond collecting material under four buckets.
- The paper reads more like a large literature inventory than a critical survey. Many subsections list methods and short descriptions, but there is limited discussion on what actually distinguishes successful approaches, what evaluation patterns stay across modalities, where the real empirical bottlenecks are etc.
- The textual data section (3.4) is also notably shallower than the other three modality sections.

**Requested Changes:**

- Most importantly, please conduct a full citation audit.  The number of errors found in spot checks raises concerns about the overall reliability of the bibliography.
- Add a proper survey methodology section. State literature sources, search terms or search process, inclusion / exclusion criteria, time cutoff.
- Clarify the novelty claim around the taxonomy. Either justify why this taxonomy is materially new relative to adjacent surveys, or soften the claim.
- Sharpen the "behavioral data" definition to distinguish this survey from a general representation learning survey.

---

> ### Author Response · Authors · 2026-05-11
> **Reply to Reviewer QAmS (Part 1)**
>
> Thank you very much for your valuable review and constructive comments on our manuscript. To better facilitate and move forward the review process, we are providing our response to the comments we have received so far in advance.
>
> At present, we have received comments from two reviewers and have revised the manuscript accordingly. The latest version of the manuscript is therefore based on the feedback provided by these two reviewers. Once we receive the comments from the third reviewer, we may further update the manuscript to address any additional suggestions or concerns.
>
> **[Regarding: Citation audit (Comments 1 and Requested Changes 1)]**
>
> We sincerely thank the reviewer for the careful spot-check of our bibliography, which prompted us to conduct a thorough and systematic audit of the entire reference list. We have addressed all identified issues and performed a comprehensive verification pass across the full manuscript:
>
> 1. Author-citation mismatches: We corrected all instances where in-text author names did not match the corresponding bibliography entries (e.g., the "Mohapatra et al." / Butera et al. mismatch has been fixed), and corrected citations that pointed to incorrect references.
> 2. Method name inconsistencies: We unified method names throughout the text and tables to ensure consistency (e.g., the LLMREC/RecLLM discrepancy has been resolved).
> 3. Missing references: We identified and supplemented references that were missing or incompletely cited.
> 4. Duplicate entries: We performed a full deduplication of the bibliography, removing all redundant entries that referred to the same paper.
> 5. Publication year verification: We verified the publication status of all cited works. For papers that have been formally published, we updated both the in-text citations and table entries to reflect the official publication year; for preprint-only works, we retained the preprint year.
> 6. Other citation issues: We also fixed several additional minor issues discovered during our audit.
>
> We would like to briefly explain the root cause of these issues. Due to the broad scope of this survey spanning four distinct data forms, our manuscript includes a substantial number of references, which were managed across multiple `.bib` files organized by topic. This multi-file management strategy, while necessary for organizational purposes, inadvertently introduced duplicates and inconsistencies across files. We have now consolidated and thoroughly verified the entire bibliography to eliminate such issues.
>
> We are grateful to the reviewer for raising this concern, and we welcome any further corrections should additional issues be identified.
>
> ---
>
> **[Regarding: Novel taxonomy (Comments 2 and Requested Changes 3)]**
>
> We agree that a modality-based top-level split can appear natural. Our intention in claiming novelty is not that prior work has never grouped methods by data form, but that existing adjacent surveys are typically confined to a single modality (e.g., tabular data, event sequences, temporal graphs, textual behavioral data) or a single application domain (e.g., recommender systems). In contrast, our survey treats behavioral data representation learning as a unified problem and organizes the literature across four major behavioral data forms, i.e., tabular, event sequences, dynamic graphs, and textual data, within one framework. Importantly, our taxonomy is not only a four-bucket collection: within each data form, we further provide finer-grained method classifications, and we connect them to downstream applications, shared modeling modules, and modality-specific datasets and benchmarks.
>
> We have revised the paper and added **Section 9 Comparison with Existing Surveys** and a dedicated comparison table against related surveys (**Table 15**), and soften the novelty wording where appropriate.
>
> ---
>
> **[Regarding: Descriptive  (Comments 3)]**
>
> We thank the reviewer for the constructive comment that the original manuscript was overly descriptive in several places. In the revision, we have strengthened the manuscript’s critical-survey component by adding synthesis paragraphs and takeaway statements across multiple sections to highlight key methodological insights and trade-offs. We also clarified the motivations behind different modeling paradigms by explicitly discussing the limitations of earlier approaches and how subsequent methods address them. We believe these changes make the survey more analytical and improve its value as a critical synthesis of the literature.
>
> Moreover, based on the feedback from both **you** and **Reviewer WxS8**, we have added a new section on ***Cross-Modal Synthesis and Comparative Analysis*** (**Section 6** in the new manuscript) to further sharpen and strengthen the survey.

---

> > ### Author Response · Authors · 2026-05-11
> > **Reply to Reviewer QAmS (Part 2)**
> >
> > **[Regarding: Textual data section (Comments 4)]**
> >
> > We would like to clarify that the perceived shallowness of Section 3.4 is due to the recency of this research direction. As noted in Table 5, most advances in leveraging textual behavioral data have emerged only after 2023. Compared to other modalities, which have been studied for over a decade with well-established methodological ecosystems, the body of work on textual behavioral data is relatively limited and rapidly evolving. This further demonstrates the timeliness of our survey.
> >
> > ---
> >
> > **[Regarding: Survey methodology section (Requested Changes 2)]**
> >
> > We have added a section about **Survey Methodology** (**Section 2** in the new manuscript).
> >
> > ---
> >
> > **[Regarding: "Behavioral data" definition (Requested Changes 4)]**
> >
> > The definition of "behavioral data" has been discussed in Section 3.2 (in the original manuscript). We further sharpen this definition by introducing three explicit criteria: behavioral data must be (1) **individual-generated** — produced by an entity (individual) making decisions or reactions, as opposed to passive environmental measurements; (2) **temporally situated** — occurring within a temporal context that captures the dynamics of behavior; and (3) **action-oriented** — reflecting observable actions, interactions, or decisions rather than static attributes alone.
> >
> > This distinguishes our scope from: (a) passive sensor data without agent involvement (e.g., weather time series), (b) static demographic profiles (which describe who someone is, not what they do), and (c) general-purpose representation learning on images/text without behavioral context.

---

### Review · Reviewer_WxS8 · 2026-04-09

**Summary Of Contributions:**

This paper gives a solid survey of behavioral data representation learning across different data types, including tabular data, event sequences, dynamic graphs, and text. The main idea is to put these pretty different lines of work under one umbrella and organize them with a modality-based taxonomy. The paper also goes beyond methods and includes downstream applications, datasets, benchmarks, and some discussion of newer directions like LLMs, multimodal learning, interpretability, and fairness. So overall, the contribution is mainly in collecting and organizing a large body of literature in one place.

**Audience:**

Yes

**Audience Explanation:**

Yes. I think at least some TMLR readers would care about this paper because it sits across several active areas, like tabular learning, sequence modeling, dynamic graphs, and language-based methods. It also covers applications, datasets, and benchmarks, so even as a survey, it could still be useful for people looking for a broad overview of a pretty scattered literature.

**Broader Impact Concerns:**

No major concerns beyond the usual ones for this area. Since the paper is about behavioral data, I think it would still be good to briefly mention privacy, bias/fairness, profiling, and possible misuse in sensitive domains like healthcare or finance. A short broader impact statement would be enough.

**Claims And Evidence:**

Yes

**Claims Explanation:**

My main reason for rating it this way is that the paper is clearly broad in scope and covers a lot of material, which is useful. I can see it being a good starting point for readers who want a big-picture overview of this area. I also liked that it does not just list models, but also tries to connect them to applications and datasets, which makes the survey more useful in practice.

That said, I think some of the framing is a bit stronger than what the paper fully delivers right now. The taxonomy is clear enough, but to me it feels more like a sensible way to organize the literature than a really new way of thinking about the field. Also, while the paper covers many topics, the synthesis across modalities is still not that deep. In a lot of places, it feels more like several parallel mini-surveys than one fully integrated story. I also think the paper is stronger on coverage than on critical analysis: it summarizes a lot of methods, but it is less strong when it comes to clearly comparing trade-offs, limitations, and what readers should take away across different method families.

**Requested Changes:**

1. The paper should do a better job explaining what is actually new about the proposed taxonomy. Right now, the main structure is a split by data modality, which is reasonable and easy to follow, but I am not fully convinced yet that this comes across as a genuinely new conceptual taxonomy rather than mostly a clean way of organizing the literature.

2. I would like to see stronger synthesis across modalities. At the moment, a lot of the paper reads like separate surveys for tabular data, event sequences, dynamic graphs, and text. That is still useful, but if the goal is to present a unified view of behavioral data representation learning, the paper should say more clearly what really ties these areas together.

3. The paper should strengthen the more critical part of the survey. It covers a lot of methods, but it is less strong on comparing trade-offs, limitations, and practical differences between method families. In several places, I wanted more discussion of what works when, where the gaps are, and what readers should actually take away.

Non-critical.
- It would help to add one or two cross-modal summary tables. For example, a table comparing modalities by representation type, typical tasks, supervision setting, evaluation setup, and common applications would make the paper much more useful.

- I think the positioning relative to existing surveys could be a bit sharper. The paper would be stronger if it more clearly explained what it adds beyond narrower surveys in tabular learning, dynamic graphs, sequential modeling, or text/LLM-based behavioral analysis.

- The datasets / benchmarks / applications parts are useful already, but they could be more actionable. For example, it would help to say a bit more clearly which datasets or benchmarks are most standard, which are newer, and how readers might choose between them.

---

> ### Author Response · Authors · 2026-05-11
> **Reply to Reviewer WxS8 (Part 1)**
>
> Thank you very much for your valuable review and constructive comments on our manuscript. To better facilitate and move forward the review process, we are providing our response to the comments we have received so far in advance.
>
> At present, we have received comments from two reviewers and have revised the manuscript accordingly. The latest version of the manuscript is therefore based on the feedback provided by these two reviewers. Once we receive the comments from the third reviewer, we may further update the manuscript to address any additional suggestions or concerns.
>
>
> **[Regarding: Novelty (Requested Changes 1)]**
>
> We thank the reviewer for the constructive feedback. We agree that a modality-based split alone may appear to be primarily an organizational convenience. However, we would like to clarify that our survey treats behavioral data representation learning as a unified problem and organizes the literature across four major behavioral data forms within a single coherent framework. A key observation underlying this taxonomy is that the same behavioral phenomenon can be represented in different data forms depending on modeling choices, and that this representational choice fundamentally determines the appropriate learning paradigm, available model families, and applicable evaluation protocols. Within each data form, we further provide finer-grained method classifications (e.g., supervised-only vs. supervised-plus for tabular deep learning; five self-supervised paradigms for event sequences; five temporal modeling strategies for CTDGs; and three LLM roles for textual data), and we systematically connect these method taxonomies to downstream applications, shared modeling modules, and modality-specific datasets and benchmarks.
>
> To address this concern more directly, we have revised the paper to better articulate these points, added **Section 9 Comparison with Existing Surveys** along with a dedicated comparison table (**Table 15**) that explicitly contrasts our coverage against representative adjacent surveys, and softened the novelty wording where appropriate to more accurately reflect our contribution as a systematic unification rather than a claim of entirely new conceptual categories.
>
> ---
>
> **[Regarding: Cross-Modal Synthesis (Requested Changes 2)]**
>
> We fully agree that the original manuscript would benefit from stronger synthesis across modalities. In the revised version, we have added a new **Section 6** **Cross-Modal Synthesis and Comparative Analysis** that is specifically designed to address this concern. This section provides an integrated analysis from four complementary angles: (1) shared technical building blocks across modalities, identifying convergent themes such as attention mechanisms, self-supervised pre-training, temporal encoding, and memory mechanisms, and analyzing how each modality adapts them differently; (2) core assumptions and trade-offs of each modality, examining what information each representation preserves and discards, along with cross-cutting tensions such as expressiveness vs. efficiency and interpretability vs. performance; (3) practical method selection guidance organized by task requirements, data characteristics, and application-modality alignment patterns observed in the literature; and (4) two consolidated cross-modal comparison tables (**Tables 6 and 7**) that systematically compare all four modalities along dimensions including entity representation, core assumptions, typical supervision settings, strengths, limitations, and application alignment. We believe this new section transforms the paper from four parallel mini-surveys into a more cohesive and unified narrative.
>
> ---
>
> **[Regarding: Critical Analysis and Trade-Offs (Requested Changes 3)]**
>
> We appreciate this suggestion and agree that the original manuscript was stronger on coverage than on critical comparison. We have made two types of revisions to address this. First, we have added the "**Summary and Takeaways**" paragraphs at the end of each major subsection. Each takeaway now includes explicit comparisons of when different method families within the modality are most and least effective, concrete practical guidelines, and identified open gaps and unresolved challenges. Second, the new **Section 6 Cross-Modal Synthesis and Comparative Analysis** contains dedicated discussions of cross-cutting trade-offs and modality selection guidance, providing readers with actionable advice on what works when and where the gaps remain, both within and across modalities.

---

> > ### Author Response · Authors · 2026-05-11
> > **Reply to Reviewer WxS8 (Part 2)**
> >
> > **[Regarding: Cross-Modal Summary Tables (Non-Critical 1)]**
> >
> > Following the reviewer's suggestion, we have added two cross-modal summary tables in the new Section 6. **Table 6** compares the four modalities along key dimensions: representation, core assumptions, information preserved, information lost, typical supervision, key strengths, key limitations, and best-fit applications. **Table 7** further evaluates individual method families within each modality along five practically oriented axes, scalability, label efficiency, temporal precision, interpretability, and cold-start handling, providing a fine-grained reference for model selection.
> >
> > **[Regarding: Positioning Relative to Existing Surveys (Non-Critical 2)]**
> >
> > We agree that the positioning could be sharper. In the revised manuscript, we have added **Section 9 Comparison with Existing Surveys** with **Table 15**, which systematically compares our survey against twelve representative adjacent surveys across multiple coverage dimensions. This comparison makes explicit that prior surveys typically provide either deep coverage of a single modality or broad discussion within a single application domain, but not a unified comparison across major behavioral data forms together with their associated method taxonomies, applications, and evaluation resources, which is the specific gap our survey aims to fill.
> >
> > **[Regarding: Actionability of Datasets and Benchmarks (Non-Critical 3)]**
> >
> > We appreciate this suggestion. Regarding datasets, we note that the datasets listed in our survey have been selected precisely because they are among the most commonly adopted in the community; their widespread use across multiple studies makes them reliable choices for evaluating and comparing methods regardless of their release date. The benchmark tables include publication years, which naturally indicate recency and allow readers to identify newer evaluation frameworks.

---

### Review · Reviewer_TUWo · 2026-06-05

**Summary Of Contributions:**

This is survey paper on behavioral data representation learning. The paper consist of multiple parts: (i, Sec 4) discussion of approaches for the main data modalities: tabular data, sequential data, graph data, and textual data, (ii, Sec 5) discussion of a number of application areas, covering specific work in these areas, (iii, Sec 6) a short cross-modal discussion, and (iv, Sec 7) key modules, such as time encoding, and finally (v, Sec 8) dataset and benchmarks. The paper focuses on summarizing the available literature, but does not include own experimental results (which is fine for a survey paper).

I feel this paper aims to do too much and, by doing so, achieves too little. I recommend that the authors reconsider the scope, focus, goals, and target audience of this survey.

Strength:

S1. Comprehensive. The survey covers a substantial amount of work in each of its main areas, all of which can be relevant when working with behavioral data.

S2. High-level overview. Provides an accessible high-level overview of the main areas and it's main approaches. The considered papers are grouped into a coarse-grained taxonomy, presented in overview tables and briefly discussed in text.

S3. Relevant topic. A thorough survey and guide for behavioral learning would be highly valuable for both practitioners and researchers working with such data.

Weaknesses:

W1. Lack of rigor. The paper focuses on "representation learning" for "behavioral data". It neither clearly defines and discusses both of these focus areas, however. In more detail, the paper does not clearly argue what behavioral data is and, more importantly, what it's unique properties and challenges are. Many of the considered approaches are not specific to behavioral learning, so that it remains unclear what this survey really aims to do. Likewise, the paper does not establish clear goals and tasks for representation learning in this area, and it discusses methods along the lines of techniques used, rather than relevant tasks and scenarios. In fact, most of the discussion appears to focus on prediction tasks rather than representation learning.

W2. Lack of focus. As indicated above, the scope of this survey is substantial. As the paper also states at the end (e.g., Tab 15), individual surveys for each of the areas do exist and cover these areas thoroughly. Instead of drawing from these surveys, this paper tries to present everything at once, instead of focusing on the particular concerns around behavioral data. It's also not clear who the target audience of this paper is: the required background knowledge varies wildly across individual sections (see W3).

W3. Lack of clarity. The paper generally presents a high-level overview of each area, which I like. It then, however, discusses a large number of papers in a "list style", essentially summarizing each paper one-by-one with a sentence each. I did not find these summaries particularly helpful and more often than not, it did not become clear to me what the key points are and why these papers are included in the survey and what their contribution really is. Likewise, the overview tables mainly list relevant work, but does not provide much more insight than grouping them by method used (e.g., RNN-based, TPP-based, MLP-based, ...).

W4. Lack of insight. Summarizing the above weaknesses, I ultimately feel that this paper does not provide a suitable guide for researchers and practitioners working with behavioral data; I did not gain much insight from it. It does not become clear where one would need to look to solve which problems. I feel that a focus on cross-modal learning, a more thorough summary of the individual strength and weaknesses of key approaches instead of individual papers, how they would work together for multi-modal problems (which appear to be common in behavioral learning), and what key challenges and directions for future work are would be much more insightful.

W5. Key approaches missing. The paper does not cover recent foundation models such as TabPFN for tabular data, KumoRFM for relational data, TimesFM for time-series data, or ULTRA for knowledge graphs. These models are already highly impactful, and they are highly relevant for the application areas considered in this survey. I view the complete omission of this line of work highly critical.

**Audience:**

Yes

**Audience Explanation:**

The survey clearly fits the scope of TMLR and, if done well, can be highly useful.

**Claims And Evidence:**

No

**Claims Explanation:**

I answered no here because the survey repeatedly makes statements and claims without backing them up by citations. A good example of this is Table 7 (cross model comparison): it contains the author's assessments along a number of dimensions, but it does not clearly argue for any of these assessments. There are also longer parts or even entire paragraphs without backing citations. Note that I am not saying that the author's assessments are off, but instead that they need more justification and thoughtful discussion.

**Requested Changes:**

I am not listing anything concrete here, as I feel this survey needs a complete repositioning. My key thoughts and concerns are summarized above.

---

> ### Author Response · Authors · 2026-06-16
> **Reply to Reviewer TUWo (Part 1)**
>
> We thank the reviewer for the constructive feedback. In the revision, we made substantial changes to clarify the scope, strengthen the conceptual framing, mitigate list-style presentation where it obscured insight, and better justify our claims.
>
> In addition to the manuscript revisions, we would like to clarify the following points：
>
> **[Regarding: W1. Lack of rigor]**
>
> We revised the early sections to explicitly define behavioral data and behavioral data representation learning. In particular, we now emphasize that behavioral data are characterized by recurring properties: temporal situatedness, relationality, heterogeneity, incompleteness, sparsity, and non-stationarity. We further map these properties to representation requirements: feature-interaction modeling, temporal state tracking, continuous-time modeling, relational aggregation, semantic encoding, and cross-modal alignment. This mapping clarifies why some general representation learning methods are included: our criterion is not whether a method was designed exclusively for behavioral data, but whether its assumptions address a recurring behavioral-data challenge.
>
> We also clarified the relationship between prediction tasks and representation learning. Many behavioral representation methods are trained or evaluated through objectives such as next-event prediction, classification, ranking, link prediction, reconstruction, or language modeling. We now explicitly treat these objectives as mechanisms for shaping and evaluating representations, rather than as the final scope of the survey.
>
> **[Regarding: W2. Lack of focus]**
>
> We agree that the survey has a broad scope. This breadth is intentional: our goal is to provide a comprehensive entry point into behavioral data representation learning, covering the major data forms, modeling paradigms, applications, datasets, and benchmarks that a reader may need to navigate the area. The main target audience includes researchers who want to understand the existing landscape of behavioral data representation learning, researchers looking for new research opportunities across modalities and applications, and practitioners who want to apply representation learning methods to real behavioral systems. We also hope the survey is useful for readers from adjacent areas, such as recommendation, healthcare, finance, social computing, and graph learning, who may be familiar with one modality but want to understand how related behavioral representation problems are handled elsewhere.
>
> In this revision, we preserve the original high-level taxonomy and structure because we want the survey to remain easy to search and navigate. A reader should be able to quickly locate a target method family, modality, application scenario, dataset, or benchmark. At the same time, we strengthened the interpretive layer around this structure: the revised text now explains the representation assumptions, preserved signals, limitations, and trade-offs behind each family, so that the survey serves not only as a reference catalogue but also as a guide for extending research.
>
> **[Regarding: W3. Lack of clarity]**
> We understand the concern about list-style writing. For a survey with broad coverage, concise method-by-method descriptions are sometimes useful because they allow readers to quickly identify what a particular method does, what problem it targets, and where it fits in the literature. However, we agree that such descriptions alone are not sufficient, because they can obscure the main story and make it difficult to see why the methods are included.
>
> To address this, we added a stronger synthesis layer throughout the paper. Specifically, we added or revised paragraphs that summarize each method family in terms of: (1) the behavioral representation problem it addresses, (2) the type of behavioral signal it preserves, such as aggregated attributes, temporal order, exact timing, relational context, or semantic intent, (3) the scenarios where it is most suitable, and (4) the trade-offs or limitations it introduces. We also revised “Summary and takeaways” paragraphs and cross-paragraph synthesis statements after many subsections to connect individual papers into broader methodological trends.
>
> We also added synthesis statements in the application section so that domains such as recommendation, healthcare, social media, finance, and gaming are discussed in terms of decision-critical behavioral signals rather than as lists of representative papers.
>
> We still keep concise method tables because they are useful for navigation in a broad survey, but the main text has been revised to connect individual papers into broader methodological trends and representation-centered insights.

---

> > ### Author Response · Authors · 2026-06-16
> > **Reply to Reviewer TUWo (Part 2)**
> >
> > **[Regarding: W4. Lack of insight]**
> >
> > We added a dedicated cross-modal synthesis and comparative analysis section. This section compares shared technical building blocks, core assumptions, modality-specific trade-offs, and method-selection criteria. It explains, for example, when tabular representations are sufficient, when event sequences are more appropriate, when dynamic graphs become necessary, and when LLM-based semantic representations are useful. We also expanded the future directions to emphasize principled multi-modal fusion, behavioral foundation models, non-stationarity, cold-start transfer, representation-aware evaluation, and interpretability/fairness/privacy. These additions are intended to make the survey more useful as a guide for choosing, comparing, and combining representation learning methods.
> >
> > **[Regarding: W5. Key approaches missing]**
> >
> > We thank the reviewer for pointing out this important omission. We have revised the manuscript to explicitly cover the recent foundation-model line of work mentioned by the reviewer. Specifically, we now discuss TabPFN for tabular data, KumoRFM for relational data, TimesFM for time-series data, and ULTRA for knowledge-graph reasoning in the relevant sections and future-direction discussion. We also added TabPFN, KumoRFM, and TimesFM to the corresponding method tables.
> >
> > In the revised discussion, we explain why these models are relevant to behavioral data representation learning: TabPFN uses prior-data pre-training and in-context learning to support rapid adaptation on tabular prediction tasks; KumoRFM extends in-context foundation modeling to multi-table relational data; TimesFM provides a decoder-only foundation model for time-series forecasting; and ULTRA points toward transferable knowledge-graph reasoning through relation-aware inductive generalization. At the same time, we clarify that applying these foundation models to behavioral data still requires attention to behavioral-specific challenges, including population shift, schema shift, non-stationarity, temporal grounding, and relational grounding.
> >
> > **[Regarding: Method assessments]**
> >
> > We agree that qualitative comparisons require clearer justification. Some statements in the previous version reflected our synthesis of the cited literature rather than direct claims made by a single paper, but we agree that these syntheses should be more clearly grounded. In the revision, we revisited the relevant original references and added citations where our statements were based on specific mechanisms, empirical observations, or known limitations from prior work. We also revised several broad statements to make clear when they are our qualitative synthesis rather than direct benchmark conclusions.
> >
> > For the cross-modal comparison table, we added an explicit explanation of the rating criteria and a rating-rationale appendix. The ratings are now presented as qualitative syntheses of the mechanisms and empirical findings discussed throughout the survey, rather than as absolute benchmark scores. We explain the criteria behind scalability, label efficiency, temporal precision, interpretability, and cold-start handling, and we provide citations supporting these assessments.
> >
> > **Overall,** the revision aims to reposition the paper from a broad catalogue of methods toward a representation-centered survey of behavioral data: what makes behavioral data distinctive, what each representation family preserves or loses, how different modalities complement each other, and how researchers and practitioners can choose or extend methods for specific behavioral-analysis scenarios.

---

### Author Response · Authors · 2026-06-16
**To Editors and Reviewers**

Dear Editors and Reviewers,

Thank you very much for the time and effort you have devoted to reviewing our manuscript.

Since the review process began, we have made ***two rounds of revisions*** to the manuscript.

- The *first* revision, marked in *red* in the manuscript, mainly addressed the concerns raised by **Reviewer QAmS** and **Reviewer WxS8**.

- The *second* revision, which is the latest version and is marked in *blue*, mainly focuses on addressing the comments from **Reviewer TUWo**.

We believe that the second revision also helps to further address some of the concerns raised by **Reviewer QAmS** and **Reviewer WxS8**. Therefore, we kindly ask you to refer to the latest version of the manuscript when evaluating our paper.

Thank you again for your valuable comments and constructive suggestions.



Best regards,

Authors of the Paper6938